# Transformers are Efficient Compilers, Provably

**Xiyu Zhai**
University of Washington
xiyuzhai@cs.washington.edu

**Runlong Zhou**
University of Washington
vectorzh@cs.washington.edu

**Liao Zhang**
University of Innsbruck
zhangliao714@gmail.com

**Simon S. Du**
University of Washington
ssdu@cs.washington.edu

## Abstract

Transformer-based large language models (LLMs) have demonstrated surprisingly robust performance across a wide range of language-related tasks, including programming language analysis and generation. In this paper, we take the first steps towards a formal investigation of using transformers as compilers from an expressive power perspective. We introduce a representative programming language, **Mini-Husky**, which encapsulates key features of modern C-like languages. We show that if the input code sequence has bounded depth in both Abstract Syntax Tree (AST) and type inference, then the number of parameters required by transformers depends only on the *logarithm of the input sequence length* to handle compilation tasks, such as AST construction, symbol resolution, and type analysis. A significant technical challenge stems from the fact that transformers operate at a low level, where each layer processes the input sequence as raw vectors without explicitly associating them with predefined structure or meaning. Our primary contribution is the development of a domain-specific language, **Cybertron**, which generates formal proofs of the transformer's expressive power for compiler tasks. We further establish that recurrent neural networks (RNNs) require a linear number of parameters relative to the input sequence, leading to an exponential separation between transformers and RNNs. Finally, we empirically validate our theoretical results by comparing transformers and RNNs on compiler tasks within **Mini-Husky**.

## 1 Introduction

Transformers (Vaswani, 2017) have demonstrated remarkable proficiency across various domains, achieving near-expert performance in solving International Mathematical Olympiad problems (Google Deepmind, 2024) and excelling in complex reasoning tasks in science, coding, and mathematics (OpenAI, 2024a). They also handle routine coding tasks with high precision and have been integrated into code editors to significantly boost programmers' productivity (cur, 2024; Taelin, 2023a). Despite these advancements, the full extent of their underlying capabilities remains only partially understood.

In this paper, we aim to deepen our understanding of transformers' abilities to perform compilation tasks. Empirically, transformer-based LLMs have shown rapid progress in code generation and compilation. For example, MetaLL (Cummins et al., 2024) enables LLMs to optimize code by interpreting compiler intermediate representations (IRs), assembly language, and optimization techniques. Gu (2023) highlights the ability of LLMs to generate high-quality test cases for Golang compilers. Surprisingly, Taelin (2023b) demonstrates that models like Sonnet-3.5 can compile legacy code into modern languages like TypeScript, outperforming the now obsolete AgdaJS compiler (Agda Development Team, 2024). In parallel, Armengol-Estapé & O'Boyle (2021) investigates the application of a sequence-to-sequence Transformer model for translating C source code into corresponding x86 assembly instructions, demonstrating its performance through training and empirical evaluation. Guo

& Moses (2022) highlights the superior performance of LLMs in optimizing low-level code generated from C programs, surpassing traditional compiler-based optimization techniques.

To formally study this problem in a controlled setup, we designed a C-like programming language called **mini-husky**, which encapsulates key features of modern C-like languages such as (Flanagan, 2011) and Rust (Klabnik & Nichols, 2023). We focus on three representative compilation tasks: abstract syntax tree (AST) construction, symbol resolution, and type analysis. The AST is a recursive structure that represents the input as a tree. From the perspective of programming language design, the AST is considered the true representation of the input, with the textual code serving merely as a convenient interface for human users (Alfred et al., 2007). All syntactic and semantic processing can then be interpreted as specific operations on these trees. Symbol resolution involves verifying the validity of references to entities and flagging errors for undefined symbols. Type analysis encompasses both type inference, which assigns types to variables without explicit annotations, and type checking, which identifies mismatches between actual and expected types.

We demonstrate that under the *clean code principle* (Martin, 2008), transformers can efficiently perform AST construction, symbol resolution, and type analysis, where efficiency means that these tasks can be conducted by transformers with a number of parameters that scale logarithmically with the input code length. To our knowledge, this is the first theoretical demonstration that transformers can function as compilers in a parameter-efficient manner.

We further compare transformers and recurrent neural networks (RNNs). By connecting the type analysis task with the associative recall, we show even under the *clean code principle* (Martin, 2008), RNNs require a memory size that scales *linearly* with the input sequence length to successfully perform type analysis. Consequently, for type analysis in compilation, transformers can be *exponentially more efficient* than RNNs. We also empirically validate our theoretical findings by demonstrating the superiority of transformers.

**Technical Challenges and Our Technique.**

Proving that transformers can perform compilation tasks presents several challenges:

- **Transformers operate at too low a level**. Transformers process sequences of floating-point vectors, akin to raw bits in computers, and proving their ability to perform specific tasks is similar to writing specialized parallel machine code. Previous work (Yao et al., 2021) often resorts to graphical illustrations for readability, even for basic tasks.
- **Compilers are exceedingly high-level**. Compilers are among the most complex programming endeavors of our time. Compilation involves numerous sophisticated procedures, some of which are undecidable or computationally expensive, such as code optimization (Alfred et al., 2007)) and type analysis (Pierce, 2002). For example, type analysis in complex type systems poses significant challenges, often requiring the development of advanced logical frameworks (Dunfield & Krishnaswami, 2019).

To overcome these challenges, we design a domain-specific language (DSL) called **Cybertron** to serve as the proof vehicle, i.e., a major part of our proof consists of reasoning about type-correct code in Cybertron that represents a transformer. Without using **Cybertron**, writing an equivalent natural language proof would be too complex and intractable. Using code to prove propositions is not new to computer science; it is, in fact, the norm in interactive theorem proving (ITP) (Harrison et al., 2014). To the best of our knowledge, we are the first to apply this approach to understanding neural networks.

**Contributions.** We summarize our contributions below:

- **A testbed for compilation tasks**: We introduce **Mini-Husky**, a simple yet representative C-like programming language, designed to formally assess transformers' capabilities in programming language processing. We anticipate that **Mini-Husky** will become a standard testbed for this purpose.
- **Expressive power theory of transformers for several compilation tasks**: We provide a formal proof that, when the input code sequence has bounded AST depth and inference depth, the number of parameters in transformers only needs to scale logarithmically with the input sequence length to handle compilation tasks such as AST construction,

symbol resolution, and type analysis. To the best of our knowledge, this is the first study exploring the power of transformers for these compilation tasks.

- **Transformers vs. RNNs**: Theoretically, we demonstrate a negative result, showing that the number of parameters in RNNs must scale linearly with the input sequence length to perform type analysis correctly. This result establishes an exponential separation between transformers and RNNs. We further empirically confirm the advantage of transformers for the type analysis task.
- **A Domain-Specific Language for Proofs**: Given the challenges in formal proofs, we design a domain-specific language, **Cybertron**, to serve as a proof vehicle. We believe that **Cybertron**, and the general approach of using DSLs for analysis, can have broader applications in understanding transformers and other architectures.

## 2 Related Work

**Expressive Power of Transformers.** A line of work studies the expressive power of attention-based models. One direction focuses on the universal approximation power (Yun et al., 2019; Bhattamishra et al., 2020b;c; Dehghani et al., 2018; Pérez et al., 2021). More recent works present fine-grained characterizations of the expressive power for certain functions in different settings, sometimes with statistical analyses (Edelman et al., 2022; Elhage et al., 2021; Likhosherstov et al., 2021; Akyürek et al., 2022; Zhao et al., 2023; Yao et al., 2021; Anil et al., 2022; Barak et al., 2022; Garg et al., 2022; Von Oswald et al., 2022; Bai et al., 2023; Olsson et al., 2022; Akyürek et al., 2022; Li et al., 2023; Hao et al., 2022; Pérez et al., 2019; Strobl, 2023; Chiang et al., 2023; Wei et al., 2022; Wang et al., 2022; Feng et al., 2023; Li et al., 2024). There are also characterizations of transformers to be as powerful as universal computers if put in a looped context (Giannou et al., 2023). The most related one is Yao et al. (2021) where the authors prove constructively that bounded depth Dyck language can be recognized by encoder-only hard attention transformers, which has similarities to our settings of bounded depth programming language recognized encoder-only hard attention transformers. The major difference is that we introduce concepts and tasks from programming language theory Pierce (2002) to study the semantic powers of transformers.

**Transformers vs. RNN.** It is important to understand the comparative advantages and disadvantages of transformers against RNNs. Empirically, synthetic experiments have shown an advantage of transformers against RNNs for long range tasks (Bhattamishra et al., 2023; Arora et al., 2023). Theoretically, there has been a rich line of work focusing on comparing transformers and RNNs in terms of recognizing formal languages (Bhattamishra et al., 2020a; Hahn, 2019; Merrill et al., 2021), which show that the lack of recursive structure of transformers prevent them from recognizing some formal languages that RNNs can recognize. However, the gap can be mitigated when we consider the bounded length of input or bounded grammar depth (Liu et al., 2022; Yao et al., 2021), which is quite reasonable in practice and is used in this paper. On the other side, prior work (Jelassi et al., 2024; Wen et al., 2024) proves a representation gap between RNNs and Transformers in repeating a long sequence. In summary, it is somehow intuitive that recursive structures with limited memory perform badly at tasks which requires information retrieval. Our paper shows that semantic analysis for programming languages is such a task.

**DSLs for Transformers.** We note that we are not exactly the first one to employ a domain-specific language to understand the expressive powers of transformers. Previously, DSLs with simple typings like RASP (Weiss et al., 2021) were proposed to prove constructively that transformers can do various basic sequence-to-sequence operations. Lindner et al. (2023) writes a compiler that compiles RASP into actual transformers, Friedman et al. (2023) shows that RASP can be learned, and Zhou et al. (2023) uses RASP to prove that simple transformers can perform certain algorithms. The major difference between RASP and our DSL **Cybertron** is that **Cybertron** has a powerful algebraic type system that helps prove complicated operations beyond simple algorithms.

## 3 Preliminaries

The major innovation in the transformer architecture is that it uses self-attention solely without a conjunction with a recurrent network (Vaswani, 2017), which processes input tokens in a distributed manner. This capability enables the model to handle long-range dependencies, a crucial feature for language tasks. We use hard attention and simplified position encoding to simplify our theoretical reasoning.

**Attention.** In practice, attention heads use **soft attention**. Given model dimension $d_{\text{model}}$, number of heads $H$, and a finite set of token positions Pos, an attention layer with simplified position encoding is defined as a function $f_{\text{attn}} : \mathbb{R}^{\text{Pos} \times d_{\text{model}}} \to \mathbb{R}^{\text{Pos} \times d_{\text{model}}}$ given by

$$\forall p \in \text{Pos}, \quad f_{\text{attn}}(X)_p := W_O \text{ Concat} \left( \text{Attn}^{(1)}(X)_p, \ldots, \text{Attn}^{(H)}(X)_p \right), \quad (1)$$

where the $h$th attention head is defined using soft attention as: $\text{Attn}^{(h)}(X)_p := \sum_{p' \in \text{Pos}} \alpha_{p,p'}^{(h)} V_{p'}^{(h)}$. The attention weights $\alpha_{p,p'}^{(h)}$ given by: $\alpha_{p,p'}^{(h)} = \frac{\exp\left( Q_p^{(h)\top} K_{p'}^{(h)} + \lambda^{(h)\top} \Psi_{p'-p} \right)}{\sum_{p'' \in \text{Pos}} \exp\left( Q_p^{(h)\top} K_{p''}^{(h)} + \lambda^{(h)\top} \Psi_{p''-p} \right)}$, where $W_O \in \mathbb{R}^{d_{\text{model}} \times d_{\text{model}}}$ are trainable parameters, $Q_p^{(h)}, K_p^{(h)}, V_p^{(h)} \in \mathbb{R}^{d_{\text{model}}/H}$ are linear transformations of $X_p$, $\lambda^{(h)} \in \mathbb{R}^2$ depends on the head, and $\Psi_q = \begin{pmatrix} q \\ 1_{q>0} \end{pmatrix} \in \mathbb{R}^2$ accounts for relative position.

For theoretical convenience, we use average hard attention, commonly used in theoretical analysis of transformer (Yao et al., 2021; Hao et al., 2022; Pérez et al., 2019). Average hard attention can be viewed as the limit of soft attention when the attention logits become infinitely large. The hard attention head is defined as:

$$\text{Attn}^{(h)}(X)_p := \frac{1}{|S_p|} \sum_{p' \in S_p} V_{p'}^{(h)}, \quad \text{where } S_p = \arg \max_{p' \in \text{Pos}} \left( Q_p^{(h)\top} K_{p'}^{(h)} + \lambda^{(h)\top} \Psi_{p'-p} \right) \quad (2)$$

In other words, hard attention selects the positions $p'$ that maximize the attention score for each position $p$, and averages the corresponding value vectors $V_{p'}^{(h)}$.

Although our analysis employs hard attention, it can be viewed as a theoretically grounded approximation of soft attention in settings where a single attention logit significantly outweighs the others. In such cases, the softmax function assigns exponentially diminishing weights to non-maximal logits as the gap increases. In the limit, soft attention approaches a one-hot distribution, effectively becoming hard attention. This approximation is commonly adopted in theoretical studies (Yao et al., 2021) and aligns with empirical observations, particularly when attention distributions are sharply concentrated due to scaling effects or learned sparsity.

**Feed-Forward Layer.** Given model dimension $d_{\text{model}}$, and a finite set of token positions Pos, a feed-forward layer is a fully connected layer applied independently to each position, defined as a function $f_{\text{ffn}} : \mathbb{R}^{\text{Pos} \times d_{\text{model}}} \to \mathbb{R}^{\text{Pos} \times d_{\text{model}}}$ given by

$$\forall p \in \text{Pos}, \quad f_{\text{ffn}}(X)_p = W_2 \sigma_{\text{ReLU}} \left( W_1 X_p + b_1 \right) + b_2, \quad (3)$$

where $W_1 \in \mathbb{R}^{d_{\text{ffn}} \times d_{\text{model}}}$ and $W_2 \in \mathbb{R}^{d_{\text{model}} \times d_{\text{ffn}}}$ are trainable weight matrices, $b_1 \in \mathbb{R}^{d_{\text{ffn}}}$ and $b_2 \in \mathbb{R}^{d_{\text{model}}}$ are trainable bias vectors, $d_{\text{ffn}}$ is the hidden dimension of the feed-forward layer, chosen to be $2d_{\text{model}}$, as commonly used in practice, $\sigma_{\text{ReLU}}$ is the ReLU activation function.

**Encoder-Only Transformer.** Encoder-only transformers consist solely of the encoder stack, making them ideal for tasks like classification, regression, and sequence labeling that do not require sequence generation. Each encoder layer includes a multi-head self-attention mechanism and a feed-forward network, allowing the model to capture complex dependencies and contextual information.

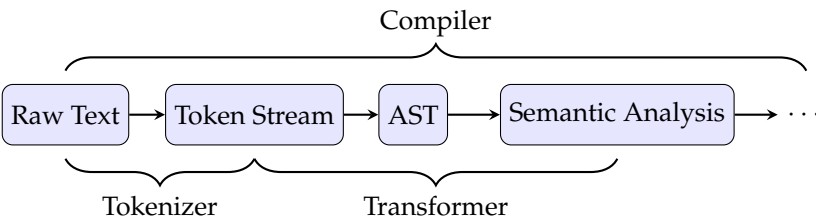

Figure 1: Programming language processing pipeline.

One can define it using the following recurrence,

- The input is given by: $X^{(0)} = X$.
- For each layer $l = 1, 2, \ldots, L$:
  - Compute attention output: $\hat{X}^{(l)} = X^{(l-1)} + f_{\text{attn}}^{(l)}\left(X^{(l-1)}\right)$,
  - Compute feed-forward output: $X^{(l)} = \hat{X}^{(l)} + f_{\text{ffn}}^{(l)}\left(\hat{X}^{(l)}\right)$.

In the above, $f_{\text{attn}}^{(l)}$ are the attention layers, and $f_{\text{ffn}}^{(l)}$ are the feed-forward layers, with the same model dimension $d_{\text{model}}$, number of heads $H$, and set of token positions Pos. For simplicity, layer normalization is ignored. See Appendix C for full details of transformers and other architectures.

## 4 Programming Language Processing and The Target C-Like Language: Mini-Husky

Recently, transformers have expanded to support code analysis and generation (Nijkamp et al., 2023; Chen et al., 2021; Anysphere, 2023). Programming languages offer a cleaner foundation for studying language understanding, as their syntactic and semantic tasks are precisely defined. To formally study the language processing capabilities of transformers, we design **Mini-Husky**, a representative mix of modern C-like languages with strong typing and typical syntactic features. It supports user-defined types (e.g., structs, enums) and enforces strict type equality, disallowing implicit conversions. Lexical scoping, including shadowing, ensures proper variable accessibility based on block structures, type inference, and type checking. These features make compiling *Mini-Husky* a representative task to evaluate transformers' capabilities in syntactic and semantic tasks like symbol resolution and type checking. See Appendix E for the full details of **Mini-Husky**.

The standard pipeline of processing programming languages is shown in Figure 1 (Alfred et al., 2007). The input text is initially decomposed into a sequence of lexical elements such as literals, identifiers, punctuation symbols, and language keywords; this sequence is referred to as the *token stream*. Subsequently, the token stream is parsed into a hierarchical, tree-structured representation that reflects the grammatical structure of the input. Finally, syntactic and semantic analyses are conducted on the resulting tree to verify correctness and extract meaningful information. Afterward, an intermediate language program is generated based on the syntactic and semantic analysis, which is further optimized and finally transformed into targeted machine code. To simplify the presentation, we assume the tokenizer has been provided a priori. Below we describe the programming language processing tasks investigated in the paper.

**Abstract Syntax Tree Construction.** Abstract Syntax Tree (AST) is a hierarchical, tree-like representation of the syntactic structure of source code in a programming language. Unlike the raw text of the code, the AST abstracts away surface syntax details, capturing the essential elements and their relationships in a structured form. Each node in the AST corresponds to a construct occurring in the source code, such as expressions, statements, or declarations. This representation is central to various stages of language processing, enabling efficient syntax checking, semantic analysis, and code generation. The formal definition of ASTs is standard in the programming language literature but is lengthy, so we

defer it to Appendix A. The AST construction task's final output is the collection of all AST nodes. We will show transformers can construct AST efficiently.

**Symbol Resolution.** In programming languages, symbols are functions, types, generics, variables, macros, etc. They are defined somewhere and can be used by referring to the corresponding identifier or path in a certain scope. The scope can be within a certain tree of modules, or within a certain curly braced scope within one module. For simplicity, we only consider curly braced scope. In **Mini-Husky**, the following showcases symbol resolution.

```
1   pub fn f() {
2       fn f1() {}
3
4       let a = 1;
5       let x = a;
6       let a = 2;
7       {
8           let a = 3;
9           { let a = 4; }
10          let y = a;
11      }
12      let z = a;
13  }
14
15  fn g() { f() }
```

The outer function $f$ is accessible everywhere in the body of function $g$. However, the inner function $f_1$ can only be used inside the body of $f$ as it is defined within the body. For variables with the same identifier `a`, `a` from line 4 is accessible from line 5, the second is accessible from line 12, the third is accessible from line 10, and the fourth is not accessible from anywhere. Thus $x = 1, y = 3, z = 2$.

The output of the **symbol resolution** task is the collection of symbol resolution results on all applicable tokens. More concretely, the output is a sequence of values of type `Option<SymbolResolution>` where `Option<SymbolResolution>` is the type `SymbolResolution` with a null value added for non-applicability and `SymbolResolution` is the type storing the result of the symbol resolution, being either a success with a resolved symbol of type `Symbol` or a failure with an error of type `SymbolResolutionError`. We shall prove that transformers can do symbol resolution and that attention is crucial.

**Type Analysis.** In general, types are essential for conveying the intended usage of the written functions and specifying constraints. As a first exploration of this topic, we try to make the type analysis in **Mini-Husky** as simple as possible yet able to bring out the essential difficulty. The type system consists of four sequential components: (1) *Type definition*, (2) *Type specification*, (3) *Type inference*, and (4) *Type checking*. Due to the page limit, here we only introduce (4) *Type checking* because it is the final step and this is a crucial step which separates transformers and RNNs. See Appendix E.1 for details of (1) *Type definition*, (2) *Type specification*, and (3) *Type inference*.

Type checking ensures that the typed expressions agree with its expectations. For simplicity, we do not allow implicit type conversion, so the agreement means exact equality of types. The arguments of function calls are expected to have types according to the definition of the function. The operand type of field access must be a struct type with a field of the same name. The type of the last expression of the function body or the expr in the return statement must be equal to the return type of the function. For variables defined in the `let` statement, If the types are annotated, the types of the left-hand side and right-hand side should be in agreement.

```
1   //  Type Error: the return type is `i32`, yet the last expression is of type `f32`
2   fn f(a: i32) -> i32 { 1.1 }
3
4   struct A { x: i32 }
5
6   fn g() {
7       // Type Error: `x` is of type f32 but it's assigned by a value of type `i32`
8       // Type Error: the first argument of `f` is expected to be of type `i32` but gets a float literal instead
9       let x: f32 = f(1.1);
10      // Type Error: no field named `y`
```

```
11     let y = A { x: 1 }.y;
12  }
```

The above incorporates typical examples of type disagreements that count as type errors. A compiler should be able to report these errors.

The **type analysis** task's final output is the collection of all type errors. More concretely, the output is a sequence of `Option<TypeError>` , where `Option<TypeError>` denoted the type `TypeError` will a null value added and `TypeError` is the type storing the information of a type error. The position of type errors agrees with the source tokens leading to these errors.

## 5 Expressive Power of Transformers as Efficient Compilers

In this section we discuss main theoretical results about the expressive power of transformers to perform compilation tasks: AST construction, symbol resolution, and type analysis. In Section 5.4, we discuss **Cybertron**, a DSL specifically designed for our proof.

### 5.1 Abstract Syntax Tree Construction

We start with a definition that characterizes low-complexity code.

**Definition 1** (code with Bounded AST-Depth). *Let* $\text{MiniHusky}_D$ *be the set of token sequences that can be parsed into valid ASTs in **Mini-Husky** with a depth less than D.*

$D$ in the above definition is small for all programming languages in practice, and a linear dependency on $D$ is acceptable, but the linear dependency on the length of the token sequence $L$ is not.

The bounded depth assumption is both practically reasonable and theoretically grounded. In practical software development, the *clean code principle* (Martin, 2008) requires one to write code with as little nested layer as possible for greater readability. Readability is of the utmost importance because "Programs are meant to be read by humans and only incidentally for computers to execute" (Abelson et al., 1996). Moreover, while real-world code can exhibit significant depth—and in extreme cases, can even be NP-hard to analyze—in practice, most codebases maintain relatively shallow nesting for the sake of readability and maintainability. For example, although the Linux kernel comprises tens of millions of lines of code, the majority of its functions are short and have limited structural depth. Theoretically, the bounded-depth assumption is essential for understanding the limitations of transformer-based models. The transformer architecture inherently supports only finite-depth computation. This limitation has been discussed in prior work, such as (Yao et al., 2021), which highlights the importance of architectural depth when modeling more complex behaviors.

This assumption of bounded hierarchical depth is not limited to just programming languages, but is often seen as applicable to natural languages (Frank et al., 2012; Brennan & Hale, 2019; Ding et al., 2017), motivating Yao et al. (2021) to have a similar boundedness assumption. Below is the main result for AST construction using transformers.

**Theorem 1.** *There exists a transformer encoder of model dimension and number of layers being $O(\log L + D)$ and number of heads being $O(1)$ that represents a function that maps any token sequence of length L in $\text{MiniHusky}_D$ to its abstract syntax tree represented as a sequence.*

We note $\log L$ is small because 64-bit computers can only process context length at most $2^{64}$ and $D$ is small by assumption. Therefore, there exists a transformer with an almost constant number of parameters that is able to process comparatively much longer context length. See full proof details in Appendix F.

### 5.2 Symbol Resolution

Next, we show that transformers can effectively perform symbolic resolution as $\log L$ and $D$ are almost constant as compared with context length $L$. The proof details are in Appendix G.

**Theorem 2.** *There exists a transformer encoder of model dimension and number of layers being $O(\log L + D)$ and number of heads being $O(1)$ that represents a function that maps any token sequence of length $L$ in* $\text{MiniHusky}_D$ *to its symbol resolution represented as a sequence of values of type* `Option<SymbolResolution>` *.*

## 5.3 Type Analysis

We need an additional definition to characterize the complexity of code for type analysis.

**Definition 2** (code with Bounded AST-Depth and Type-Inference-Depth)**.** *We use* $\text{MiniHuskyAnnotated}_{D,H}$ *to denote the subset of* $\text{MiniHusky}_D$ *with the depth of type inference no more than H. The depth of type inference is the number of rounds of computation needed to infer all the types using the type-inference algorithm (described in Appendix E.1).*

In practice, $H$ is significantly smaller than the context length $L$ for reasonably written code because it is upper bounded by the number of statements in a function body which is required to be small according to the *clean code principle* (Martin, 2008). Below, we present the main result of using transformers for type analysis. See full details in Appendix H.

**Theorem 3.** *For $L, D, H \in \mathbb{N}$, there exists a transformer encoder of model dimension, and number of layers being $O(\log L + D + H)$ and number of heads being $O(1)$ that represents a function that maps any token sequence of length $L$ in* $\text{MiniHuskyAnnotated}_{D,H}$ *to its type errors represented as a sequence of values of type* `Option<TypeError>` *.*

## 5.4 Proof Vehicle: Cybertron, a Domain-Specific Language

Here we highlight our main proof technique. Proving that transformers can express complex algorithms and software like compilers is a significant challenge due to the inherent differences between how transformers operate and the nature of high-level tasks they are expected to perform. Transformers process input at a low level, where each layer manipulates raw token sequences as vectors without predefined structure or meaning. However, high-level tasks—such as constructing ASTs and performing type and symbol analysis—require handling complex, structured information that depends on long-range relationships and interactions across the input. Bridging the gap between this raw, unstructured processing and the structured, multi-step logic required for these tasks introduces significant difficulty. Compilers, for instance, typically rely on rule-based, step-by-step operations that are abstract and sequential, which transformers must simulate through their attention mechanisms and feedforward layers. The challenge is further compounded by the need to formally prove that transformers can handle such tasks efficiently and accurately, despite operating in a fundamentally different manner. To address these challenges, we propose a domain-specific language (DSL) called **Cybertron**, which allows us to *systematically prove* that transformers are capable of expressing complex algorithms while maintaining sufficient readability.

A key feature of **Cybertron** is its expressive type system, which provides strong correctness guarantees. The type system ensures that every value is strongly typed, making it easier to reason about function composition and ensuring the validity of our proofs. This type system is crucial for managing how transformers represent and manipulate both local and global types—where local types correspond to individual tokens and global types refer to sequences of tokens, encapsulating broader program information.

What transformers output is a representation in sequences of vector of sequences of values in these types. As types are mathematically interpreted in this paper as a discrete subset of a vector space, **Cybertron** allows us to construct transformers with automatic value validity guarantees if the **Cybertron** code is type-correct.

In **Cybertron**, complex functions are broken down into "atomic" operations through propositions on function compositions and computation graphs (Propositions 11,13,14,2). It is straightforward to prove that these "atomic" operations are representable by transformers, either by feedforward layers or attention layers. For example:

- **Feedforward layers:** boolean operations like AND (Proposition 6), OR (Proposition 7), or NOT (Proposition 5), or operations over option types like `Option::or` (Proposition 9) being applied to each token in a sequence.
- **Attention layers**: operations that require information transmission between tokens such as `nearest_left` and `nearest_right` that collect for each token the nearest left/right non-nil information (Proposition 15).

This approach allows us to break down complex operations into primitive tasks that transformers can simulate. Feedforward layers handle local operations on individual tokens, while attention layers manage long-range dependencies and interactions between tokens, simulating the multi-step reasoning required for higher-level tasks.

**Cybertron**'s expressive type system and function composition framework help bridge the gap between the low-level processing transformers perform and the high-level reasoning necessary for complex tasks like compilation. For full details, including the mathematical foundations of **Cybertron**'s type system and function composition, see Appendix D.

# 6 Comparisons between Transformers and RNN

Now we compare transformers and RNNs from both theoretical and empirical perspectives.

## 6.1 A Lower Bound for RNNs for Type Checking

Previously, it has shown that RNN is provably less parameter efficient than transformers for associative recall (Wen et al., 2024). Observing that the type checking step covers associative recall, we obtain the following lower bound for RNNs.

**Theorem 4.** *For $L, D, H \in \mathbb{N}$, for any RNN that represents a function that maps any token sequence of length $L$ in* $\text{MiniHuskyAnnotated}_{D,H}$ *with $D, H = O(1)$ to its type errors represented as a sequence of values of type* `Option<TypeError>` *, then its state space size is at least $\Omega(L)$.*

Theorem 3 and 4 give a clear separation between transformers and RNNs in terms of the compilation capability. If the input codes satisfy $D, H \ll L$, which is typically the case under the *clean code principle* (Martin, 2008), then transformers at most need $O\left((\log L + D + H)\right)$ number of parameters, which is significantly smaller what RNNs require, $\Omega(L)$.

## 6.2 Empirical Comparison between Transformers and RNNs

We validate our theoretical results by conducting experiments on synthetic data.

**Dataset construction.** The synthetic dataset is parameterized by $n$ (the number of data pieces), $f$ (the number of functions in a data piece), $a$ (the maximum number of arguments of any function), $c$ (the maximum number of function calls involved in any function), $d$ (the minimum distance between the declaration and the first call of a function, as well as the minimum distance between its consecutive calls), $v$ (the probability of using a variable in a function call), and $e$ (the error rate of using an incorrect type in a function call).

The names of the functions are drawn randomly and uniquely from a list of English words. For each of the arguments of any function, its symbol is randomly drawn from another list of English words and its type is randomly drawn from {Int, Float, Bool}. All the called functions must be declared and not called by at least $d$ functions ahead of the current one. *For each argument of any function call,* with probability $v$, the argument variable of the enclosing function is used regardless of its type, with probability $(1 - v)(1 - e)$, a literal of the correct type is used, and with probability $(1 - v)e$ an incorrect type literal is used. For integers, the literals are from $\{0, 1, \ldots, 99\}$; for floats, the literals are from $\{0.1, 1.1, \ldots, 99.1\}$; for booleans, the literals are from {true, false}. The training dataset and evaluation dataset use *disjoint* lists for function names and argument symbols. See Appendix J for an example.

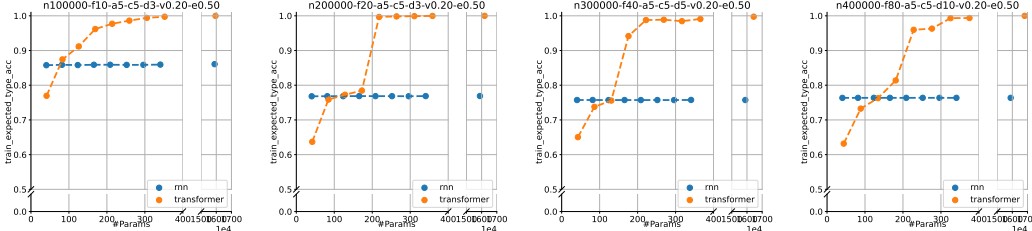

Figure 2: Figures depicting the accuracy of the expected type (see Section 5.3) across different models, measured by their number of trainable parameters. Training accuracies are better indicators of the expressive power of the models (instead of generalizability) than evaluation accuracies. We also report evaluation accuracies in Appendix J.

We use synthetic data instead of real code because our paper is theoretically focused, and the experiments only serve to validate our theory. Hence, using synthetic data gives a clean, controllable setting to compare transformers and RNNs.

**Model and training.** We use customized BERT models (Devlin et al., 2019) and bidirectional RNN models (Schuster & Paliwal, 1997) in our experiments. To control the model size (i.e., the number of trainable parameters), we adjust only the hidden sizes while keeping other hyperparameters constant. Detailed model specifications can be found in Table 1. For both transformers and RNNs, we use the hyperparameters listed in Table 2 in Appendix J.

**Results.** We experimented with multiple combinations of models (Table 1) and datasets (Table 2). For each combination, we conducted independent runs using a fixed set of $k = 5$ random seeds. When plotting the figures, we took the top $t = 5$ training/evaluation losses/accuracies from each run and averaged over all the $k \times t$ values. We plotted separate figures for each dataset and separate sub-figures for each metric. In each sub-figure, the $x$-axis represents the number of trainable parameters, and the $y$-axis represents the averaged values. Results are shown in Figure 2. They demonstrate that customized BERT models are able to perform better at type checking than bidirectional RNN models when both scale up, corroborating our theories. Other results are in Appendix J.

## 7 Conclusions and Future Work

We demonstrated that transformers can efficiently handle a number of syntactic and semantic analysis tasks in C-like languages, using Cybertron to prove their capacity for tasks like AST generation, symbol resolution, and type analysis. We showed theoretical and empirical advantages of transformers over RNNs, particularly in their ability to manage long-range dependencies with logarithmic parameter scaling.

Future research directions include extending our theoretical framework to encompass universal transformers, which may eliminate the current reliance on the assumption of bounded program depth in our analysis Shaw et al. (2024). Another promising avenue is the generalization of our approach from foundational forms of static code analysis to more comprehensive categories of static analysis, potentially incorporating asynchronous constructs and related features. Lastly, a valuable line of inquiry involves formalizing practical aspects of code generation with the goal of rigorously characterizing the expressive power of transformer-based models in such contexts.

## Acknowledgement

XZ acknowledges the support of NSF through awards DMS-2031883 and PHY-2019786. LZ acknowledges the ERC PoC project *FormalWeb3* no. 101156734 and the University of Innsbruck doctoral scholarship *promotion of young talent*. SSD acknowledges the support of NSF DMS 2134106, NSF CCF 2212261, NSF IIS 2143493, NSF IIS 2229881, Sloan Fellowship, the AI2050 program at Schmidt Sciences, Amazon, and Apple.

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

## A   Tree

Trees are one of the most fundamental objects to study in computer science. However, its exact definition differs for different domains. The trees used in "abstract syntax tree" (Section B) is more restrictive than that in mathematics, which we call "typed tree", so that one can define recursive computation more rigorously.

### A.1   What are Trees

Trees in data structures have slightly additional meaning as compared to trees in mathematics. In this paper, all trees are trees in data structures. For clarity, we lay down the precise definition of trees in data structure.

**Definition 3** (Tree). *A tree T is a set of nodes storing elements such that the nodes have a parent-child relationship that satisfies the following:*

- *If T is not empty, it has a special node called the **root** that has no parent.*

- *Each node v of T other than the root has a unique parent node w; each node with parent w is a child of w.*

*We denote the nodes of T as* $\mathrm{N}(T)$.

**Definition 4** (Recursive Definition of a Tree). *A tree T is either empty or consists of a node r (the root) and a possibly empty set of trees whose roots are the children of r.*

However, the above definition is too permissive. We shall define a typed version as follows:

**Definition 5** (Typed Tree). *A tree type consists of a set of values V and a set of relationships* $C \subseteq V \times \mathbb{N}$, *and a typed tree under this type is any tree T such that for each node, a value* $v \in V$ *is assigned such that* $(v, n) \in C$ *where n is the number of the children of the node.*

**All trees in this paper are typed**.

**Example 1** (Abstract syntax tree (AST) as Typed Tree). *Consider an AST for a simple arithmetic expression. Let the set of values V be:*

$$V = \{\ num\ ,\ add\ ,\ sub\ ,\ mul\ ,\ div\ \}$$

*and the set of relationships* $C \subseteq V \times \mathbb{N}$ *specify the allowed number of children for each value:*

$$C = \{(\ num\ ,0), (\ add\ ,2), (\ sub\ ,2), (\ mul\ ,2), (\ div\ ,2)\}$$

*An example AST for the arithmetic expression* $(3 + 5) \times 2$ *is the following typed tree:*

- *The root node is labeled* mul *(multiplication), and it has two children.*

  - *The left child is labeled* add *(addition), and it has two children:*
    - *The left child of* add *is labeled* num *with the value 3.*
    - *The right child of* add *is labeled* num *with the value 5.*
  - *The right child of* mul *is labeled* num *with the value 2.*

*This tree conforms to the typing rules because:*

- num *has 0 children,*

- add *has 2 children,*

- mul *has 2 children,*

*all of which satisfy the relationships in C.*

### A.2 Representations of Trees

It's also important to talk about tree representations. We are studying transformers, and then it's necessary to represent large trees as a sequence, otherwise the model dimension is not large enough to contain the information locally. Let's first talk about the classical **arena pattern** used in system programming for representing trees and we shall slightly adapt it to our use case for studying transformers.

**Arena Pattern.** To represent trees efficiently in memory, especially when trees are frequently modified (such as insertions or deletions of nodes), an arena pattern is often used. The arena pattern provides a way to manage memory allocation for tree structures, allowing for efficient memory usage and avoiding fragmentation. Here's how the arena pattern works in the context of tree representation:

**Definition 6** (Arena Pattern in Tree Representation)**.** *In the arena pattern, a tree is represented by an array (or vector) of nodes, called an* ***arena****. Each node in the arena contains:*

- *An element or value stored in the node.*

- *References (often indices or pointers) to the node's children and possibly to its parent.*

*The key characteristics of the arena pattern are:*

- *Memory Contiguity: All nodes are stored contiguously in memory within the arena, which allows for efficient traversal and modification operations.*

- *Fixed Capacity: The arena has a fixed or dynamically resizable capacity, and nodes are added sequentially. This avoids the overhead of allocating individual nodes on the heap.*

- *Index-based References: Instead of using pointers, the nodes reference each other using indices within the array, which simplifies memory management and can lead to cache-friendly operations.*

- *Efficient Allocation and Deallocation: Nodes can be efficiently allocated and deallocated within the arena without requiring complex memory management techniques like garbage collection or reference counting.*

The arena pattern is particularly useful in scenarios where the structure of the tree is highly dynamic or when performance is critical. It allows for a simple and efficient way to manage and traverse trees without the typical overhead associated with more traditional pointer-based tree representations.

**Adaptations for Transformers** For transformers, inputs, intermediate values and outputs are all sequences. So the trees are represented as sequences of nodes with node reference representable by token position encoding. Based on the representation, transformers will be able to perform various kinds of recursive tree operations, as we shall see.

## B Context Free Grammar

In this section, we lay down the well-known definitions of context free grammar, derivations, and parse trees. To define an abstract syntax tree (AST), one commonly resorts to generation rules, such as context-free grammars (CFG) (Alfred et al., 2007) and parsing expression grammars (PEG) (Ford, 2004). In most cases, just generation rules themselves are not sufficient to define properly a language. Many practical languages like C and C++ cannot be solely described by these rules (David, 2009) so that they can reuse the limited set of special characters on the keyboard. Furthermore, semantic constraints like type correctness are intrinsically contextual and cannot be expressed through CFG or similar rules. However, CFG or other rules provide a valuable construct, the AST. With an AST, one can refine the language definition by putting restrictions on the syntax tree through tree operations. Effectively, a language can be seen as a subset of trees, not as a subset of strings. Semantic

analysis like symbol resolution and type checking can be described effectively based on trees. In short, CFG standalone is hardly practical but it provides a useful and clear foundation to build definitions upon.

A context-free grammar (CFG) is defined as a 4-tuple $G = (V, \Sigma, R, S)$, where:

- $V$ is a finite set of variables (non-terminal symbols).
- $\Sigma$ is a finite set of terminal symbols, disjoint from $V$. Sequences of $\Sigma$, i.e., elements of $\Sigma^*$ are called (literal) strings.
- $R \subset V \times (V \cup \Sigma)^*$ is a finite set of production rules, where each rule is of the form $A \to \alpha$, with $A \in V$ and $\alpha \in (V \cup \Sigma)^*$.
- $S \in V$ is the start symbol.

Given a context-free grammar $G = (V, \Sigma, R, S)$, we define derivation as follows:

- A **derivation** is a sequence of steps where, starting from the start symbol $S$, each step replaces a non-terminal with the right-hand side of a production rule.
- Formally, we write $u \Rightarrow v$ if $u = \alpha A \beta$ and $v = \alpha \gamma \beta$ for some production $A \to \gamma$ in $R$, where $\alpha, \beta \in (V \cup \Sigma)^*$ and $A \in V$.
- A **leftmost derivation** is a derivation in which, at each step, the leftmost non-terminal is replaced.
- A **rightmost derivation** is a derivation in which, at each step, the rightmost non-terminal is replaced.
- We denote a **derivation sequence** as $S \Rightarrow^* w$, where $w \in \Sigma^*$ is a string derived from $S$ in zero or more steps.

A **parse tree** (or **syntax tree**) for a context-free grammar $G = (V, \Sigma, R, S)$ is a tree that satisfies the following conditions:

- The root of the tree is labeled with the start symbol $S$.
- Each leaf of the tree is labeled with a terminal symbol from $\Sigma$ or the empty string $\epsilon$.
- Each internal node of the tree is labeled with a non-terminal symbol from $V$.
- If an internal node is labeled with a non-terminal $A$ and has children labeled with $X_1, X_2, \ldots, X_n$, then there is a production rule $A \to X_1 X_2 \ldots X_n$ in $R$.
- The yield of the parse tree, which is the concatenation of the labels of the leaves (in left-to-right order), forms a string in $\Sigma^*$ that is derived from the start symbol $S$.

## C  Neural Architectures

In this section, we lay down the precise mathematical definitions of neural architectures we are going to use in our proof.

**Definition 7** (Single-Layer Fully Connected Network with $4\times$ Intermediate Space)**.**

*Given model dimension $d_{model}$, a single-layer feed-forward network with an intermediate space expanded to 4 times the input dimension is a function from $\mathbb{R}^{d_{model}}$ to $\mathbb{R}^{d_{model}}$, denoted by $f_{fcn}$ and defined as follows:*

*given $X \in \mathbb{R}^{d_{model}}$, weights $W_1 \in \mathbb{R}^{4d_{model} \times d_{model}}$, $W_2 \in \mathbb{R}^{d_{model} \times 4d_{model}}$, and biases $B_1 \in \mathbb{R}^{4d_{model}}$, $B_2 \in \mathbb{R}^{d_{model}}$, the output $f_{fcn}(X)$ is computed as:*

$$f_{fcn}(X) = W_2 \sigma_{\text{ReLU}}(W_1 X + B_1) + B_2,$$

*where $\sigma_{\text{ReLU}} : \mathbb{R}^{4d_{model}} \to \mathbb{R}^{4d_{model}}$ is the Rectified Linear Unit activation function applied element-wise, defined by:*

$$\sigma_{\text{ReLU}}(z) = \big(\max(z_1, 0), \max(z_2, 0), \ldots, \max(z_{4d_{model}}, 0)\big)^\top,$$

*for $z = (z_1, z_2, \ldots, z_{4d_{model}})^\top \in \mathbb{R}^{4d_{model}}$.*

The choice of a $4\times$ intermediate space is common in practice, often used in Transformer architectures. Interestingly, this empirical choice turns out to have a useful theoretical property: it allows the network to express any affine transformation, as we'll see in the following proposition.

**Proposition 1.** *A single-layer fully connected network with a $4\times$ intermediate space, as defined previously, can express any affine map from $\mathbb{R}^{d_{model}}$ to $\mathbb{R}^{d_{model}}$.*

*Proof.* Let $f : \mathbb{R}^{d_{model}} \to \mathbb{R}^{d_{model}}$ be any affine map given by $f(X) = AX + b$, where $A \in \mathbb{R}^{d_{model} \times d_{model}}$ and $b \in \mathbb{R}^{d_{model}}$. We will construct weights $W_1 \in \mathbb{R}^{4d_{model} \times d_{model}}$, $W_2 \in \mathbb{R}^{d_{model} \times 4d_{model}}$ and biases $B_1 \in \mathbb{R}^{4d_{model}}$, $B_2 \in \mathbb{R}^{d_{model}}$ such that $f_{\text{fcn}}(X) = f(X)$ for all $X \in \mathbb{R}^{d_{model}}$.

Define:

$$W_1 = \begin{pmatrix} I_{d_{model}} \\ -I_{d_{model}} \\ 0 \\ 0 \end{pmatrix}, \quad B_1 = 0 \in \mathbb{R}^{4d_{model}},$$

where $I_{d_{model}}$ is the $d_{model} \times d_{model}$ identity matrix, and $0$ represents zero matrices of appropriate dimensions. Set:

$$W_2 = \begin{pmatrix} A & -A & 0 & 0 \end{pmatrix}, \quad B_2 = b.$$

For any $X \in \mathbb{R}^{d_{model}}$, compute:

$$f_{\text{fcn}}(X) = W_2\, \sigma_{\text{ReLU}}(W_1 X + B_1) + B_2$$

$$= \begin{pmatrix} A & -A & 0 & 0 \end{pmatrix} \sigma_{\text{ReLU}}\left( \begin{pmatrix} X \\ -X \\ 0 \\ 0 \end{pmatrix} \right) + b$$

$$= \begin{pmatrix} A & -A & 0 & 0 \end{pmatrix} \begin{pmatrix} \sigma_{\text{ReLU}}(X) \\ \sigma_{\text{ReLU}}(-X) \\ 0 \\ 0 \end{pmatrix} + b$$

$$= A\, \sigma_{\text{ReLU}}(X) - A\, \sigma_{\text{ReLU}}(-X) + b.$$

Note that $\sigma_{\text{ReLU}}(X) - \sigma_{\text{ReLU}}(-X) = X$, we have:

$$f_{\text{fcn}}(X) = AX + b = f(X).$$

Therefore, the network can represent any affine map from $\mathbb{R}^{d_{model}}$ to $\mathbb{R}^{d_{model}}$. □

**Definition 8** (Single-Layer Feed Forward Network with $4\times$ Intermediate Space). *Given model dimension $d_{model}$ and position set* Pos, *the Transformer Feed Forward Network is a function $f_{ffn} : \mathbb{R}^{\text{Pos} \times d_{model}} \to \mathbb{R}^{\text{Pos} \times d_{model}}$ defined as follows:*

*For an input $X \in \mathbb{R}^{\text{Pos} \times d_{model}}$, the output $f_{ffn}(X)$ is computed by applying the single-layer feed-forward network $f_{fcn}$ (as defined previously) independently to each position:*

$$f_{ffn}(X)_p = f_{fcn}(X_p) \quad \forall p \in \text{Pos}$$

*where $X_p \in \mathbb{R}^{d_{model}}$ is the $p$-th row of $X$, corresponding to the $p$-th position in the input sequence.*

Next, we define the attention mechanism, which is a key component of the Transformer architecture. This definition presents a hard attention layer with a simplified position encoding. We use hard attention here for theoretical simplicity, as it represents a discrete limit of the more commonly used soft attention mechanism. Hard attention forces the model to make a clear choice about which inputs to focus on, which can simplify analysis and provide clearer insights into the model's behavior. It can be viewed as the limiting case of soft attention as the temperature approaches zero, where the softmax operation becomes increasingly peaked and eventually converges to a one-hot vector.

**Definition 9** (Hard Attention Layer with Simplified Position Encoding). *Given model dimension $d_{model}$, number of heads H, and number of layers L, a transformer with simplified position encoding and hard attention is defined to be a function $f_{attn} : \mathbb{R}^{\text{Pos} \times d_{model}} \to \mathbb{R}^{\text{Pos} \times d_{model}}$ defined by*

$$\forall p \in \text{Pos}, f_{attn}(X)_p := W_O \text{Concat} \left( Attn^{(1)}(X)_p, \dots, Attn^{(H)}(X)_p \right), \tag{4}$$

*where the hth attention head uses hard attention, defined as:*

$$Attn^{(h)}(X)_p := \frac{1}{|S_p|} \sum_{p' \in S_p} V_{p'}^{(h)}, \tag{5}$$

*where*

- $W_O \in \mathbb{R}^{d_{model} \times d_{model}}$ *are trainable parameters;*

- $S_p = \arg\max_{p' \in \text{Pos}} \left( {Q_p^{(h)}}^\top K_{p'}^{(h)} + \lambda^{(h)\top} \Psi_{p'-p} \right)$ *with $Q_p^{(h)}, K_p^{(h)}, V_p^{(h)}, \lambda^{(h)}, \Psi_q$ defined by*

  - $Q_p^{(h)} = W_Q^{(h)} X_p, K_p^{(h)} = W_K^{(h)} X_p$ *are vectors of dimension $d_{model}/H$, with trainable parameters $W_Q^{(h)}, W_K^{(h)} \in \mathbb{R}^{(d_{model}/H) \times d_{model}}$;*

  - $V_p^{(h)} = W_V^{(h)} X_p$ *are vectors of dimension $d_{model}/H$, linear transformations of $X_p$ with trainable parameters $W_V^{(h)} \in \mathbb{R}^{(d_{model}/H) \times d_{model}}$;*

  - $\lambda^{(h)} \in \mathbb{R}^2$ *are constants depending only on head count h;*

  - $\Psi_q \in \mathbb{R}^2$ *are 2-dimensional vectors depending on relative position q but not on head count h. It is explicitly defined as*

$$\Psi_q = \begin{pmatrix} q \\ 1_{q>0} \end{pmatrix}. \tag{6}$$

  *This formulation allows for both past and future masking.*

Having defined the basic components, we can now proceed to describe the full Transformer architecture. This definition builds upon the previously introduced concepts, incorporating them into a complete model structure.

**Definition 10** (Transformer). *A **Transformer** is a function $f_{tf} : \mathbb{R}^{\text{Pos} \times d_{model}} \to \mathbb{R}^{\text{Pos} \times d_{model}}$ that maps an input sequence to an output sequence through a series of layers, each consisting of a multi-head attention mechanism and a position-wise feed-forward network (MLP).*

*Given:*

- *Input sequence $X \in \mathbb{R}^{\text{Pos} \times d_{model}}$, where $\text{Pos}$ is the set of positions and $d_{model}$ is the model dimension.*

- *Number of layers L.*

- *Number of attention heads H.*

*The Transformer computes the output $Y = X^{(L)}$ through recursive application of attention and feed-forward layers:*

- *Initialization is given by:*

$$X^{(0)} = X.$$

- *For each layer $l = 1, 2, \ldots, L$:*

    - *Compute attention output:*

    $$\hat{X}^{(l)} = X^{(l-1)} + f_{attn}^{(l)} \left( X^{(l-1)} \right)$$

    - *Compute feed-forward output:*

    $$X^{(l)} = \hat{X}^{(l)} + f_{ffn}^{(l)} \left( \hat{X}^{(l)} \right)$$

*Here:*

- $f_{attn}^{(l)}$ *are **hard attention layers with simplified position encoding** as previously defined. It operates on $X^{(l-1)}$ and produces an output in $\mathbb{R}^{\text{Pos} \times d_{model}}$.*

- $f_{ffn}^{(l)}$ *are **feed-forward networks with $4\times$ intermediate space** as previously defined. It operates position-wise on $\hat{X}^{(l)}$ and produces an output in $\mathbb{R}^{\text{Pos} \times d_{model}}$.*

*Remark* 1. For simplicity, we have omitted the Layer Normalization component typically present in Transformer architectures. This simplification allows us to focus on the core attention and feed-forward mechanisms while maintaining the essential structure of the Transformer.

We use $\text{Tf}_{H,L}^{d_{\text{model}}}$ to denote the set of transformers of model size $d_{\text{model}}$, number of heads $H$ and number of layers $L$ as functions from $\mathbb{R}^{d_{\text{model}}*}$ to $\mathbb{R}^{d_{\text{model}}*}$.

For purpose of proof, we shall also need residual multi-layer perceptron. Functions over local types are first represented by multi-layer perception, then by Proposition 2 applications of these functions over sequences can be representable by transformers. Residual multi-layer perceptron can be assembled through composition or computer graph, as we shall see.

Here's the definition of a residual MLP Network.

**Definition 11** (Residual Multi-Layer Perceptron). *A Residual Multi-Layer Perceptron (ResMLP) is a function $f_{resmlp} : \mathbb{R}^{d_{model}} \to \mathbb{R}^{d_{model}}$ defined recursively by*

$$X^{(0)} = X, \quad X^{(l)} = X^{(l-1)} + f_{fcn} \left( X^{(l-1)} \right), \quad l = 1, 2, \ldots, L, f_{resmlp}(X) = X^{(L)}$$

*where $X \in \mathbb{R}^{d_{model}}$ is the input vector, $L$ is the total number of layers, and $f_{fcn} : \mathbb{R}^{d_{model}} \to \mathbb{R}^{d_{model}}$ is the* Single-Layer Fully Connected Network with $4\times$ Intermediate Space *as previously defined in Definition 7.*

We use $\text{ResMlp}_L^{d_{\text{model}}} \subset \mathbb{R}^{d_{\text{model}}^{\mathbb{R}^{d_{\text{model}}}}}$ to represent the set of residual MLPs with dimension $d_{\text{model}}$ and $L$ layers, as defined in Definition 11.

The following proposition is quite basic. It demonstrates that any function representable by a ResMLP can be applied position-wise by a Transformer.

**Proposition 2** (Position-wise ResMLP Application is Representable by Transformers). *Let $f : \mathbb{R}^{d_{model}} \to \mathbb{R}^{d_{model}}$ be a function representable by a Residual Multi-Layer Perceptron (ResMLP) as defined in Definition 11. Then the function $F : \mathbb{R}^{\text{Pos} \times d_{model}} \to \mathbb{R}^{\text{Pos} \times d_{model}}$, defined by applying $f$ position-wise,*

$$F(X)_p = f(X_p), \quad \forall p \in \text{Pos},$$

*is representable by a Transformer as defined in Definition 10.*

*Proof.* Since $f$ is representable by a ResMLP with $L$ layers, it is defined recursively by

$$X^{(0)} = X, \quad X^{(l)} = X^{(l-1)} + f_{\text{fcn}}\left(X^{(l-1)}\right) \text{ for } l = 1, \ldots, L,$$

and

$$f(X) = X^{(L)},$$

where $f_{\text{fcn}} : \mathbb{R}^{d_{\text{model}}} \to \mathbb{R}^{d_{\text{model}}}$ is the Single-Layer Fully Connected Network with $4\times$ intermediate space (Definition 7).

We construct a Transformer with $L$ layers such that, for any input sequence $X \in \mathbb{R}^{\text{Pos} \times d_{\text{model}}}$, the output $Y = f_{\text{tf}}(X)$ satisfies $Y_p = f(X_p)$ for all $p \in \text{Pos}$.

To achieve this, we configure the Transformer so that the attention mechanism outputs zero at each layer. This can be done by setting the attention weights to zero, ensuring $f_{\text{attn}}(X^{(l-1)}) = 0$. Consequently, the update equations simplify to

$$\hat{X}^{(l)} = X^{(l-1)}.$$

We then set the feed-forward network $f_{\text{ffn}}$ in the Transformer to have the same weights and biases as $f_{\text{fcn}}$ in the ResMLP. The Transformer layer update becomes

$$X^{(l)} = \hat{X}^{(l)} + f_{\text{ffn}}\left(\hat{X}^{(l)}\right) = X^{(l-1)} + \left(f_{\text{fcn}}\left(X_p^{(l-1)}\right)\right)_{p \in \text{Pos}}.$$

This recursion matches that of the ResMLP applied position-wise to $X$. Therefore, after $L$ layers, the Transformer output satisfies $f_{\text{tf}}(X)_p = f(X_p)$ for all $p \in \text{Pos}$.

$\square$

# D    Cybertron

## D.1    Introduction

It's often difficult to directly prove that transformers or in general other low level forms of computation can express complicated algorithms and even complex software. There are way too many details as compared with typical mathematical proofs in machine learning theory. Hence, we propose the domain specific language Cybertron, where we can systematically prove transformers can express complicated algorithms and complex software with sufficient readability.

(Note: Cybertron is fundamentally different from Mini-Husky! Mini-Husky is the target language that we want transformers to analyze yet Cybertron is the domain specific language we use to prove that transformers can do that.)

RASP (Weiss et al., 2021) is quite close to Cybertron in terms of its design purpose. However, Cybertron is more powerful with advanced algebraic type system, global and local function constructions, etc. These additional mechanisms replace a significant part of the chore in proofs with automatic type checking. Thus, using Cybertron one can argue operations more complicated than simple algorithms can be simulated by transformers.

In the broader perspective of computer science, it's common to use code to prove things. In fact, in the formal verification community, mathematical proofs are viewed as a special case of a much larger universe of possible proof systems (mathlib Community, 2019; Massot, 2024) and constructive proof using code (Harrison et al., 2014; Farooqui, 2021; Jung et al., 2018) is far more applicable with great soundness to the most general settings. In our case, our code doesn't serve as the whole proof but as an important part that contains most of the chores. However, it's totally possible to build a full-fledged formalized proof, despite it might be too costly for a single paper to do.

Essentially, Cybertron works as follows:

1. one builds complicated functions from the composition of smaller functions. We have lemmas that prove that the composed functions are representable by certain architectures given that smaller functions are representable.

2. there is an algebraic type system and every value is strongly typed and immutable, making it highly readable and easy to reason about;

3. there is a distinction between global and local types/functions. Local types are those information hovered over a single token, and global types are sequences of local types, i.e., the collection of information over the whole token stream. One can define a global function by mapping a local function.

4. there are many functions that is defined externally, requiring external explanation that they can be represented by transformers.

It's implemented as a subset of the Rust programming language that can be understood as computation graphs over sequences. It can be executed for testing purposes and we've tested our implementation for a range of inputs and validated its correctness.

### D.2 Philosophy: Sequential Representation of Everything

Before going through the full details, let's first talk about the fundamental philosophy behind transformer and Cybertron.

One of the fundamental reasons transformers can be easily adapted across multiple modalities, including NLP and CV, is their sequence-to-sequence operation. Everything can be represented as an arbitrary-length sequence. Texts are sequences of words, images are sequences of image patches, videos are sequences of spacetime patches (OpenAI, 2024b), and even graphs with sparse spatial structures can be represented as sequences of indexed nodes with additional information like parent node indices. Since inputs of various modalities can be cast into vector sequences, transformers can be applied to different domains without modifications to their architecture (Dosovitskiy et al., 2020).

Interestingly, this sequence-based thinking is not new. **We've actually been representing everything as sequences since the very early days of computer science.** This has been the foundation of how data is stored and processed in computers. However, sequence representations were traditionally viewed as low-level and sometimes inefficient for practical use, prompting the development of higher-level abstractions for programming. The rise of transformers, with their scalable learning capabilities, encourages us to reconsider the significance of sequence-based representations.

From a systems perspective, viewing everything as a sequence is the foundational approach in computer science. Data in a computer is stored as a continuous stream of bits. Whether it's text, images, videos, or graphs, this data is represented in computer memory as an ordered sequence of bits. This aligns with how transformers handle different types of input by transforming them into sequences of vectors. Thus, the sequence-based operation of transformers mirrors the sequence-based representation of data in computer memory.

In essence, **if a data structure can be represented in computer memory using $N$ bits, it can be processed as a sequence of bits of length** $N$. This natural sequence representation in memory is consistent with how transformers process data, which makes them particularly flexible across different modalities. For example, recent state-of-the-art approaches Wu et al. (2024) show that transformers can even be trained directly on raw bits of data, further emphasizing this connection.

Moreover, this sequence-based viewpoint offers fresh insights when applied to the domain of programming, particularly in areas such as code generation and analysis. With tools like ChatGPT and Copilot being widely used by developers, the impact of transformers on programming workflows is growing. Understanding the complexity of algorithms and programs expressed in sequence form becomes an interesting area of study, as it reveals new possibilities for how we approach computation.

In comparison to traditional systems like CPUs and human cognition, transformers are highly parallel but shallow in their operation. A transformer processes data in a fixed number of layers, while a CPU executes $10^9$ cycles per second, and humans may take days to process information like reading a book. Transformers, therefore, represent a

fundamentally different computational model that is worth studying further in the context of sequence-based operations.

**Example 2.** *Image to Sequence: In computer memory, an image is typically stored as a continuous block of pixel values, often in row-major order, where each pixel's value is encoded as bits in a sequence. When a transformer processes an image, it divides the image into patches (e.g., $16 \times 16$ pixels), and each patch is flattened into a vector of pixel values. This creates a sequence of patches, where each patch corresponds to a vector. The way transformers represent these patches as a sequence closely aligns with how the image data is sequentially stored in computer memory.*

**Example 3.** *Video to Sequence: A video is stored in computer memory as a sequence of frames, where each frame is essentially an image. In a similar manner to images, these frames are stored as continuous pixel values. Transformers process videos by dividing the frames into spacetime patches, where each patch captures a small region of space over a short segment of time. These spacetime patches are flattened and arranged into a sequence for the transformer to process. The sequential ordering of these patches matches how video frames and pixel data are stored in computer memory.*

**Example 4.** *Graph to Sequence: In computer memory, a graph is typically stored using an adjacency list or adjacency matrix, where nodes and their connections (edges) are stored sequentially in a data structure. Transformers process graphs by representing each node and its features as a vector, and then creating a sequence of these vectors. The sequence may also encode additional information, such as the parent-child relationships between nodes. This sequence-based representation of graphs is consistent with how graph data is stored in memory, where nodes and edges are arranged in a structured order.*

**Example 5.** *Text to Sequence: Text is naturally stored in computer memory as a sequence of characters or words, where each character is encoded as a sequence of bits (such as ASCII or Unicode values). When transformers process text, they convert each word into a word embedding, which is a vector of real numbers. The sequence of word embeddings corresponds to the sequence of characters or words stored in memory. This natural sequential representation of text in both memory and transformers ensures efficient handling of linguistic data.*

**Example 6.** *AST (Abstract Syntax Tree) to Sequence: In computer memory, an abstract syntax tree (AST) is typically stored as a tree-like structure, where each node represents a component of the program (e.g., operators, variables, or statements). However, this tree can be linearized into a sequence by traversing it in a specific order (e.g., pre-order traversal). When transformers process an AST, they convert it into a sequence of tokens, where each token corresponds to a node in the tree. This sequential representation of the tree in transformers mirrors how the tree is stored as nodes and edges in memory, and how it can be flattened into a linear sequence.*

In conclusion, the sequence-based representation in transformers is not just a novel approach for deep learning but is deeply rooted in how data has been stored and processed in computer memory since the early days of computing. This consistency between how data is stored in memory and how transformers process data as sequences is a key factor in their adaptability across different domains.

### D.3  Local and Global Types

Now we define the type foundation of Cybertron.

Types are fundamental objects for programming language theory. Here we use types to faciliate our proofs. Type signatures contain rich information that help guarantee correctness of the program. Here, we choose a mathematical definition of types that is most convenient for the discussion in this paper. We introduce the notion of "local type". Roughly speaking, they are types without heap allocation and intended to be represented with $\mathbb{R}^{d_{\text{model}}}$ over a single token. For more complicated heap-allocated data structures like trees, graphs, etc., we shall represent them by sequences of these "local type"s, which translate directly to vector sequences for transformers.

**Definition 12** (Local Type). *Given a base space $B$ with at least two elements and a countably infinite identification space $\Psi$, a local type $\mathcal{T}$ over $B$ is a finite set $S$ together with an embedding $\phi$ from $S$ to $B^d$ and some fixed $d \in \mathbb{N}$ and an identification $\psi \in \Psi$.*

*For convenience, define Set* $(\mathcal{T}) = S$, $d_{\mathcal{T}} = d$ *and* $\phi_{\mathcal{T}} = \phi$ *and* $\psi_{\mathcal{T}} = \psi$. *And let* $0_B$, *and* $1_B$ *be two different elements of B. And* $B^0 := \{0_B\}$ *so that* $|B^i| = |B|^i$ *holds for all* $i \in \mathbb{N}$.

*Remark* 2. We need $B$ to be at least size 2, so that $B^d$ can be as large as we want for $d$ large enough. For typical computer representation, we can take $B$ to be $\mathbf{2} = \{0, 1\}$. For transformers or neural networks in general, we can take $B$ to be $\mathbb{R}$ if we ignore precision. If we don't ignore precision, $B$ should be some finite set of floating point numbers. Thus, we shall keep the generality of $B$ throughout our discussion as all of these settings are important.

*Remark* 3. The role of identification $\psi_{\mathcal{T}} \in \Psi$ is to make two types mathematically different even if they have the same underlying set, encoding dimension, and encoding. Basically we are establishing a specialized type of theory tailored towards the expressive power of transformers upon a foundation of intuitive set theory.

**Example 7** (Finite Set). *In mathematics, we have the finite set denoted by* $[n] = \{0, 1, \ldots, n-1\}$. *Here we use a slightly different notation for a type with underlying set* $[\![n]\!]$ *and some encoding.*

**Example 8** (Position Encoding). *Position encoding can be viewed as the encoding of a type denoted by Pos $(n)$ with the underlying set being* $[n]$ *where n is the context length. Although it has the same underlying set as type* $[\![n]\!]$, *it is a different type for a different purpose and might have different encoding.*

*If B is* $\mathbb{R}$, *then the position encoding can be understood as the encoding of type* $[\![L]\!]$ *where L is the context length. More explicitly, we have*

$$\phi(x) = (e^{\mathring{\imath}L^{-i/d}x})_{i \in [d/2]}, \tag{7}$$

*viewed as a d dimensional* $\mathbb{R}$*-vector through the natural conversion of* $\mathbb{C}$ *to* $\mathbb{R}^2$, *since d is even.*

It's too cumbersome to manually give the underlying set and the encoding. Here we introduce a classical concept from programming language theory Program (2013) that makes it super easy to construct new types and make things fairly readable.

**Definition 13** (Finite Algebraic Data Type, Mathematical Forms). *We define two ways of creating new types by combining existing types:*

1. *Sum type. Given types* $\mathcal{T}_i = (S_i, \phi_i, d_i)$ *over base space B for* $i = 1, \ldots, n$, *we define the sum type of* $\mathcal{T}_i$, *denoted by* $\sum_{i=1}^n \mathcal{T}_i$, *as follows,*

   - *let* $S = (\{1\} \times S_1) \sqcup \ldots \sqcup (\{n\} \times S_n)$;
   - *let* $d = d_{[\![n]\!]} + \max_{i=1}^n d_i$;
   - *let* $\phi : S \to B^d$ *be such that*

   $$\forall i \in [\![n]\!], s \in S_i, \phi((i, s)) = \phi_{[\![n]\!]}(i) \oplus \phi_i(s) \in B^{d_{[\![n]\!]} + d_i} \subseteq B^d. \tag{8}$$

   *Note that* $|S| = \sum_{i=1}^n |S_i|$, *thus the name sum type.*

2. *Product type. Given Local Types* $\mathcal{T}_i = (S_i, \phi_i, d_i)$ *over base space B for* $i = 1, \ldots, n$, *we define the product type of* $\mathcal{T}_i$, *denoted by* $\prod_{i=1}^n \mathcal{T}_i$, *as follows,*

   - *let* $S = S_1 \times \ldots \times S_n$;
   - *let* $d = \sum_{i=1}^n d_i$;
   - *let* $\phi : S \to B^d$ *be such that*

   $$\forall s = (s_1, \ldots, s_n) \in S, \phi(s) = \phi_1(s_1) \oplus \ldots \phi_n(s_n) \in B^d. \tag{9}$$

   *Note that* $|S| = \prod_{i=1}^n |S_i|$, *thus the name product type.*

Although we can define things and refer to things in terms of mathematical equations, it's sometimes cumbersome to do so. So we shall frequently refer to types using a programming language form, like `CybertronForm` or more complicated things like `Option<T>` a builtin generic type.

**Definition 14** (Unit Type). *The unit type is a type with $S = \{0\}$ and $\phi : S \to B^0, 0 \mapsto 0_B$. In Cybertron, it's denoted as* `()` *.*

**Definition 15** (Array Type). *Given a type $\mathcal{T}$, the array type of $\mathcal{T}$ with length $\ell \in \mathbb{N}$ is the type with $S = S(\mathcal{T})^\ell, d = \ell d_\mathcal{T}$ and $\phi : S \to B^{\ell d_\mathcal{T}}, (s_1, \ldots, s_\ell) \mapsto \phi_\mathcal{T}(s_1) \oplus \ldots \oplus \phi_\mathcal{T}(s_\ell)$. It's denoted by $\mathcal{T}^\ell$. In Cybertron, it's denoted as* `[T;N]` *.*

**Definition 16** (Vector Type of Finite Capacity). *Given a type $\mathcal{T}$, the vector type of finite capacity of $\mathcal{T}$ with maximal length $\ell \in \mathbb{N}$ is the type with $S = \bigsqcup_{i=1}^\ell Set(\mathcal{T})^i, d = d_{[\![\ell]\!]} + \ell d_\mathcal{T}$ and $\phi : S \to B^{d_{[\![\ell]\!]} + \ell d_\mathcal{T}}, (s_1, \ldots, s_i) \mapsto \phi_{[\![\ell]\!]}(i) \oplus \phi_\mathcal{T}(s_1) \oplus \ldots \oplus \phi_\mathcal{T}(s_i) \oplus 0_B \oplus \ldots \oplus 0_B$ with just enough number of copies of $0_B$ such that the dimensionality matches. It's denoted by $\mathcal{T}^{\leq \ell}$. In cybertron, it's denoted as* `BoundedVec<T,N>` *.*

However, it's cumbersome and obtuse to define and operate in mathematical forms only. So we shall give a definition closer to actual programming that is more convenient and easy to read.

**Definition 17** (Finite Algebraic Data Type, the Code Forms). *We define two ways to create new types:*

1. *Enum type. An enum type is the sum type of a finite set of variant types. Each variant type is associated with a different identifier and can be*

   - *unit like, a unit type;*
   - *struct like, a product of several types, each called a field of the variant, and associated with an identifier;*
   - *tuple like, a product of several types, each called a field of the variant, but not associated with an identifier.*

   *Syntactically, an enum type is specified as follows,*

```
enum <type-name> {
    <identifier> {              // 1st variant, struct like
        <identifier>: <type>, // 1st named field of 1st variant
        <identifier>: <type>, // 2nd named field of 1st variant
        ...
    },
    <identifier> {              // 2nd variant, struct like
        <identifier>: <type>, // 1st field of 2nd variant
        ...
    },
    <identifier> (              // 3rd variant, tuple like
        <type>,                 // 1st tuple field of 3rd variant
        <type>,                 // 2nd tuple field of 3rd variant
        ...
    ),
    <identifier>,               // 4th variant, unit like
    ...
}
```

   *For example,*

```
enum Expr {
    Variable(IdentToken),   // 1st variant, tuple like
    Binary {                // 2nd variant, struct like
        lopd: ExprId,
        opr: BinaryOprToken,
        ropd: ExprId,
    },
    Prefix {                // 3rd variant, struct like
        opr: PrefixOprToken,
        opd: ExprId,
    },
    Suffix {                // 4th variant, struct like
        opd: ExprId,
        opr: SuffixOprToken,
    },
    Panic,                  // 5th variant, unit like
}
```

2. *Struct type. A struct type is just the product type of*

```
1  struct <type-name> {
2      <identifier>: <type>,
3      <identifier>: <type>,
4      ...
5  }
```

```
1  struct A {
2      a: i32
3  }
```

To show how convenient this is, we can define the very useful option type as follows,

**Definition 18** (Option type). *For a local type* `T` *, we can define an option type as*

```
1  enum Option<T> {
2      Some(T),
3      None
4  }
```

**Definition 19** (Global Types). *Global types are defined to be sequences of local types.*

**Example 9** (Representation of Graphs). *Graphs can be represented as sequences of its nodes. We can use position index to use as node references.*

### D.4 Computation Graph

For convenience, we shall use computation graph as a vehicle to describe complicated computation processes. Computation graph is close to actual computation process and one can derive an understanding of the computation difficulty from the graph's mathematic properties (width, depth, etc.)

**Definition 20** (Directed Simple Graph). *A directed simple graph $G$ is a pair $(V, E)$ where $V$ is a finite set, and $E \subseteq V \times V$ is called edges.*

In the following, we shall simplify the "directed simple graph" to just graph.

**Definition 21** (Computation Graph). *A computation graph is an acyclic directed graph $G = (V, E)$ with additional structures:*

1. *for each vertex $v \in V$, there is an associated type, denoted by $T_v$;*

2. *for each vertex $v \in V$ with a positive number of incoming edges, let $v_1, \ldots, v_n$ be the other vertices for the incoming edges, then there is an associated function $f_v$ from $T_{v_1} \times \cdots \times T_{v_n}$ to $T_v$.*

A computation graph naturally generates a function from **source vertices** to **sink vertices**. Let $v_1^{\text{in}}, \ldots, v_n^{\text{in}}$ be the set of vertices with no incoming edges, and let $v_1^{\text{out}}, \ldots, v_m^{\text{out}}$ be the set of vertices with no outgoing edges. Then we can construct a function from $T_{v_1^{\text{in}}} \times \cdots \times T_{v_n^{\text{in}}}$ to $T_{v_1^{\text{out}}} \times \cdots \times T_{v_m^{\text{out}}}$ in the following obvious manner:

1. let $(x_1, \ldots, x_n) \in T_{v_1^{\text{in}}} \times \cdots \times T_{v_n^{\text{in}}}$ be an input;

2. for each $v_i^{\text{in}}$, assign it with value $x_i$;

3. for each vertex $v \in V$ with all its incoming vertices $v_1, \ldots, v_l$ assigned with a value, assign it with the value $f_v(x_{v_1}, \ldots, x_{v_l})$ where $x_{v_i}$ denotes the value assigned to $v_i$;

4. repeat the process until all vertices are assigned a value, then take $(x_{v_1^{\text{out}}}, \ldots, x_{v_m^{\text{out}}})$ as the output.

Our goal is to make a hypothesis class using the above graph. To control the statistical and computational complexity, we put restrictions on the choice of $T_v$ and $f_v$, as follows:

**Definition 22** (Restricted Computation Graph). *Let $\mathcal{U}$ be a set of types, and for any $A, B \in \mathcal{U}$, there is a set of functions $\mathrm{Mor}(A, B)$ from $A$ to $B$. We require that $T_v, T_v^{in} \in \mathcal{U}$ and $f_v \in \mathrm{Mor}(T_v^{in}, T_v)$ where $T_v^{in} := \prod_{v'v \in E} T_{v'}$. We also require that the underlying graph $G$ satisfies certain conditions (width, depth, etc.)*

**Definition 23** (Restricted Computation Graph Of Sequences). *Let $\mathcal{U}$ be a universe such that for a set of types $\mathcal{U}_0$ all types in $\mathcal{U}$ are of the form $A^*$ for some type $A \in \mathcal{U}_0$, and $\mathrm{Mor}(A^*, B^*)$ are functions that preserve sequence lengths.*

Given a restriction, the class of functions generated by restricted computation graphs is the central object to study in this paper. We shall use an even more restricted computation graph of sequences. We shall argue about the class of functions formed that

1. it's rich enough to contain many interesting operations including SQL, compiler (type inference, static analysis)
2. it's computationally reasonable, and can be represented by transformers with pragmatic bounds
3. it has a reasonable statistical complexity

As a corollary, our theories suggest that transformers can possibly learn to do many interesting things with reasonable computational and statistical complexity.

To our knowledge, this is the first theoretical paper that gives pragmatic optimistic bounds for the power of transformers in a wide range of meaningful language tasks.

Now we introduce graph-theoretical measures that will play key roles in our new complexity theory.

The most basic one is the following:

**Definition 24** (Depth of Graph). *The depth of a computation graph is defined to the length of the longest path, denoted by $d_G$.*

For convenience, we define the following vertex-wise depth.

**Definition 25** (Depth of Graph Vertex). *The depth of a vertex $v$ of a computation graph is defined as the length of the longest path with end $v$, denoted by $d_v$.*

The smaller $d_G$ is, the more parallel the computation is.

However, we shall discuss a more nuanced measure, containment, as follows:

### D.5 Functions over Local Types

**Definition 26** (Functions over Local Types). *Given Local Types $\mathcal{T}, \mathcal{R}$, the functions from $\mathcal{T}$ to $\mathcal{R}$ are defined to be just the functions from $\mathrm{Set}\,(\mathcal{T})$ to $\mathrm{Set}\,(\mathcal{R})$.*

*Remark* 4. The domains and codomains are all finite sets, so there aren't many constraints we want to enforce here. Basically, these are "discrete" functions.

**Definition 27** (Functions over Algebraic Data Types). *Let $\mathcal{T}, \mathcal{S}_1, \ldots, \mathcal{S}_m, \mathcal{R}$ be Local Types, and suppose that $\mathcal{T}$ is an algebraic data type, then we can construct functions from $\mathcal{T} \times \mathcal{S}_1 \times \ldots \times \mathcal{S}_m$ to $\mathcal{R}$ as follows,*

1. *suppose that $\mathcal{T}$ is the sum type of $\mathcal{T}_1, \ldots, \mathcal{T}_n$. Then given functions $f_i : \mathcal{T}_i \times \mathcal{S}_1 \times \cdots \times \mathcal{S}_m$ for $i = 1, \ldots, n$, we can construct a function $f$, by letting*

$$f((i, t), s_1, \ldots, s_m) = f_i(t, s_1, \ldots, s_m), \qquad (10)$$

   *for each $t \in \mathrm{Set}\,(T_i), s_1 \in \mathrm{Set}\,(\mathcal{S}_1), \ldots, s_m \in \mathrm{Set}\,(\mathcal{S}_m)$.*

   *(Note that we use the pair $(i, t)$ because the underlying set of $\mathcal{T}$ is $\bigsqcup_{i=1}^n \{i\} \times \mathrm{Set}\,(\mathcal{T}_i)$.)*

2. *suppose that $\mathcal{T}$ is the product type of $\mathcal{T}_1, \ldots, \mathcal{T}_n$. Then given a function $f_* : \mathcal{T}_1 \times \cdots \times \mathcal{T}_n \times \mathcal{S}_1 \times \cdots \times \mathcal{S}_m$ for $i = 1, \ldots, n$, we can construct a function $f$, by letting*

$$f((t_1, \ldots, t_n), s_1, \ldots, s_m) = f_*(t_1, \ldots, t_n, s_1, \ldots, s_m), \qquad (11)$$

for each $t \in \text{Set}(T_i), s_1 \in \text{Set}(\mathcal{S}_1), \ldots, s_m \in \text{Set}(\mathcal{S}_m)$.

It is not enough to just mathematically construct. We should also discuss how neural networks can represent these functions. We define the representation of functions over Local Types formally as follows:

**Definition 28** (Representation of Functions over Local Types Using Multi-Layer Perceptions). *Let $\mathcal{T}, \mathcal{R}$ be Local Types. Given a function $f$ from $\mathcal{T}$ to $\mathcal{R}$, we say it is representable by MLP of dimension $d \geq \max\{d_\mathcal{T}, d_\mathcal{R}\}$ and number of layers L, if there exists $\tilde{f} \in \text{ResMlp}_L^d$ such that*

$$\iota_1 \circ \phi_\mathcal{R} \circ f = \tilde{f} \circ \iota_2 \circ \phi_\mathcal{T}, \tag{12}$$

*where $\iota_1 : \mathbb{R}^{d_\mathcal{R}} \to \mathbb{R}^d$ and $\iota_2 : \mathbb{R}^{d_\mathcal{T}} \to \mathbb{R}^d$ are the canonical embeddings by putting zeros to fit the dimensionalities.*

Here are some trivially true facts:

**Proposition 3.** *[Identities are Representable] For any Local Type $\mathcal{T}$, the identity map $\text{Id}_\mathcal{T}$ is representable in $\text{ResMlp}_1^{d_\mathcal{T}}$.*

*Proof.* Just take $W_0^{(1)} = I_d, W_1^{(1)} = W_2^{(2)} = 0, B_1^{(1)} = B_2^{(2)} = 0$. $\qquad\square$

**Proposition 4.** *[Equality is Representable] The equality function for any local type $\mathcal{T}$ is representable in $\text{ResMlp}_2^{2d}$, where $d$ is the encoding dimension of $\mathcal{T}$.*

*Proof.* Let $x, y \in \mathcal{T}$ be the inputs. We encode them as $\phi_\mathcal{T}(x), \phi_\mathcal{T}(y) \in \mathbb{R}^d$. The equality function can be represented as:

$$f_{\text{eq}}(x, y) = \min\left(1, A \sum_{i=1}^d |\phi_\mathcal{T}(x)_i - \phi_\mathcal{T}(y)_i|\right),$$

where $A$ is a large enough positive constant such that the RHS is either 1 or 0.

This can be implemented in two-layer ResMLP with dimension $2d$. $\qquad\square$

**Proposition 5.** *[Boolean NOT is Representable] The Boolean NOT function is representable in $\text{ResMlp}_1^1$.*

*Proof.* It's affine. $\qquad\square$

**Proposition 6.** *[Boolean AND is Representable] The Boolean AND function is representable in $\text{ResMlp}_1^2$.*

*Proof.* Represent each Boolean value as a binary flag within a 1-dimensional vector. Then AND is just taking the minimum. By $\min(a, b) = b - \sigma_{\text{ReLU}}(b - a)$, we're done. $\qquad\square$

**Proposition 7.** *[Boolean OR is Representable] The Boolean OR function is representable in $\text{ResMlp}_1^2$.*

*Proof.* Represent each Boolean value as a binary flag within a 1-dimensional vector. Then OR is just taking the maximum. By $\max(a, b) = a + \sigma_{\text{ReLU}}(b - a)$, we're done. $\qquad\square$

**Proposition 8.** *[THEN_SOME is Representable] The function* `Bool::then_some` *:* `Bool` $\times$ `T` $\to$ `Option T` *returns* `Some t` *if the boolean is true and* `None` *otherwise. This function is representable in $\text{ResMlp}_1^{d+1}$.*

*Proof.* Encode the boolean as a binary flag in a $(d+1)$-dimensional vector, where the first component indicates the boolean value and the remaining $d$ components hold the value of type `T` . The residual MLP $f_{\text{resmlp}}$ constructs the output `Option T` by assembling the flag and the value split into positive and negative parts influenced by the flag:

$$f_{\text{resmlp}}(X) = \begin{pmatrix} x_1 \\ \sigma_{\text{ReLU}}\left(x_{2:d+1} - Ax_1\right) - \sigma_{\text{ReLU}}\left(-x_{2:d+1} - Ax_1\right) \end{pmatrix}.$$

Here, $A$ is a vector of dimension $d$ with all entries positive and large enough to ensure proper thresholding. Specifically, each entry of $A$ should be larger than the maximum absolute value that can be represented in the corresponding dimension of type `T` . This ensures that when $x_1 = 1$, the subtraction $x_{2:d+1} - A$ will always be negative, and when $x_1 = 0$, it will not affect the value.

When the flag is true ($x_1 = 1$), $\sigma_{\text{ReLU}}\left(x_{2:d+1} - A\right) = 0$ and $\sigma_{\text{ReLU}}\left(-x_{2:d+1} - A\right)$ retains the negated value, resulting in `Some t` . When the flag is false ($x_1 = 0$), both ReLU terms preserve the value, yielding `None` . Thus, $f_{\text{resmlp}}$ effectively implements `Bool::then_some` within a single layer of the MLP. $\square$

**Proposition 9.** *[Option Or is Representable] Let* `T` *be a local type, let* `Option::or` *be the function that maps two values* `a,b` *of type* `Option T` *to a value* `c` *of type* `Option T` *such that* `c` *is equal to* `a` *when* `a` *is not none, and equal to* `b` *otherwise. Then* `Option::or` *is representable in* $\text{ResMlp}_1^{2(d+1)}$.

*Proof.* Each `Option T` is represented as a $(d+1)$-dimensional vector, where the first component is a binary flag indicating the presence (1 for `Some` , 0 for `None` ), and the remaining $d$ components encode the value. Given inputs $a, b \in$ `Option T` , the residual MLP $f_{\text{resmlp}}$ processes the concatenated vector

$$X = \begin{pmatrix} a_{\text{flag}} \\ a_{\text{val}} \\ b_{\text{flag}} \\ b_{\text{val}} \end{pmatrix}.$$

The MLP is designed to separate $b_{\text{val}}$ into positive and negative parts ($b_+, b_-$ respectively) influenced by $a_{\text{flag}}$. Specifically, it computes:

$$\begin{aligned} f_{\text{resmlp}}(X) &= a_{\text{val}} + \sigma_{\text{ReLU}}\left(b_+ - Aa_{\text{flag}}\right) - \sigma_{\text{ReLU}}\left(b_- - Aa_{\text{flag}}\right) \\ &= a_{\text{val}} + \sigma_{\text{ReLU}}\left(b_{\text{val}} - Aa_{\text{flag}}\right) - \sigma_{\text{ReLU}}\left(-b_{\text{val}} - Aa_{\text{flag}}\right), \end{aligned} \tag{13}$$

where $A$ is a vector with large positive entries that ensures the ReLU activation zeroes out the non-selected parts based on the flag. When $a_{\text{flag}} = 1$, the terms involving $b$ are suppressed, resulting in $c = a$. Conversely, when $a_{\text{flag}} = 0$, the positive part of $b$ remains, effectively selecting $b$. Thus, $f_{\text{resmlp}}$ accurately implements the `Option::or` function, demonstrating that it is representable within $\text{ResMlp}_1^{2(d+1)}$. $\square$

**Proposition 10** (Field Access Is Representable in ResMlp). *For algebraic data type, either struct field access, enum discriminator, and variant field access can be represented in* $\text{ResMlp}_1^d$ *where d is the encoding dimension.*

*Proof.* Obvious because these operations are linear. $\square$

**Proposition 11.** *[Composition of Functions Representable in ResMlp] For local types* $\mathcal{T}, \mathcal{S}, \mathcal{R}$, *with maps* $f : \mathcal{T} \to \mathcal{S}$ *and map* $g : \mathcal{S} \to \mathcal{R}$ *representable in* $\text{ResMlp}_{L_1}^{d_1}$ *and* $\text{ResMlp}_{L_2}^{d_2}$ *respectively. Then* $g \circ f$ *is representable in* $\text{ResMlp}_{L_1+L_2}^{\max\{d_1,d_2\}}$.

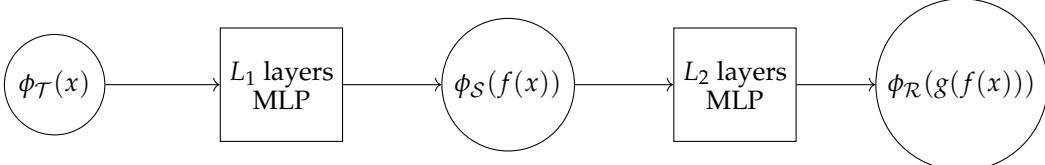

Figure 3: Transformation from $\phi_{\mathcal{T}}(x)$ to $\phi_{\mathcal{S}}(f(x))$ to $\phi_{\mathcal{R}}(g(f(x)))$ with MLP layers.

*Proof.* Obvious by using the first $L_1$ layers to map from $\mathcal{T}$ to $\mathcal{S}$ and using the rest $L_2$ layers to map from $\mathcal{S}$ to $\mathcal{R}$. The process can be visualized as in Figure 3.

$\square$

**Proposition 12.** *[Computation Graph of Functions Representable in ResMlp] Let $\mathcal{G}$ be a computation graph, with each vertex $v$ being of some local type $\mathcal{T}_v$, and the construction functions are representable in* $\text{ResMlp}_{L_v}^{d_v}$. *For convenience, if $v$ is a source vertex, $d_v$ is defined to be the encoding dimension of $\mathcal{T}_v$ and $L_v = 0$. Then the function induced by the computation graph is representable in* $\text{ResMlp}_{\text{Depth}(\mathcal{G})(\max_{v \in \mathcal{G}} L_v + 1) + 1}^{\sum_{v \in \mathcal{G}} d_v}$.

*Proof.* We construct a global residual multi-layer perceptron (ResMLP) that simulates the computation graph $\mathcal{G}$ by aggregating and updating the states of all vertices simultaneously. Let $D = \sum_{v \in \mathcal{G}} d_v$ be the total dimension, where $d_v$ is the dimension associated with vertex $v$. The global ResMLP will have a depth of $\text{Depth}(\mathcal{G})(\max_v L_v + 1)$.

Consider the concatenated state vector $X^{(t)} \in \mathbb{R}^D$, which is a concatenation of the states of all vertices:

$$X^{(t)} = \left( X_v^{(t)} \right)_{v \in \mathcal{G}},$$

where $X_v^{(t)} \in \mathbb{R}^{d_v}$ is the state of vertex $v$ at layer $t$.

Initialization occurs at depth zero, corresponding to the source vertices of the computation graph. The state vector $X^{(0)}$ is set by assigning the input vectors to the source vertices and initializing all other vertices to zero. Formally, if $V_0$ denotes the set of source vertices, then:

$$X_v^{(0)} = \begin{cases} x_v & \text{if } v \in V_0, \\ 0 & \text{otherwise,} \end{cases}$$

where $x_v \in \mathbb{R}^{d_v}$ is the input to source vertex $v$. Because $X_v^{(0)}$ is of dimensionality $d_v$ equal to the encoding dimension, this agrees with our convention for representing functions over local types.

We proceed inductively over the depth levels of the computation graph. For each depth level $k = 1, 2, \dots, \text{Depth}(\mathcal{G})$, we perform the following steps in the global ResMLP.

1. Input Aggregation Layer. We apply a linear transformation to gather the outputs from the predecessor vertices of each vertex at depth $k$ and feed them as inputs to these vertices. Specifically, we define a linear mapping $W_{\text{agg}}^{(k)} \in \mathbb{R}^{D \times D}$ such that:

   $$\tilde{X}^{(t_k)} = W_{\text{agg}}^{(k)} X^{(t_{k-1})},$$

   where $t_{k-1}$ is the layer after processing depth $k - 1$, and $\tilde{X}^{(t_k)}$ is the aggregated input for the vertices at depth $k$. The matrix $W_{\text{agg}}^{(k)}$ rearranges and combines the outputs from predecessor vertices to provide the correct inputs to each vertex at depth $k$. Specifically, for each vertex $v$ at depth $k$, and for each predecessor $u$ of $v$ in the computation graph, the matrix $W_{\text{agg}}^{(k)}$ contains entries that copy the output of $u$ into the input positions of $v$. All other entries in $W_{\text{agg}}^{(k)}$ are set to zero or identity as appropriate.

2. Local Computation Layers. For each vertex $v$ at depth $k$, we simulate its local ResMLP of depth $L_v$. Since the depths $L_v$ may vary, we pad the local ResMLPs to have a uniform depth $L = \max_v L_v$ by adding identity mappings where necessary. The updates for vertex $v$ are computed as:

$$X_v^{(t_k+1)} = \tilde{X}_v^{(t_k)} + f_{\text{fcn}_v}\left(\tilde{X}_v^{(t_k)}\right),$$
$$X_v^{(t_k+k')} = X_v^{(t_k+k'-1)} + f_{\text{fcn}_v}\left(X_v^{(t_k+k'-1)}\right), \quad \text{for } k' = 2,\ldots,L_v,$$
$$X_v^{(t_k+k')} = X_v^{(t_k+k'-1)}, \quad \text{for } k' = L_v+1,\ldots,L.$$

Here, $f_{\text{fcn}_v}$ denotes the single-layer fully connected network (as per Definition 7) for vertex $v$.

3. State Update. After completing the local computations for depth $k$, we update the global state vector $X^{(t_k+L)}$ by concatenating the updated states of all vertices:

$$X^{(t_k+L)} = \left(X_v^{(t_k+L)}\right)_{v\in\mathcal{G}}.$$

The total number of layers added for depth $k$ is $L+1$, accounting for the input aggregation layer and the $L$ layers simulating the local ResMLPs.

By repeating this process for each depth level $k = 1, 2, \ldots, \text{Depth}(\mathcal{G})$, we simulate the entire computation graph within a global ResMLP of depth $\text{Depth}(\mathcal{G})(\max_v L_v + 1)$.

Lastly, we use the final layer to perform a linear mapping so that the output is in the correct linear representation, clearing out the intermediate values.

Therefore, the function computed by the global ResMLP is equivalent to the function induced by the computation graph $\mathcal{G}$, and it is representable in $\text{ResMlp}^D_{\text{Depth}(\mathcal{G})(\max_v L_v+1)}$. $\square$

*Remark* 5. We only prove things around MLPs here. Later, we shall show that this will imply that the induced map operation over sequences can be represented by transformers.

### D.6 Functions over Global Types

The task we want transformers to express is too complicated to be cleanly described in one shot. So we introduce the following lemma to significantly simplify things. The lemma shall be useful for our future papers on this topic.

**Proposition 13.** *[Composition of Functions Representable in Transformers] For local types $\mathcal{T}$, $\mathcal{S}$, $\mathcal{R}$, with maps $f : \mathcal{T}^* \to \mathcal{S}^*$ and $g : \mathcal{S}^* \to \mathcal{R}^*$ representable in $\text{Tf}^{d_1}_{H_1,L_1}$ and $\text{Tf}^{d_2}_{H_2,L_2}$ respectively. Then the composition $g \circ f$ is representable in $\text{Tf}^{\max\{d_1,d_2\}}_{\max\{H_1,H_2\},L_1+L_2}$.*

*Proof.* This is basically the same as the proof of Proposition 11. $\square$

**Proposition 14.** *[Computation Graphs of Functions Representable in Transformers] Suppose we have a computation graph $G = (V, E)$ with types $\mathcal{T}_v = T_v^*$ together with encoding map $\psi_v : T_v \to \mathbb{R}^{d_v}$ and decoding map $\phi_v : \mathbb{R}^{d_v} \to T_v$, satisfying $\phi_v \circ \psi_v \equiv \text{id}_{T_v}$, and there exists some positive integer $d_0$ such that for each $v \in V$, $f_v$ can be represented in*

$$\text{Tf}^d_{H_v,L_v}$$

*Let $f$ be the function generated by the computation graph. Then $f$ can be represented in $\text{Tf}^d_{H,L}$ if $d \geq \sum_v d_v + Hd_0$, $L \geq \frac{|G|}{H} + d_G$ where $d_G$ is the depth of the graph.*

*Remark* 6. This doesn't really cover the above. The bound in Proposition 14 isn't always tight for model dimension when the computation graph is deep and Proposition 13 complements it.

*Proof.* WLOG, assume that $d = \sum_{v \in V} d_v + Hd_0$. Then

$$\mathbb{R}^d = \underbrace{\left( \bigoplus_{v \in V} \mathbb{R}^{d_v} \right)}_{C} \oplus \underbrace{\left( \bigoplus_{h \in [H]} \mathbb{R}^{d_0} \right)}_{A}. \tag{14}$$

Here $C$ stands for "cache" used for storing computed values, and $A$ stands for "active" used for storing intermediate computation results.

Make an order of all the nodes in the graph, say $V = \left\{ v_1, \ldots, v_{|G|} \right\}$ such that $\text{Depth}(v_i) \leq \text{Depth}(v_j)$ if $i \leq j$.

We now imagine the transformer computation process as gradually evaluating the value of each vertex, starting from $v_1$ to $v_{|G|}$. Every $\max_v L_v$ layers form a layer group, and after each layer group, at most $H$ vertices are assigned values. The equation 14 implies that we have enough memory to cache the computed values and intermediate values in small transformers.

Now let this process continue until we compute all the values. It must be finite because after each layer group, at least one of the vertices is computed. But this bound is too loose. We claim the following:

**Claim**: the number of layer groups where less than $H$ vertices are assigned values is smaller than $\text{Depth}(G)$.

**Sketch of Proof of Claim**: for any layer group where less than $H$ vertices are assigned, all the vertices that aren't assigned after this layer group must have larger depth than any vertices that are assigned values before this layer group, otherwise such a vertice can be evaluated in this layer group. Define the depth of any layer group to be the smallest depth of vertices evaluated in this layer group. Then for any unsatiated layer group, it must have a larger depth than the previous layer group. But depth can only increase $\text{Depth}(G)$ times, thus there are at most $\text{Depth}(G)$ unsatiated layer groups.

**Proof of Claim**: let $V_1, \ldots, V_l$ be the vertices evaluated at each layer group. Note that $l$ is a different symbol than $L$ and means that the number of layer groups rather than the number of layers.

For convenience, let $D_i$ be the minimum of the depths of vertices in $V_i$.

Suppose that the $i$th layer group is unsatiated, then $i < l$. We want to show that $D_i < D_{i+1}$. Suppose otherwise, i.e., $D_i = D_{i+1}$. Because the $i$th layer group is unsatiated, for any $v \in V_{i+1}$, $v$ must have dependencies that haven't been evaluated before the $i$th layer group. Choose $v_0 \in V_i, v_1 \in V_{i+1}$ such that $\text{Depth}(v_0) = \text{Depth}(v_1) = D_i = D_{i+1}$. Note that any dependency of $v_1$ must have smaller depths than $v_0$, then must have already be evaluated before the $i$th layer group. Contradiction!

Now given the claim, we have that for all but at most $\text{Depth}(G)$ choices of $i = 1, \ldots, l$, we have $|V_i| = H$, then we have

$$|G| = \sum_{i=1}^{l} |V_i| \geq (l - \text{Depth}(G))H \tag{15}$$

Then $l \leq \frac{|G|}{H} + \text{Depth}(G)$.

Then $L \leq l \cdot \max_{v \in G} L_v = \left( \frac{|G|}{H} + \text{Depth}(G) \right) \max_{v \in G}$.

$\square$

**Proposition 15.** *[Nearest Left/Right] For any local type* `T` *, consider the function that maps a sequence of type* `Option<T>` *to nearest left/right neighbors that are not none. It's representable in* $\text{Tf}_{1,1}^{d+1}$

*Proof.* There is only one layer and one head needed, so we can omit the layer and head index.

WLOG, we consider the nearest left case.

We just need to make the attention exponential look like this:

$$Q_p^\top K_{p'} + \lambda \Psi_{p'-p} = a_{\text{flag},p'} - 1_{p'-p>0}, \tag{16}$$

where $a_{\text{flag},p'} \in \{0,1\}$ indicates whether the value at position $p'$ is some or none.

We set $V_{p'}$ to represent the value of type `Option<T>`.

For the starter token $p_0$, we make it such that

$$Q_p^\top K_{p_0} + \lambda \Psi_{p_0-p} = 1, \tag{17}$$

and

$$V_{p_0} = \mathbf{0}, \tag{18}$$

so that when there are no some to the left, it will give us none. $\square$

**Proposition 16.** *[Nearest Two Left/Right] For any local type* `T` *, consider the function that maps a sequence of type* `Option<T>` *to nearest two left/right neighbors that are not none. It's representable in* $\text{Tf}_{O(1),O(1)}^{O(d)}$ *where d is the encoding dimension of* `T` *.*

*Proof.* We can utilize Proposition 15 and 14.

The nearest two left or right is equivalent to first computing the nearest left/right, and then packing them together into one and compute its nearest left/right. The process is represented by a small constant computation graph, then we're done. $\square$

### D.7 Syntax and Semantics of Cybertron

Having laid the necessary mathematical foundation behind **Cybertron**, we now turn to explaining its surface—its **syntax** and **semantics**. Cybertron serves as a syntax sugar for expressing local and global computation graphs, which are the vehicles used to demonstrate the expressive power of transformers. In Cybertron, computations are divided into two layers: the **local world** and the **global world**. These layers play distinct but complementary roles in constructing computation graphs.

#### D.7.1 Local World

The **local world** in Cybertron corresponds to the feed-forward layers of a transformer, focusing on computations over **local types**. Local types represent individual tokens or data points, and computations in this world handle operations on tokens independently of their surrounding context.

**Data Types.** Local types in Cybertron include basic types such as `Bool`, `Idx`, `Pos`, `Fin<n>`, `BoundedVec<T, N>`, etc. These types are essential for building local computation graphs that operate over individual tokens. Compound types, like **structs** and **enums**, can also be defined for more complex token representations. These types serve as the building blocks for the local computation graphs that transform data at the token level.

```
1  struct Node {
2      id: Idx,
3      position: Pos,
4  }
5
6  enum Operation {
7      Add {
8          lhs: Pos,
9          rhs: Pos,
```

```
10      },
11      Multiply {
12          factor: Pos,
13      },
14  }
```

**Functions.**   Functions in the local world define operations upon information over individual tokens. These operations form nodes in the local computation graphs. For instance, operations like binary or unary expressions, conditionals, and matches on token types are transformed into computation graphs by handling each individual token's data.

```
1   fn process_ast(ast: AstData) -> Option<Role> {
2       match ast {
3           AstData::LetInit { pattern, initial_value, .. } => {
4               Some(Role::LetStmt { pattern, initial_value })
5           }
6           AstData::Defn { keyword, ident, .. } => {
7               Some(match keyword {
8                   DefnKeyword::Struct => Role::StructDefn(ident),
9                   DefnKeyword::Enum => Role::EnumDefn(ident),
10                  DefnKeyword::Fn => Role::FnDefn(ident),
11              })
12          }
13          _ => None,
14      }
15  }
```

**Control Flow.**   In the local world, control flow structures such as `if` and `match` are transformed into computation graphs by treating each branch or arm as an expression that returns an `Option` based on conditions. These `Option` values are then combined using the `Option::or` function. According to **Proposition 9**, `Option::or` maps two `Option<T>` values and returns the first non-`None` value, or the second one otherwise. This allows conditional branches to be represented in computation graphs as sequential option evaluations, where the first matching condition provides the result.

### D.7.2   Global World

The **global world** extends beyond individual tokens to sequences of tokens, represented as **global types**. These global types are denoted as `Seq<T>`, where `T` is a local type. The global world represents the full transformer, focusing on operations involving sequences of tokens, including variable definitions, expressions involving variable references, and function calls.

**Variable Definitions.**   Variables in the global world are defined using global types, which represent sequences of local tokens. These definitions correspond to nodes in the global computation graph.

**Expressions.**   Expressions in the global world consist of references to variables or function calls. Since the global world operates over sequences of tokens, these expressions are translated into sequence-level operations in the computation graph.

**Function Calls.**   Function calls are key elements of the global world. They are represented by applying global functions to sequences of tokens. Cybertron provides **map functions** to elevate local functions to global functions by mapping them across sequences. Additionally, **attention methods** like `nearest_left` and `nearest_right` handle dependencies between tokens in the sequence by identifying relationships based on their positions.

```
1   let result = seq_of_values.nearest_left();
```

In the global world, computation graphs are built by composing map functions and attention methods. These graphs, unlike those in the local world, do not include control flow mechanisms.

### D.8 Dyck Language

This section demonstrates how the **local world** in Cybertron operates over token-level computations and how the **global world** handles sequence-level operations. We use a Dyck language example to explain the interactions between these two worlds. The example processes a sequence of delimiters (like parentheses) and checks for matching pairs.

**Local World.** In Cybertron, the **local world** operates on individual tokens. Here, the local types are simple, such as `Delimiter` and `PreAst`, which represent information associated with individual tokens. These types allow for token-level operations like comparisons and transformations.

We define a `struct` to represent a delimiter and an `enum` to classify delimiters as either left or right. These definitions reflect local types, as they hold information over a single token.

```
1  // Define a struct `Delimiter` that wraps a `u8` value.
2  #[derive(Debug, Clone, Copy, PartialEq, Eq)]
3  pub struct Delimiter(u8);
4
5  // Define an enum `PreAst` which represents a left or right delimiter.
6  #[derive(Debug, Clone, Copy, PartialEq, Eq)]
7  pub enum PreAst {
8      LeftDelimiter(Delimiter),
9      RightDelimiter(Delimiter),
10 }
```

Here, the local types `Delimiter` and `PreAst` define operations upon individual tokens, representing fundamental units of computation graph at the local level. The local world is responsible for handling these small, token-level computations independently of the global sequence.

**Global World.** In the **global world**, Cybertron operates on sequences of tokens, treating the collection of local types as a single unit of computation. The global world introduces global types such as `Seq<Option<PreAst>>`, which represents a sequence of optional delimiters.

The global world handles sequence-level operations by applying functions like `nearest_left` and `nearest_right` to capture the relationships between tokens in the sequence.

The following function operates on a sequence of `PreAst`, reducing matched pre-asts. The recursive application of `step` gives us the classifier for Dyck language.

```
1  fn step(pre_asts: Seq<Option<PreAst>>) -> Seq<Option<PreAst>> {
2      let pre_asts_nearest_left = pre_asts.nearest_left();
3      let pre_asts_nearest_right = pre_asts.nearest_right();
4      step_aux.apply(pre_asts_nearest_left, pre_asts, pre_asts_nearest_right)
5  }
```

**Local Worlds.** The `step_aux` function matches tokens based on their nearest neighbors within the sequence, eliminating pre-asts if a match is found.

```
1  fn step_aux(
2      pre_ast_nearest_left: Option<(Idx, PreAst)>,
3      pre_ast: Option<PreAst>,
4      pre_ast_nearest_right: Option<(Idx, PreAst)>
5  ) -> Option<PreAst> {
6      match pre_ast? {
7          PreAst::LeftDelimiter(delimiter) => match pre_ast_nearest_right {
8              Some((_, PreAst::RightDelimiter(delimiter1))) if delimiter1 == delimiter => None,
9              _ => pre_ast,
10         },
11         PreAst::RightDelimiter(delimiter) => match pre_ast_nearest_left {
12             Some((_, PreAst::LeftDelimiter(delimiter1))) if delimiter1 == delimiter => None,
13             _ => pre_ast,
14         },
15     }
16 }
```

In this example, the global function `step` uses `nearest_left` and `nearest_right` to capture sequence-level dependencies, while the local function `step_aux` uses conditional logic to check for matching pairs of delimiters. The local world handles token-level logic, while the global world coordinates operations across the entire sequence. This separation reflects how Cybertron handles computations at different levels of granularity.

Thus, this example illustrates how Cybertron leverages both the local and global worlds to build comprehensive computation graphs in a convenient, comprehensive yet rigorous manner. The local world performs individual tokenwise operations, and the global world captures relationships between tokens in a sequence, demonstrating how Cybertron enables transformers to express complex computations.

## E   Mini-Husky Details

Here's the BNF grammar of the Mini-Husky language:

⟨ast⟩ ::= ⟨literal⟩
        | ⟨ident⟩
        | ⟨prefix⟩
        | ⟨binary⟩
        | ⟨suffix⟩
        | ⟨delimited⟩
        | ⟨separated_item⟩
        | ⟨call⟩
        | ⟨let_init⟩
        | ⟨if_stmt⟩
        | ⟨else_stmt⟩
        | ⟨defn⟩

⟨literal⟩ ::= ...

⟨ident⟩ ::= ...

⟨prefix⟩ ::= ⟨prefix_opr⟩ ⟨ast⟩

⟨binary⟩ ::= ⟨ast⟩ ⟨binary_opr⟩ ⟨ast⟩

⟨suffix⟩ ::= ⟨ast⟩ ⟨suffix_opr⟩

⟨delimited⟩ ::= ⟨left_delimiter⟩ ⟨separated_item⟩* ⟨right_delimiter⟩

⟨separated_item⟩ ::= [⟨ast⟩] ⟨separator⟩

⟨call⟩ ::= ⟨ast⟩ ⟨left_delimiter⟩ ⟨ast⟩* ⟨right_delimiter⟩

⟨let_init⟩ ::= let ⟨ast⟩

⟨if_stmt⟩ ::= if ⟨ast⟩ ⟨delimited⟩

⟨else_stmt⟩ ::= ⟨if_stmt⟩ else (⟨delimited⟩ | ⟨else_stmt⟩)

⟨defn⟩ ::= ⟨defn_keyword⟩ ⟨ident⟩ ⟨ast⟩

⟨prefix_opr⟩ ::= + | - | ! | ...

⟨binary_opr⟩ ::= + | - | * | / | ...

⟨suffix_opr⟩ ::= ++ | -- | ...

⟨left_delimiter⟩ ::= '(' | [ | {

⟨right_delimiter⟩ ::= ')' | ] | }

⟨separator⟩ ::= , | ;

⟨defn_keyword⟩ ::= def | fn | ...

Below is a sample piece of codes:

```
1  struct Dog { weight: f32, .. }
2
3  fn see_vet(dog: Dog) -> f32 {
4      assert dog.weight < 100;
```

```
5     let mut fee = dog.weight * 10.0;
6     fee +=100.0;
7     return fee
8  }
```

It should be noted that the above is not the full story. There are additional constraints put on the ASTs. However, these can be easily implemented as tree functions that are easy for transformers to express. As we are focusing on higher level language processing capabilities, we ignore the details here.

Additionally, we need to require that for semantic correctness, we must have proper symbol resolution and type correctness.

### E.1 Additional Details about Compiler Tasks.

The outputs of the tasks are defined using Cybertron as follows:

- The construction of AST task's final output is the collection all AST nodes. More concretely, the output is a sequence of Option<Ast> with length equal to the input token sequence's length, where Option<Ast> denoted the type Ast will a null value added and Ast is the type storing the information of a node, including its parent, and its data of type AstData . In Cybertron, we define Ast and AstData explicitly as follows:

```
1  /// Represents a node in an Abstract Syntax Tree (AST).
2  ///
3  /// Each `Ast` node has a reference to its parent node (if any) and holds
4  /// the associated syntax data (such as expressions, statements, or other
5  /// constructs defined in the `AstData` enum).
6  pub struct Ast {
7      /// The index of the parent node in the AST, if it exists.
8      ///
9      /// - `Some(Idx)`: The node has a parent, and `Idx` represents its position.
10     /// - `None`: The node is the root or does not have a parent.
11     pub parent: Option<Idx>,
12     /// The data associated with this AST node.
13     pub data: AstData,
14 }
15
16 /// Enumeration representing different types of Abstract Syntax Tree (AST) nodes
17 pub enum AstData {
18     /// Represents a literal value (e.g., integer, string)
19     Literal(Literal),
20     /// Represents an identifier (e.g., variable name)
21     Ident(Ident),
22     /// Represents a binary expression (e.g., `x + y`, `a * b`)
23     Binary {
24         /// Index of the left operand
25         lopd: Idx,
26         /// Operator in the binary expression (e.g., `+`, `*`)
27         opr: BinaryOpr,
28         /// Index of the right operand
29         ropd: Idx,
30     },
31     ... // other variants
32 }
```

- The output of the **symbol resolution** task is the collection of symbol resolution results on all applicable tokens. More concretely, the output is a sequence of values of type Option<SymbolResolution> where Option<SymbolResolution> is the type SymbolResolution with a null value added for non-applicability and SymbolResolution is the type storing the result of the symbol resolution, being either a success with a resolved symbol of type Symbol or a failure with an error of type SymbolResolutionError . In Cybertron, we define SymbolResolution explicitly as follows:

```
1  // an enum type definition, basically a tagged union type
2  pub enum SymbolResolution {
3      Ok(Symbol), // enum type variant for success with a resolved symbol
4      Err(SymbolResolutionError), // enum type variant for failure with an error
5  }
```

- The **type analysis** task's final output is the collection of all type errors. More concretely, the output is a sequence of `Option<TypeError>`, where `Option<TypeError>` denoted the type `TypeError` will a null value added and `TypeError` is the type storing the information of a type error. The position of type errors agrees with the source tokens leading to these errors. In Cybertron, we define `TypeError` explicitly as follows:

```
1  // This enum represents various kinds of type errors
2  pub enum TypeError {
3      // This variant indicates a type mismatch
4      // `expected` is the type that was anticipated
5      // `actual` is the type that was encountered
6      TypeMismatch { expected: Type, actual: Type },
7  }
```

One can expand the definition to include other kinds of type errors.

(1) *Type definition.* Types are either identified uniquely by a single identifier like `<identifier>`, or builtin generic types `Option<<identifier>>` or `Vec<<identifier>>`. Users can define custom types without generics like the following (f32 means float32 and i32 means int32 below):

```
1  struct Dog { weight: f32 }
2
3  enum Animal {
4      Dog,
5      Cat,
6  }
```

This part is actually a part of the AST task and type definition is a variant of the `AstData` type:

```
1  /// Enumeration representing different types of Abstract Syntax Tree (AST) nodes
2  pub enum AstData {
3      ...
4      /// Represents a function or variable definition
5      ///
6      /// # defn
7      ///
8      Defn {
9          /// The keyword in the definition (e.g., `fn`, `enum`)
10         keyword: DefnKeyword,
11         /// Index of the identifier in the definition
12         ident_idx: Idx,
13         /// The identifier being defined (e.g., function name, variable name)
14         ident: Ident,
15         /// Index of the content or body of the definition
16         content: Idx,
17     },
18 }
```

(2) *Type specification.* Each appeared variable has a unique type, either by specification or speculation. All parameters of a function must be specified explicitly by users. Variables defined by let statements might or might not be specified, as follows:

```
1  fn f(a: i32) { // type of `a` must be specified
2      let x: i32 = a; // type of `x` specified
3      let y = a; // type of `y` unspecified
4  }
```

The return type of functions must be specified. The field type of structs and enum variants must be specified. the type of expressions of function calls and field access will be determined correspondingly.

The output of the task is the collection of all type signatures, represented as a sequence of values of type `Option<TypeSignature>` where `TypeSignature` is the type holding the essential information of type specifications. In Cybertron, `TypeSignature` is defined as,

```
1  pub struct TypeSignature {
2      pub key: TypeSignatureKey,
3      pub ty: Type,
```

```
 4   }
 5
 6   pub enum TypeSignatureKey {
 7       FnParameter { fn_ident: Ident, rank: Rank },
 8       FnOutput { fn_ident: Ident },
 9       StructField { ty_ident: Ident, field_ident: Ident },
10   }
```

(3) *Type inference.* As discussed above, not all variables have their types specified.

```
1   fn f() {
2       let x: i32 = 1;
3       let y = x;
4       let z = y;
5   }
```

In the above code, 1 is an ambiguous literal that can be of type `i32`, `i64`, `u32`, `u64`, etc, and the types of `y` and `z` is not specified. However, one easily sees that there exists one and only one choice of the types of `1`, `y`, and `z` such that the whole code is type correct. Utilizing this property, the user can opt out of a significant portion of type specification, achieving static guarantees.

**A Type Inference Algorithm:** For simplicity, we shall prove transformers can implement a simple type inference algorithm: we maintain a table of type assignments for variables. We update the entries of the table by means of reduction, i.e., assuming the whole code is correctly typed and infer more and more unspecified types until we encounter errors or all types are inferred. The process is largely parallel, and we call the number of rounds needed the depth of type inference.

In the above code, the first round, we determine that the type of both `1` and the type of `y` are equal to the type of `x` which is `i32`. But we have no way to determine the type of `z` because the type of `y` is unknown at the first round. In the second round, `z` can be determined to be of type `i32` because the type of `y` is already inferred.

The output of the task is the collection all types inferred, represented as a sequence of values of type `Option<TypeInference>` where `TypeInference` is the type holding the inferred type. In Cybertron, `TypeSignature` is defined as,

```
1   pub struct TypeInference {
2       pub ty: Type,
3   }
```

# F    Transformer AST Proof

## F.1    High Level Overview

*Proof Sketch of Theorem 1.* The idea is to construct ASTs in a bottom-up manner with full parallelism. We shall recursively produce the final ASTs in at most $D$ steps. We shall maintain two values, called `pre_asts` and `asts`. `asts` represents ASTs that have already been allocated, although they might not have been fully initialized. `pre_asts` represents tokens that have yet to form ASTs and new ASTs that have not been fully initialized. For each round, we try to create new ASTs from `pre_asts` and update `asts` and `pre_asts`. For the $n$-th round, we provably allocated all ASTs with a depth no more than $n$. Then for the $D$-th round, all ASTs are properly constructed and allocated. Each round can be represented by a transformer of $O(1)$ number of heads, model dimension $O(\log L + D)$, and $O(1)$ number of layers. Therefore, the end-to-end process is then representable by a transformer of $O(1)$ number of heads, model dimension $O(\log L + D)$, and $O(\log L + D)$ number of layers.    $\square$

Here we give the full details of the proof of transformers being able to parse ASTs.

On a high level, we are going to see the parsing of ASTs as an assembly process. First, we immediately get the atomic ones, like identifiers, literals, etc. Then we assembly all composite ASTs with enough precedence util all tokens are consumed. We can prove that at the $n$th round, all ASTs with depth no more than $n$ are already constructed. In the process, we must keep track of the unconsumed tokens and newly constructed ASTs (to be consumed as children for new ASTs in the next round, as we are going bottom up). We use `pre_asts` to denote all the unconsumed tokens and newly constructed ASTs and use `asts` to denote all the constructed(allocated) ASTs. For correctness guarantees, we give detailed type specifications for tokens, ASTs, and PreASTs as follows.

We define the Token type as follows:

```
1   /// The `Token` enum represents the various types of tokens that can be
2   /// identified during the lexical analysis phase of a compiler. Each variant
3   /// corresponds to a specific category of token that can be encountered
4   /// in the source code.
5   pub enum Token {
6       /// A literal value, which can be a number, string, or other primitive type.
7       Literal(Literal),
8       /// A reserved keyword in the language, such as `if`, `else`, `while`, etc.
9       Keyword(Keyword),
10      /// An identifier, typically representing variable names, function names,
11      /// or other user-defined symbols.
12      Ident(Ident),
13      /// An operator, such as `+`, `-`, `*`, `==`, etc., representing mathematical
14      /// or logical operations.
15      Opr(Opr),
16      /// A left delimiter, such as `(`, `{`, `[`, used to denote the beginning of
17      /// a block, list, or expression.
18      LeftDelimiter(LeftDelimiter),
19      /// A right delimiter, such as `)`, `}`, `]`, used to denote the end of a
20      /// block, list, or expression.
21      RightDelimiter(RightDelimiter),
22      /// A separator, such as `,` or `;`, used to separate elements in a list or
23      /// statements in a block.
24      Separator(Separator),
25  }
```

The type has an encoding dimenion $d_{\text{Token}} = \Theta(\log L)$, which is large enough to faithfully represent its information.

More specifically, the types `Literal`, `Keyword`, `Ident`, `Opr`, `LeftDelimiter`, `RightDelimiter`, `Separator` are local types assumed to have encoding dimension less than $d_{\text{Token}}$. `Keyword`, `Opr`, `LeftDelimiter`, `RightDelimiter`, `Separator` are small, so they can be encoded in a straight-forward manner entirely using $d_{\text{Token}}$. However, `Literal` and `Ident` are larger than representable by a limited number of bits because potentially a `Literal` can be a string literal of arbitrary length and an `Ident` can also be of arbitrary length. This can be solved through methods like interning, which gives all literals and identifiers that actually appear in the input distinct encodings. As the context length is $L$, the number of different literals/identifiers are bounded by context length and interning needs $O(d_{\text{Token}}) = O(\log L)$ to work. As far as our theories are concerned, it's totally reasonable to assume that all these types are assumed to have encoding dimension less than $d_{\text{Token}} = O(\log L)$.

We define AST type as follows:

```
1   /// Represents a node in an Abstract Syntax Tree (AST).
2   ///
3   /// Each `Ast` node has a reference to its parent node (if any) and holds
4   /// the associated syntax data (such as expressions, statements, or other
5   /// constructs defined in the `AstData` enum).
6   pub struct Ast {
7       /// The index of the parent node in the AST, if it exists.
8       ///
9       /// - `Some(Idx)`: The node has a parent, and `Idx` represents its position.
10      /// - `None`: The node is the root or does not have a parent.
11      pub parent: Option<Idx>,
12      /// The data associated with this AST node.
13      ///
14      /// This field holds the actual syntax information, which is typically
```

```
15      /// defined by the `AstData` enum. This could represent literals, expressions,
16      /// statements, and other constructs in the source language.
17      pub data: AstData,
18  }
```

Note that we intentionally structure the tree by always storing the parent but not necessarily storing all children information. In our assumptions, we only control the depth of ASTs but don't control the number of children. More specifically, a function can have as many statements as possible. To avoid overflowing, we don't store all children information. As we shall see, parent information alone is enough for transformers to perform tree operations.

The AstData is the most complicated we define in this paper, as follows:

```
1   /// Enumeration representing different types of Abstract Syntax Tree (AST) nodes
2   pub enum AstData {
3       /// Represents a literal value (e.g., integer, string)
4       Literal(Literal),
5       /// Represents an identifier (e.g., variable name)
6       Ident(Ident),
7       /// Represents a prefix expression (e.g., `!x`, `-x`)
8       ///
9       /// # exprs
10      ///
11      Prefix {
12          /// Operator in the prefix expression (e.g., `!`, `-`)
13          opr: PrefixOpr,
14          /// Operand index of the expression
15          opd: Idx,
16      },
17      /// Represents a binary expression (e.g., `x + y`, `a * b`)
18      Binary {
19          /// Index of the left operand
20          lopd: Idx,
21          /// Operator in the binary expression (e.g., `+`, `*`)
22          opr: BinaryOpr,
23          /// Index of the right operand
24          ropd: Idx,
25      },
26      /// Represents a suffix expression (e.g., `x++`, `y--`)
27      Suffix {
28          /// Index of the operand
29          opd: Idx,
30          /// Operator in the suffix expression (e.g., `++`, `--`)
31          opr: SuffixOpr,
32      },
33      /// Represents a delimited expression (e.g., `(x + y)`, `{a, b, c}`)
34      Delimited {
35          /// Index of the left delimiter in the expression
36          left_delimiter_idx: Idx,
37          /// The left delimiter (e.g., `(`, `{`)
38          left_delimiter: LeftDelimiter,
39          /// The right delimiter (e.g., `)`, `}`)
40          right_delimiter: RightDelimiter,
41      },
42      /// Represents an item separated by a separator (e.g., elements in an array or list)
43      SeparatedItem {
44          /// Index of the content, if any
45          content: Option<Idx>,
46          /// The separator (e.g., `,`, `;`)
47          separator: Separator,
48      },
49      /// Represents a function call or array access (e.g., `f(...)`, `a[...]`)
50      ///
51      /// things like `f(...)` or `a[...]`
52      Call {
53          /// Index of the caller (e.g., function or array)
54          caller: Idx,
55          /// The left delimiter of the call (e.g., `(`, `[`)
56          left_delimiter: LeftDelimiter,
57          /// The right delimiter of the call (e.g., `)`, `]`)
58          right_delimiter: RightDelimiter,
59          /// Index of the delimited arguments in the call
60          delimited_arguments: Idx,
61      },
62      /// Represents a `let` statement with an initialization (e.g., `let x = 5;`)
63      ///
64      /// # stmts
65      ///
```

```
66      LetInit {
67          /// Index of the expression in the initialization
68          expr: Idx,
69          /// Index of the pattern being initialized
70          pattern: Idx,
71          /// Optional index of the initial value
72          initial_value: Option<Idx>,
73      },
74      /// Represents an `if` statement
75      If {
76          /// Index of the condition in the `if` statement
77          condition: Idx,
78          /// Index of the body of the `if` statement
79          body: Idx,
80      },
81      /// Represents an `else` statement
82      Else {
83          /// Index of the associated `if` statement
84          if_stmt: Idx,
85          /// Index of the body of the `else` statement
86          body: Idx,
87      },
88      /// Represents a function or variable definition
89      ///
90      /// # defn
91      ///
92      Defn {
93          /// The keyword in the definition (e.g., `fn`, `enum`)
94          keyword: DefnKeyword,
95          /// Index of the identifier in the definition
96          ident_idx: Idx,
97          /// The identifier being defined (e.g., function name, variable name)
98          ident: Ident,
99          /// Index of the content or body of the definition
100         content: Idx,
101     },
102 }
```

```
1  /// The `PreAst` enum represents the intermediate forms of tokens and ASTs that are
2  /// encountered during the parsing phase, before the final AST is constructed.
3  /// Each variant corresponds to a specific type of token or partial
4  /// AST node that contributes to the construction of the final AST.
5  #[derive(Clone, Copy, PartialEq, Eq)]
6  pub enum PreAst {
7      /// A reserved keyword in the language, such as `if`, `else`, `while`, etc.
8      Keyword(Keyword),
9      /// An operator, such as `+`, `-`, `*`, `==`, etc., representing mathematical
10     /// or logical operations.
11     Opr(Opr),
12     /// A left delimiter, such as `(`, `{`, `[`, used to denote the beginning of
13     /// a block, list, or expression.
14     LeftDelimiter(LeftDelimiter),
15     /// A right delimiter, such as `)`, `}`, `]`, used to denote the end of a
16     /// block, list, or expression.
17     RightDelimiter(RightDelimiter),
18     /// A partially constructed AST node, representing a more complex structure
19     /// that will be further processed to build the final AST.
20     Ast(AstData),
21     /// A separator, such as `,` or `;`, used to separate elements in a list or
22     /// statements in a block.
23     Separator(Separator),
24 }
```

```
1  /// this is beyond the scope of Cybertron
2  ///
3  /// rather a general Rust function to integrate for testing
4  pub fn calc_asts_from_input(input: &str, n: usize) -> (Seq<Option<PreAst>>, Seq<Option<Ast>>) {
5      let tokens = tokenize(input);
6      let pre_asts = calc_pre_ast_initial_seq(tokens);
7      let allocated_asts: Seq<Option<Ast>> = tokens.map(|token| token.into());
8      reduce_n_times(pre_asts, allocated_asts, n)
9  }
```

The reduce function in Cybertron is designed to progressively refine sequences of pre-abstract syntax trees (pre-ASTs) and allocated abstract syntax trees (ASTs). The function takes two input sequences: pre_asts , which is a sequence of optional pre-ASTs, and

allocated_asts , which is a sequence of optional ASTs. It returns a tuple containing the reduced sequences of pre-ASTs and allocated ASTs.

The reduction process is carried out in multiple stages, each focusing on different syntactic constructs:

1. reduce_by_opr : This step handles reduction by dealing with operators and their precedence. It simplifies expressions involving operations to form more compact ASTs.

2. reduce_by_delimited : This step reduces constructs that are delimited, such as those involving parentheses, braces, or other grouping symbols. It ensures that delimited blocks are properly nested and combined in the AST.

3. reduce_by_call : In this stage, function or method calls are reduced. This involves identifying and structuring calls within the AST, ensuring correct representation of function invocations.

4. reduce_by_stmt : This reduction step addresses statements, ensuring that individual statements are correctly parsed and represented within the AST, such as assignment statements, loops, and conditionals.

5. reduce_by_defn : Finally, reduction by definition handles the parsing of definitions, such as variable or function declarations. This step ensures that all definitions are correctly represented within the AST.

By sequentially applying these reduction steps, the reduce function progressively transforms the initial sequences into their most refined forms, ready for further syntactic or semantic analysis.

```
 1  pub fn reduce(
 2      pre_asts: Seq<Option<PreAst>>,
 3      allocated_asts: Seq<Option<Ast>>,
 4  ) -> (Seq<Option<PreAst>>, Seq<Option<Ast>>) {
 5      // Reduce ASTs by handling operators and precedence
 6      let (pre_asts, allocated_asts) = reduce_by_opr(pre_asts, allocated_asts);
 7
 8      // Reduce ASTs by handling delimited constructs like parentheses or braces
 9      let (pre_asts, allocated_asts) = reduce_by_delimited(pre_asts, allocated_asts);
10
11      // Reduce ASTs by handling function or method calls
12      let (pre_asts, allocated_asts) = reduce_by_call(pre_asts, allocated_asts);
13
14      // Reduce ASTs by handling statements, ensuring proper syntax structure
15      let (pre_asts, allocated_asts) = reduce_by_stmt(pre_asts, allocated_asts);
16
17      // Reduce ASTs by handling definitions, like variables or functions
18      let (pre_asts, allocated_asts) = reduce_by_defn(pre_asts, allocated_asts);
19
20      // Return the final reduced sequences of pre-ASTs and allocated ASTs
21      (pre_asts, allocated_asts)
22  }
```

```
 1  pub fn reduce_n_times(
 2      mut pre_asts: Seq<Option<PreAst>>,
 3      mut allocated_asts: Seq<Option<Ast>>,
 4      n: usize,
 5  ) -> (Seq<Option<PreAst>>, Seq<Option<Ast>>) {
 6      for _ in 0..n {
 7          let (pre_asts1, allocated_asts1) = reduce(pre_asts, allocated_asts);
 8          pre_asts = pre_asts1;
 9          allocated_asts = allocated_asts1;
10      }
11      (pre_asts, allocated_asts)
12  }
```

In the above definition, we actually used Rust's mutable variable semantics. However, it's straightforward to see that it translates to a computation graph that is a sequential composition of subgraphs with sequential length $n$. Because the AST's depth is bounded

by $D$, we can just take $n$ to be $D$. Each subgraph is generated from the reduce function, then they are all constant graphs constructed by global and local functions, then by Proposition 13,11 and 2 they translate to transformers with $O(\log L + D)$ depth, model dimension, and number of heads, where $\log L$ comes from the encoding of types like Token .

Below we give full details of the various reduction functions.

As these are implemented as Rust functions, they have been tested against a number of inputs. We don't guarantee an industry level of correctness, but the key point is well illustrated.

## F.2 Operators

In this section, we lay down the definition of reduce_by_opr .

```
1  pub(super) fn reduce_by_opr(
2      pre_asts: Seq<Option<PreAst>>,
3      allocated_asts: Seq<Option<Ast>>,
4  ) -> (Seq<Option<PreAst>>, Seq<Option<Ast>>) {
5      let pre_asts_nearest_left2 = pre_asts.nearest_left2();
6      let pre_asts_nearest_right2 = pre_asts.nearest_right2();
7      let new_opr_asts = new_opr_ast.apply(pre_asts_nearest_left2, pre_asts, pre_asts_nearest_right2);
8      let (pre_asts_reduced, new_parents) = reduce_pre_asts_by_opr(pre_asts, new_opr_asts);
9      let pre_asts = update_pre_asts_by_new_asts(pre_asts_reduced, new_opr_asts);
10     let allocated_asts =
11         allocate_asts_and_update_parents(allocated_asts, new_opr_asts, new_parents);
12     (pre_asts, allocated_asts)
13 }
```

```
1  /// a finite function
2  pub(crate) fn new_opr_ast(
3      nearest_left2: Option2<(Idx, PreAst)>,
4      current: Option<PreAst>,
5      nearest_right2: Option2<(Idx, PreAst)>,
6  ) -> Option<AstData> {
7      let Some(PreAst::Opr(opr)) = current else {
8          return None;
9      };
10     match opr {
11         Opr::Prefix(opr) => {
12             let Some((opd, PreAst::Ast(_))) = nearest_right2.first() else {
13                 return None;
14             };
15             if let Some((_, ast)) = nearest_right2.second() {
16                 match ast {
17                     PreAst::Keyword(_) => (),
18                     PreAst::Opr(right_opr) => match right_opr {
19                         Opr::Prefix(_) => (),
20                         Opr::Binary(right_opr) => {
21                             // every binary opr in our small language is left associative, so `<` instead of
                               `<=`
22                             if right_opr.precedence() > opr.precedence() {
23                                 return None;
24                             }
25                         }
26                         Opr::Suffix(right_opr) => {
27                             if right_opr.precedence() > opr.precedence() {
28                                 return None;
29                             }
30                         }
31                     },
32                     PreAst::Ast(_) => (),
33                     // function call or index takes higher precedence
34                     PreAst::LeftDelimiter(_) => return None,
35                     PreAst::RightDelimiter(_) => (),
36                     PreAst::Separator(_) => (),
37                 }
38             };
39             Some(AstData::Prefix { opr, opd })
40         }
41         Opr::Binary(opr) => {
42             let Some((lopd, PreAst::Ast(_))) = nearest_left2.first() else {
43                 return None;
44             };
45             let Some((ropd, PreAst::Ast(_))) = nearest_right2.first() else {
46                 return None;
```

```
47                };
48                if let Some((_, ast)) = nearest_left2.second() {
49                    match ast {
50                        PreAst::Keyword(kw) => (),
51                        PreAst::Opr(left_opr) => match left_opr {
52                            Opr::Prefix(left_opr) => {
53                                if left_opr.precedence() >= opr.precedence() {
54                                    return None;
55                                }
56                            }
57                            Opr::Binary(left_opr) => {
58                                /// every binary opr in our small language is left associative, so `>=` instead
        of `>`
59                                if left_opr.precedence() >= opr.precedence() {
60                                    return None;
61                                }
62                            }
63                            Opr::Suffix(_) => (), // actually this will be a syntax error
64                        },
65                        PreAst::Ast(_) => {
66                            if opr != BinaryOpr::LightArrow {
67                                return None;
68                            }
69                        }
70                        PreAst::LeftDelimiter(_) => (),
71                        PreAst::RightDelimiter(_) => return None,
72                        PreAst::Separator(_) => (),
73                    }
74                };
75                if let Some((_, ast)) = nearest_right2.second() {
76                    match ast {
77                        PreAst::Keyword(kw) => match kw {
78                            Keyword::ELSE => return None,
79                            _ => (),
80                        },
81                        PreAst::Opr(right_opr) => match right_opr {
82                            Opr::Prefix(_) => (), // actually this will be a syntax error
83                            Opr::Binary(right_opr) => {
84                                /// every binary opr in our small language is left associative, so `<` instead
        of `<=`
85                                if right_opr.precedence() > opr.precedence() {
86                                    return None;
87                                }
88                            }
89                            Opr::Suffix(right_opr) => {
90                                if right_opr.precedence() >= opr.precedence() {
91                                    return None;
92                                }
93                            }
94                        },
95                        // function call or index takes higher precedence
96                        PreAst::LeftDelimiter(_) => return None,
97                        PreAst::RightDelimiter(_) => (),
98                        PreAst::Ast(_) => (),
99                        PreAst::Separator(_) => (),
100                   }
101               };
102               Some(AstData::Binary { lopd, opr, ropd })
103           }
104           Opr::Suffix(opr) => {
105               let Some((opd, PreAst::Ast(_))) = nearest_left2.first() else {
106                   return None;
107               };
108               if let Some((_, ast)) = nearest_left2.second() {
109                   match ast {
110                       PreAst::Keyword(_) => (),
111                       PreAst::Opr(right_opr) => match right_opr {
112                           Opr::Prefix(right_opr) => {
113                               if right_opr.precedence() > opr.precedence() {
114                                   return None;
115                               }
116                           }
117                           Opr::Binary(right_opr) => {
118                               /// every binary opr in our small language is left associative, so `<` instead
        of `<=`
119                               if right_opr.precedence() > opr.precedence() {
120                                   return None;
121                               }
122                           }
123                           Opr::Suffix(_) => (),
124                       },
```

```
125                    PreAst::LeftDelimiter(_) => (),
126                    PreAst::RightDelimiter(_) => return None,
127                    PreAst::Ast(_) => return None,
128                    PreAst::Separator(_) => (),
129                }
130            };
131            Some(AstData::Suffix { opr, opd })
132        }
133    }
134 }
```

```
1  /// returns sequence of remaining PreAsts and new parent idxs
2  pub(crate) fn reduce_pre_asts_by_opr(
3      pre_asts: Seq<Option<PreAst>>,
4      new_asts: Seq<Option<AstData>>,
5  ) -> (Seq<Option<PreAst>>, Seq<Option<Idx>>) {
6      let new_asts_nearest_left = new_asts.nearest_left();
7      let pre_asts = reduce_pre_ast_by_new_ast.apply(pre_asts, new_asts);
8      let (pre_asts, new_parents) = reduce_pre_ast_by_opr_left
9          .apply_enumerated(new_asts_nearest_left, pre_asts)
10         .decouple();
11     let new_asts_nearest_right = new_asts.nearest_right();
12     reduce_pre_ast_by_opr_right
13         .apply_enumerated(new_asts_nearest_right, pre_asts, new_parents)
14         .decouple()
15 }
```

```
1  fn reduce_pre_ast_by_new_ast(pre_ast: Option<PreAst>, new_ast: Option<AstData>) -> Option<PreAst> {
2      if new_ast.is_some() {
3          None
4      } else {
5          pre_ast
6      }
7  }
```

```
1  fn reduce_pre_ast_by_opr_left(
2      idx: Idx,
3      new_ast_nearest_left: Option<(Idx, AstData)>,
4      pre_ast: Option<PreAst>,
5  ) -> (Option<PreAst>, Option<Idx>) {
6      let Some(pre_ast) = pre_ast else {
7          return (None, None);
8      };
9      let Some((new_ast_idx, new_ast_data)) = new_ast_nearest_left else {
10         return (Some(pre_ast), None);
11     };
12     match new_ast_data {
13         AstData::Binary { ropd: opd, .. } | AstData::Prefix { opd, .. } if opd == idx => {
14             (None, Some(new_ast_idx))
15         }
16         _ => (Some(pre_ast), None),
17     }
18 }
```

```
1  fn reduce_pre_ast_by_opr_right(
2      idx: Idx,
3      new_ast_nearest_right: Option<(Idx, AstData)>,
4      pre_ast: Option<PreAst>,
5      new_parent: Option<Idx>,
6  ) -> (Option<PreAst>, Option<Idx>) {
7      let Some(pre_ast) = pre_ast else {
8          return (None, new_parent);
9      };
10     if let Some(new_parent) = new_parent {
11         return (None, Some(new_parent));
12     }
13     let Some((new_ast_idx, new_ast_data)) = new_ast_nearest_right else {
14         return (Some(pre_ast), None);
15     };
16     match new_ast_data {
17         AstData::Binary { lopd: opd, .. } | AstData::Suffix { opd, .. } if opd == idx => {
18             (None, Some(new_ast_idx))
19         }
20         _ => (Some(pre_ast), None),
21     }
22 }
```

## F.3 Statements

In this section, we lay down the definition of `reduce_by_stmt`.

```
1  pub(super) fn reduce_by_stmt(
2      pre_asts: Seq<Option<PreAst>>,
3      allocated_asts: Seq<Option<Ast>>,
4  ) -> (Seq<Option<PreAst>>, Seq<Option<Ast>>) {
5      let pre_asts_nearest_left2 = pre_asts.nearest_left2();
6      let pre_asts_nearest_right2 = pre_asts.nearest_right2();
7      let new_stmt_asts =
8          new_stmt_ast.apply(pre_asts_nearest_left2, pre_asts, pre_asts_nearest_right2);
9      let (pre_asts, new_parents) = reduce_pre_asts_by_stmt(pre_asts, new_stmt_asts);
10     let allocated_asts =
11         allocate_asts_and_update_parents(allocated_asts, new_stmt_asts, new_parents);
12     let pre_asts = update_pre_asts_by_new_asts(pre_asts, new_stmt_asts);
13     (pre_asts, allocated_asts)
14 }
```

```
1  fn new_stmt_ast(
2      pre_ast_nearest_left2: Option2<(Idx, PreAst)>,
3      pre_ast: Option<PreAst>,
4      pre_ast_nearest_right2: Option2<(Idx, PreAst)>,
5  ) -> Option<AstData> {
6      let PreAst::Keyword(Keyword::Stmt(kw)) = pre_ast? else {
7          return None;
8      };
9      match kw {
10         StmtKeyword::Let => {
11             let Some((idx1, PreAst::Ast(ast))) = pre_ast_nearest_right2.first() else {
12                 return None;
13             };
14             if let Some((_, pre_ast)) = pre_ast_nearest_right2.second() {
15                 match pre_ast {
16                     PreAst::Keyword(_) => (),
17                     PreAst::Opr(_) | PreAst::LeftDelimiter(_) => return None,
18                     PreAst::RightDelimiter(_) => (),
19                     PreAst::Ast(_) => return None,
20                     PreAst::Separator(separator) => match separator {
21                         Separator::Comma => return None,
22                         Separator::Semicolon => (),
23                     },
24                 }
25             }
26             let (pattern, initial_value) = match ast {
27                 AstData::Binary {
28                     lopd,
29                     opr: BinaryOpr::Assign,
30                     ropd,
31                 } => (lopd, Some(ropd)),
32                 AstData::Ident(_)
33                 | AstData::Prefix { .. }
34                 | AstData::Binary { .. }
35                 | AstData::Delimited { .. }
36                 | AstData::Call { .. } => (idx1, None),
37                 _ => return None,
38             };
39             Some(AstData::LetInit {
40                 expr: idx1,
41                 pattern,
42                 initial_value,
43             })
44         }
45         StmtKeyword::If => {
46             let Some((condition, PreAst::Ast(ast1))) = pre_ast_nearest_right2.first() else {
47                 return None;
48             };
49             let Some((
50                 body,
51                 PreAst::Ast(AstData::Delimited {
52                     left_delimiter: LCURL,
53                     right_delimiter: RCURL,
54                     ..
55                 }),
56             )) = pre_ast_nearest_right2.second()
57             else {
58                 return None;
59             };
60             Some(AstData::If { condition, body })
```

```
61              }
62          StmtKeyword::Else => {
63              let Some((if_stmt, PreAst::Ast(AstData::If { .. }))) = pre_ast_nearest_left2.first()
64              else {
65                  return None;
66              };
67              let Some((
68                  body,
69                  PreAst::Ast(
70                      AstData::Delimited {
71                          left_delimiter: LCURL,
72                          right_delimiter: RCURL,
73                          ..
74                      }
75                      | AstData::If { .. }
76                      | AstData::Else { .. },
77                  ),
78              )) = pre_ast_nearest_right2.first()
79              else {
80                  return None;
81              };
82              if let Some((_, PreAst::Keyword(Keyword::ELSE))) = pre_ast_nearest_right2.second() {
83                  return None;
84              }
85              Some(AstData::Else { if_stmt, body })
86          }
87      }
88  }
```

```
1   fn reduce_pre_asts_by_stmt(
2       pre_asts: Seq<Option<PreAst>>,
3       new_asts: Seq<Option<AstData>>,
4   ) -> (Seq<Option<PreAst>>, Seq<Option<Idx>>) {
5       let new_asts_nearest_left = new_asts.nearest_left();
6       let new_asts_nearest_right = new_asts.nearest_right();
7       reduce_pre_ast_by_stmt
8           .apply_enumerated(new_asts_nearest_left, new_asts_nearest_right, pre_asts)
9           .decouple()
10  }
```

```
1   fn reduce_pre_ast_by_stmt(
2       idx: Idx,
3       new_ast_nearest_left: Option<(Idx, AstData)>,
4       new_ast_nearest_right: Option<(Idx, AstData)>,
5       pre_ast: Option<PreAst>,
6   ) -> (Option<PreAst>, Option<Idx>) {
7       if let Some((idx1, ast)) = new_ast_nearest_left {
8           match ast {
9               AstData::LetInit { expr, .. } if expr == idx => (None, Some(idx1)),
10              AstData::If {
11                  condition, body, ..
12              } if condition == idx || body == idx => (None, Some(idx1)),
13              AstData::Else { body, .. } if body == idx => (None, Some(idx1)),
14              _ => (pre_ast, None),
15          }
16      } else if let Some((idx1, AstData::Else { if_stmt, .. })) = new_ast_nearest_right
17          && if_stmt == idx
18      {
19          (None, Some(idx1))
20      } else {
21          (pre_ast, None)
22      }
23  }
```

### F.4  Generalized Call Forms

In this section, we lay down the definition of `reduce_by_call`.

```
1   pub(super) fn reduce_by_call(
2       pre_asts: Seq<Option<PreAst>>,
3       allocated_asts: Seq<Option<Ast>>,
4   ) -> (Seq<Option<PreAst>>, Seq<Option<Ast>>) {
5       let pre_asts_nearest_left2 = pre_asts.nearest_left2();
6       let pre_asts_nearest_right = pre_asts.nearest_right();
7       let new_call_asts =
8           new_call_ast.apply_enumerated(pre_asts_nearest_left2, pre_asts_nearest_right);
9       let (pre_asts, new_parents) = reduce_pre_asts_by_call(pre_asts, new_call_asts);
```

```
10      let allocated_asts =
11          allocate_asts_and_update_parents(allocated_asts, new_call_asts, new_parents);
12      let pre_asts = update_pre_asts_by_new_asts(pre_asts, new_call_asts);
13      (pre_asts, allocated_asts)
14  }
```

```
 1  fn new_call_ast(
 2      idx: Idx,
 3      pre_ast_nearest_left2: Option2<(Idx, PreAst)>,
 4      pre_ast_nearest_right: Option<(Idx, PreAst)>,
 5  ) -> Option<AstData> {
 6      let (caller, PreAst::Ast(caller_ast)) = pre_ast_nearest_left2.first()? else {
 7          return None;
 8      };
 9      let (
10          delimited_arguments,
11          PreAst::Ast(AstData::Delimited {
12              left_delimiter_idx,
13              left_delimiter,
14              right_delimiter,
15          }),
16      ) = pre_ast_nearest_right?
17      else {
18          return None;
19      };
20      if let Some((_, snd)) = pre_ast_nearest_left2.second() {
21          match snd {
22              PreAst::Keyword(kw) => match kw {
23                  Keyword::Defn(kw) => match kw {
24                      DefnKeyword::Struct | DefnKeyword::Enum => return None,
25                      DefnKeyword::Fn => match left_delimiter.delimiter() {
26                          Delimiter::Parenthesis | Delimiter::Box => return None,
27                          Delimiter::Curly => (),
28                      },
29                  },
30                  Keyword::Stmt(kw) => match kw {
31                      StmtKeyword::Let => (),
32                      StmtKeyword::If => match left_delimiter.delimiter() {
33                          Delimiter::Parenthesis | Delimiter::Box => (),
34                          Delimiter::Curly => return None,
35                      },
36                      StmtKeyword::Else => return None,
37                  },
38              },
39              PreAst::Opr(opr) => match opr {
40                  Opr::Prefix(_) | Opr::Binary(_) => match left_delimiter.delimiter() {
41                      Delimiter::Parenthesis | Delimiter::Box => (),
42                      Delimiter::Curly => return None,
43                  },
44                  Opr::Suffix(_) => return None,
45              },
46              PreAst::LeftDelimiter(_) => (),
47              PreAst::RightDelimiter(_) => return None,
48              PreAst::Ast(snd_ast) => {
49                  if let AstData::Ident(_) = snd_ast
50                      && left_delimiter == LCURL
51                  {
52                      match caller_ast {
53                          AstData::Binary {
54                              opr: BinaryOpr::LightArrow,
55                              ..
56                          }
57                          | AstData::Delimited {
58                              left_delimiter: LPAR,
59                              right_delimiter: RPAR,
60                              ..
61                          } => (),
62                          _ => return None,
63                      }
64                  } else {
65                      return None;
66                  }
67              }
68              PreAst::Separator(_) => (),
69          }
70      }
71      if left_delimiter_idx != idx {
72          return None;
73      }
74      Some(AstData::Call {
75          caller,
```

```
76          delimited_arguments,
77          left_delimiter,
78          right_delimiter,
79      })
80  }
```

```
1  fn reduce_pre_asts_by_call(
2      pre_asts: Seq<Option<PreAst>>,
3      new_asts: Seq<Option<AstData>>,
4  ) -> (Seq<Option<PreAst>>, Seq<Option<Idx>>) {
5      let new_asts_nearest_left = new_asts.nearest_left();
6      let new_asts_nearest_right = new_asts.nearest_right();
7      reduce_pre_ast_by_call
8          .apply_enumerated(new_asts_nearest_left, new_asts_nearest_right, pre_asts)
9          .decouple()
10  }
```

```
1  fn reduce_pre_ast_by_call(
2      idx: Idx,
3      new_ast_nearest_left: Option<(Idx, AstData)>,
4      new_ast_nearest_right: Option<(Idx, AstData)>,
5      pre_ast: Option<PreAst>,
6  ) -> (Option<PreAst>, Option<Idx>) {
7      if let Some((
8          idx1,
9          AstData::Call {
10             delimited_arguments,
11             ..
12         },
13     )) = new_ast_nearest_left
14         && delimited_arguments == idx
15     {
16         (None, Some(idx1))
17     } else if let Some((idx1, AstData::Call { caller, .. })) = new_ast_nearest_right
18         && caller == idx
19     {
20         (None, Some(idx1))
21     } else {
22         (pre_ast, None)
23     }
24  }
```

## F.5 Definitions

In this section, we lay down the definition of `reduce_by_defn` .

```
1  pub(super) fn reduce_by_defn(
2      pre_asts: Seq<Option<PreAst>>,
3      allocated_asts: Seq<Option<Ast>>,
4  ) -> (Seq<Option<PreAst>>, Seq<Option<Ast>>) {
5      let pre_asts_nearest_left2 = pre_asts.nearest_left2();
6      let pre_asts_nearest_right2 = pre_asts.nearest_right2();
7      let new_defn_asts =
8          new_defn_ast.apply(pre_asts_nearest_left2, pre_asts, pre_asts_nearest_right2);
9      let (pre_asts, new_parents) = reduce_pre_asts_by_defn(pre_asts, new_defn_asts);
10     let allocated_asts =
11         allocate_asts_and_update_parents(allocated_asts, new_defn_asts, new_parents);
12     let pre_asts = update_pre_asts_by_new_asts(pre_asts, new_defn_asts);
13     (pre_asts, allocated_asts)
14  }
```

```
1  fn new_defn_ast(
2      pre_ast_nearest_left2: Option2<(Idx, PreAst)>,
3      pre_ast: Option<PreAst>,
4      pre_ast_nearest_right2: Option2<(Idx, PreAst)>,
5  ) -> Option<AstData> {
6      let PreAst::Keyword(Keyword::Defn(keyword)) = pre_ast? else {
7          return None;
8      };
9      {
10         let Some((ident_idx, PreAst::Ast(AstData::Ident(ident)))) = pre_ast_nearest_right2.first()
11         else {
12             return None;
13         };
14         let Some((content, PreAst::Ast(content_ast))) = pre_ast_nearest_right2.second() else {
15             return None;
16         };
```

```
17          match keyword {
18              DefnKeyword::Struct => match content_ast {
19                  AstData::Delimited { .. } => (),
20                  _ => return None,
21              },
22              DefnKeyword::Enum => match content_ast {
23                  AstData::Delimited { .. } => (),
24                  _ => return None,
25              },
26              DefnKeyword::Fn => match content_ast {
27                  AstData::Call { .. } => (),
28                  _ => return None,
29              },
30          }
31          Some(AstData::Defn {
32              keyword,
33              ident,
34              ident_idx,
35              content,
36          })
37      }
38  }
```

```
1  fn reduce_pre_asts_by_defn(
2      pre_asts: Seq<Option<PreAst>>,
3      new_asts: Seq<Option<AstData>>,
4  ) -> (Seq<Option<PreAst>>, Seq<Option<Idx>>) {
5      let new_asts_nearest_left = new_asts.nearest_left();
6      let new_asts_nearest_right = new_asts.nearest_right();
7      reduce_pre_ast_by_defn
8          .apply_enumerated(new_asts_nearest_left, new_asts_nearest_right, pre_asts)
9          .decouple()
10 }
```

```
1  fn reduce_pre_ast_by_defn(
2      idx: Idx,
3      new_ast_nearest_left: Option<(Idx, AstData)>,
4      new_ast_nearest_right: Option<(Idx, AstData)>,
5      pre_ast: Option<PreAst>,
6  ) -> (Option<PreAst>, Option<Idx>) {
7      if let Some((idx1, ast)) = new_ast_nearest_left {
8          match ast {
9              AstData::Defn {
10                 keyword,
11                 ident_idx,
12                 ident,
13                 content,
14                 ..
15             } if ident_idx == idx || content == idx => (None, Some(idx1)),
16             _ => (pre_ast, None),
17         }
18     } else if let Some((idx1, AstData::Defn { .. })) = new_ast_nearest_right
19         && false
20     {
21         (None, Some(idx1))
22     } else {
23         (pre_ast, None)
24     }
25 }
```

# G  Transformer Symbol Resolution Proof

Here we lay down the code for symbol resolution. The actual process involves many details such as computing ranks (the exact position of an AST node among its siblings), scopes, and roles (a more precise version of AST, computed from its parent recursively), definitions and resolutions.

*Proof Sketch of Theorem 2.* First, we need to define the type for scopes. It is represented by a tiny sequence of indices of curly brace block AST that enclose the type/function/variable. We assign the scope by walking through the ASTs in a top-down manner. We not only assign scopes to item definitions, we also: (1) assign scopes to ASTs representing curly brace blocks, with these scopes equal to the scope of block itself, and (2) assign scopes to identifiers waiting to be resolved, with these scopes equal to the maximum possible

scope of its resolved definition. The computation process is easily represented in Cybertron, indicating attention is expressive enough for this calculation and it only takes $O(D)$ number of layers.

After obtaining all the scopes for all items, it takes only one additional layer to obtain the symbolic resolution through attention. As attention is expressed through the dot product of two linear projections $Q$ and $K$, we have to choose the representation of the scope type properly to finish the proof. The full details are in Appendix G. □

## G.1 Ranks

```
1  #[derive(Debug, Default, PartialEq, Eq, Clone, Copy)]
2  pub struct Rank(u8);
3
4  impl Rank {
5      fn next(self) -> Self {
6          Self(self.0 + 1)
7      }
8  }
9
10 pub fn calc_ranks(asts: Seq<Option<Ast>>) -> Seq<Option<Rank>> {
11     let counts = asts.count_past_by_attention(asts, |ast, ast1| {
12         let Some(ast) = ast else { return false };
13         let Some(ast1) = ast1 else { return false };
14         ast.parent == ast1.parent
15     });
16     (|c: usize, ast| {
17         ast?;
18         Some(Rank(c.try_into().unwrap()))
19     })
20     .apply(counts, asts)
21 }
22
23 pub fn calc_ranks1(asts: Seq<Option<Ast>>, n: usize) -> Seq<Option<Rank>> {
24     let mut ranks: Seq<Option<Rank>> = asts.map(|_| None);
25     for _ in 0..n {
26         ranks = calc_sibling_indicies_step(asts, ranks);
27     }
28     ranks
29 }
30
31 fn calc_sibling_indicies_step(
32     asts: Seq<Option<Ast>>,
33     ranks: Seq<Option<Rank>>,
34 ) -> Seq<Option<Rank>> {
35     let previous_ranks = ranks.nearest_left_filtered_by_attention(asts, asts, |ast, ast1| {
36         let Some(ast) = ast else { return false };
37         let Some(ast1) = ast1 else { return false };
38         ast.parent == ast1.parent
39     });
40     let ranks = (|ast, rank, previous_rank: Option<Option<Rank>>| {
41         let _ = ast?;
42         if let Some(rank) = rank {
43             return Some(rank);
44         }
45         let Some(previous_rank) = previous_rank else {
46             return Some(Default::default());
47         };
48         Some(previous_rank?.next())
49     })
50     .apply(asts, ranks, previous_ranks);
51     ranks
52 }
```

In the above, `count_past_by_attention` that count is representable by transformers by utilizing directly hard attention and the starter token. If the count is $c$, we shall get $c/(c+1)$ from the attention directly.

## G.2 Scopes

```
1  const D: usize = 8usize;
2
3  pub struct Scope {
4      enclosing_blocks: BoundedVec<Idx, D>,
```

```
 5  }
 6
 7  impl Scope {
 8      pub fn from_ast(idx: Idx, ast: AstData, parent_scope: Scope) -> Self {
 9          match ast {
10              AstData::Delimited {
11                  left_delimiter_idx,
12                  left_delimiter: LCURL,
13                  right_delimiter: RCURL,
14              } => Self {
15                  enclosing_blocks: parent_scope.enclosing_blocks.append(idx),
16              },
17              _ => parent_scope,
18          }
19      }
20
21      pub fn new(idx: Idx) -> Self {
22          Self {
23              enclosing_blocks: todo!(),
24          }
25      }
26
27      pub fn append(self, idx: Idx) -> Self {
28          Self {
29              enclosing_blocks: self.enclosing_blocks.append(idx),
30          }
31      }
32  }
33
34  impl Scope {
35      pub fn contains(self, other: Self) -> bool {
36          let len = self.enclosing_blocks.len();
37          if len > other.enclosing_blocks.len() {
38              return false;
39          }
40          for i in 0..len {
41              if self.enclosing_blocks[i] != other.enclosing_blocks[i] {
42                  return false;
43              }
44          }
45          true
46      }
47  }
48
49  pub fn infer_scopes(asts: Seq<Option<Ast>>, n: usize) -> Seq<Option<Scope>> {
50      let mut scopes = initial_scope.apply_enumerated(asts);
51      for _ in 0..n {
52          let parent_scopes = parent_queries(asts, scopes);
53          scopes = infer_scopes_step(asts, parent_scopes, scopes);
54      }
55      scopes
56  }
57
58  fn initial_scope(idx: Idx, ast: Option<Ast>) -> Option<Scope> {
59      let ast = ast?;
60      if ast.parent.is_some() {
61          return None;
62      }
63      let scope = Scope::default();
64      Some(Scope::from_ast(idx, ast.data, scope))
65  }
66
67  fn infer_scopes_step(
68      asts: Seq<Option<Ast>>,
69      parent_scopes: Seq<Option<Scope>>,
70      scopes: Seq<Option<Scope>>,
71  ) -> Seq<Option<Scope>> {
72      infer_scope_step.apply_enumerated(asts, parent_scopes, scopes)
73  }
74
75  fn infer_scope_step(
76      idx: Idx,
77      ast: Option<Ast>,
78      parent_scope: Option<Scope>,
79      scope: Option<Scope>,
80  ) -> Option<Scope> {
81      if let Some(scope) = scope {
82          return Some(scope);
83      }
84      Some(Scope::from_ast(idx, ast?.data, parent_scope?))
85  }
```

## G.3 Roles

```rust
#[derive(Debug, Clone, Copy, PartialEq, Eq)]
pub enum Role {
    LetStmt {
        pattern: Idx,
        initial_value: Option<Idx>,
    },
    LetStmtInner {
        pattern: Idx,
        initial_value: Idx,
    },
    LetStmtIdent,
    LetStmtTypedVariables {
        variables: Idx,
        ty: Idx,
    },
    StructDefn(Ident),
    EnumDefn(Ident),
    FnDefn(Ident),
    FnDefnCallForm {
        fn_ident: Ident,
        scope: Scope,
    },
    FnParameters {
        fn_ident: Ident,
        has_return_ty: bool,
        scope: Scope,
    },
    FnParametersAndReturnType {
        fn_ident: Ident,
        parameters: Idx,
        scope: Scope,
        return_ty: Idx,
    },
    FnBody(Ident),
    StructFields(Ident),
    FnParameter {
        fn_ident: Ident,
        rank: Rank,
        ty: Idx,
        fn_ident_idx: Idx,
        scope: Scope,
    },
    FnParameterIdent {
        scope: Scope,
    },
    FnParameterSeparated {
        fn_ident: Ident,
        rank: Rank,
        scope: Scope,
    },
    FnParameterType {
        fn_ident: Ident,
        rank: Rank,
    },
    FnOutputType {
        fn_ident: Ident,
    },
    StructField {
        ty_ident: Ident,
        field_ident: Ident,
        ty_idx: Idx,
    },
    StructFieldType {
        ty_ident: Ident,
        field_ident: Ident,
    },
    TypeArgument,
    TypeArguments,
    StructFieldSeparated(Ident),
    LetStmtVariablesType,
    LetStmtVariables,
}
```

```rust
impl Ast {
    fn role(self) -> Option<Role> {
        match self.data {
            AstData::LetInit {
                expr,
```

```
 6                  pattern,
 7                  initial_value,
 8              } => Some(Role::LetStmt {
 9                  pattern,
10                  initial_value,
11              }),
12              AstData::Defn {
13                  keyword,
14                  ident_idx,
15                  ident,
16                  content,
17              } => Some(match keyword {
18                  DefnKeyword::Struct => Role::StructDefn(ident),
19                  DefnKeyword::Enum => Role::EnumDefn(ident),
20                  DefnKeyword::Fn => Role::FnDefn(ident),
21              }),
22              _ => None,
23          }
24      }
25  }
```

```
 1  pub fn calc_roles(
 2      asts: Seq<Option<Ast>>,
 3      scopes: Seq<Option<Scope>>,
 4      n: usize,
 5  ) -> Seq<Option<Role>> {
 6      let mut roles: Seq<Option<Role>> = asts.map(|ast| ast?.role());
 7      let ranks = calc_ranks(asts);
 8      for _ in 0..n {
 9          let parent_roles = parent_queries(asts, roles);
10          roles = calc_roles_step(asts, parent_roles, roles, ranks, scopes);
11      }
12      roles
13  }
```

```
 1  fn calc_roles_step(
 2      asts: Seq<Option<Ast>>,
 3      parent_roles: Seq<Option<Role>>,
 4      roles: Seq<Option<Role>>,
 5      ranks: Seq<Option<Rank>>,
 6      scopes: Seq<Option<Scope>>,
 7  ) -> Seq<Option<Role>> {
 8      calc_role_step.apply_enumerated(asts, parent_roles, roles, ranks, scopes)
 9  }
```

```
 1  fn calc_role_step(
 2      idx: Idx,
 3      ast: Option<Ast>,
 4      parent_role: Option<Role>,
 5      role: Option<Role>,
 6      rank: Option<Rank>,
 7      scope: Option<Scope>,
 8  ) -> Option<Role> {
 9      if let Some(role) = role {
10          return Some(role);
11      }
12      let ast = ast?;
13      if let Some(role) = ast.role() {
14          return Some(role);
15      }
16      match parent_role? {
17          Role::LetStmt {
18              pattern,
19              initial_value,
20          } => match ast.data {
21              AstData::Ident(ident) if idx == pattern => Some(Role::LetStmtIdent),
22              AstData::Binary {
23                  lopd,
24                  opr: BinaryOpr::Assign,
25                  ropd,
26                  lopd_ident,
27              } if lopd == pattern => Some(Role::LetStmtInner {
28                  pattern,
29                  initial_value: ropd,
30              }),
31              _ => None,
32          },
33          Role::LetStmtInner {
34              pattern,
35              initial_value,
```

```
 36              } => {
 37                  if idx == pattern {
 38                      match ast.data {
 39                          AstData::Ident(ident) => Some(Role::LetStmtIdent),
 40                          AstData::Binary {
 41                              lopd,
 42                              lopd_ident,
 43                              opr,
 44                              ropd,
 45                          } => Some(Role::LetStmtTypedVariables {
 46                              variables: lopd,
 47                              ty: ropd,
 48                          }),
 49                          _ => todo!(),
 50                      }
 51                  } else {
 52                      None
 53                  }
 54              }
 55          Role::LetStmtIdent => todo!(),
 56          Role::FnParameterIdent { scope } => todo!(),
 57          Role::StructDefn(ident) => match ast.data {
 58              AstData::Literal(_) => todo!(),
 59              AstData::Ident(_) => None,
 60              AstData::Prefix { opr, opd } => todo!(),
 61              AstData::Binary {
 62                  lopd,
 63                  opr,
 64                  ropd,
 65                  lopd_ident,
 66              } => todo!(),
 67              AstData::Suffix { opd, opr } => todo!(),
 68              AstData::Delimited {
 69                  left_delimiter_idx,
 70                  left_delimiter,
 71                  right_delimiter,
 72              } => Some(Role::StructFields(ident)),
 73              AstData::SeparatedItem { content, separator } => todo!(),
 74              AstData::Call { .. } => todo!(),
 75              AstData::LetInit {
 76                  expr,
 77                  pattern,
 78                  initial_value,
 79              } => todo!(),
 80              AstData::Return { result } => todo!(),
 81              AstData::Assert { condition } => todo!(),
 82              AstData::If { condition, body } => todo!(),
 83              AstData::Else { if_stmt, body } => todo!(),
 84              AstData::Defn {
 85                  keyword,
 86                  ident_idx,
 87                  ident,
 88                  content,
 89              } => todo!(),
 90          },
 91          Role::EnumDefn(_) => None, // ad hoc
 92          Role::FnDefn(fn_ident) => match ast.data {
 93              AstData::Literal(_) => todo!(),
 94              AstData::Ident(_) => None,
 95              AstData::Prefix { opr, opd } => todo!(),
 96              AstData::Binary {
 97                  lopd,
 98                  opr,
 99                  ropd,
100                  lopd_ident,
101              } => todo!(),
102              AstData::Suffix { opd, opr } => todo!(),
103              AstData::Delimited {
104                  left_delimiter_idx,
105                  left_delimiter,
106                  right_delimiter,
107              } => todo!(),
108              AstData::SeparatedItem { content, separator } => todo!(),
109              AstData::Call {
110                  delimited_arguments,
111                  ..
112              } => Some(Role::FnDefnCallForm {
113                  fn_ident,
114                  scope: match scope {
115                      Some(scope) => scope.append(delimited_arguments),
116                      None => Scope::new(delimited_arguments),
```

```
117                },
118            }),
119            AstData::LetInit {
120                expr,
121                pattern,
122                initial_value,
123            } => todo!(),
124            AstData::Return { result } => todo!(),
125            AstData::Assert { condition } => todo!(),
126            AstData::If { condition, body } => todo!(),
127            AstData::Else { if_stmt, body } => todo!(),
128            AstData::Defn {
129                keyword,
130                ident_idx,
131                ident,
132                content,
133            } => todo!(),
134        },
135        Role::FnDefnCallForm { fn_ident, scope } => match ast.data {
136            AstData::Literal(_) => todo!(),
137            AstData::Ident(_) => todo!(),
138            AstData::Prefix { opr, opd } => todo!(),
139            AstData::Binary {
140                lopd,
141                opr,
142                ropd,
143                lopd_ident,
144            } => {
145                if opr == BinaryOpr::LightArrow {
146                    Some(Role::FnParametersAndReturnType {
147                        fn_ident,
148                        parameters: lopd,
149                        return_ty: ropd,
150                        scope,
151                    })
152                } else {
153                    unreachable!()
154                }
155            }
156            AstData::Suffix { opd, opr } => todo!(),
157            AstData::Delimited {
158                left_delimiter_idx,
159                left_delimiter,
160                right_delimiter,
161            } => match left_delimiter.delimiter() {
162                Delimiter::Parenthesis => Some(Role::FnParameters {
163                    fn_ident,
164                    has_return_ty: false,
165                    scope,
166                }),
167                Delimiter::Box => todo!(),
168                Delimiter::Curly => Some(Role::FnBody(fn_ident)),
169            },
170            AstData::SeparatedItem { content, separator } => todo!(),
171            AstData::Call { .. } => todo!(),
172            AstData::LetInit {
173                expr,
174                pattern,
175                initial_value,
176            } => todo!(),
177            AstData::Return { result } => todo!(),
178            AstData::Assert { condition } => todo!(),
179            AstData::If { condition, body } => todo!(),
180            AstData::Else { if_stmt, body } => todo!(),
181            AstData::Defn {
182                keyword,
183                ident_idx,
184                ident,
185                content,
186            } => todo!(),
187        },
188        Role::FnParameters {
189            fn_ident, scope, ..
190        } => match ast.data {
191            AstData::Binary {
192                lopd,
193                opr,
194                ropd,
195                lopd_ident,
196            } => {
197                if opr == BinaryOpr::TypeIs {
```

```
198                Some(Role::FnParameter {
199                    fn_ident,
200                    fn_ident_idx: lopd,
201                    rank: rank.unwrap(),
202                    ty: ropd,
203                    scope,
204                })
205            } else {
206                unreachable!()
207            }
208        }
209        AstData::SeparatedItem { .. } => Some(Role::FnParameterSeparated {
210            fn_ident,
211            rank: rank.unwrap(),
212            scope,
213        }),
214        _ => unreachable!(),
215    },
216    Role::FnBody(_) => None,
217    Role::StructFields(ty_ident) => match ast.data {
218        AstData::Binary {
219            lopd,
220            opr,
221            ropd,
222            lopd_ident,
223        } => {
224            assert_eq!(opr, BinaryOpr::TypeIs);
225            Some(Role::StructField {
226                ty_ident,
227                field_ident: lopd_ident.unwrap(),
228                ty_idx: ropd,
229            })
230        }
231        AstData::SeparatedItem { content, separator } => {
232            Some(Role::StructFieldSeparated(ty_ident))
233        }
234        _ => None,
235    },
236    Role::FnParameter {
237        fn_ident,
238        fn_ident_idx,
239        rank,
240        ty,
241        scope,
242        ..
243    } => {
244        if idx == ty {
245            Some(Role::FnParameterType { fn_ident, rank })
246        } else if idx == fn_ident_idx {
247            Some(Role::FnParameterIdent { scope })
248        } else {
249            None
250        }
251    }
252    Role::FnParameterSeparated {
253        fn_ident,
254        rank,
255        scope,
256    } => match ast.data {
257        AstData::Binary {
258            lopd,
259            opr,
260            ropd,
261            lopd_ident,
262        } => {
263            if opr == BinaryOpr::TypeIs {
264                Some(Role::FnParameter {
265                    fn_ident,
266                    fn_ident_idx: lopd,
267                    rank,
268                    ty: ropd,
269                    scope,
270                })
271            } else {
272                unreachable!()
273            }
274        }
275        _ => unreachable!(),
276    },
277    Role::StructField {
278        ty_ident,
```

```
279                field_ident,
280                ty_idx,
281            } => {
282                if idx == ty_idx {
283                    Some(Role::StructFieldType {
284                        ty_ident,
285                        field_ident,
286                    })
287                } else {
288                    None
289                }
290            }
291            Role::StructFieldSeparated(ty_ident) => match ast.data {
292                AstData::Binary {
293                    lopd,
294                    opr,
295                    ropd,
296                    lopd_ident,
297                } => {
298                    assert_eq!(opr, BinaryOpr::TypeIs);
299                    Some(Role::StructField {
300                        ty_ident,
301                        field_ident: lopd_ident.unwrap(),
302                        ty_idx: ropd,
303                    })
304                }
305                _ => unreachable!(),
306            },
307            Role::FnParameterType { .. } | Role::StructFieldType { .. } | Role::TypeArgument => {
308                match ast.data {
309                    AstData::Delimited {
310                        left_delimiter_idx,
311                        left_delimiter,
312                        right_delimiter,
313                    } => Some(Role::TypeArguments),
314                    _ => None,
315                }
316            }
317            Role::TypeArguments => match ast.data {
318                AstData::Ident(_) => Some(Role::TypeArgument),
319                AstData::Delimited {
320                    left_delimiter_idx,
321                    left_delimiter,
322                    right_delimiter,
323                } => todo!(),
324                AstData::SeparatedItem { content, separator } => todo!(),
325                AstData::Call {
326                    caller,
327                    caller_ident,
328                    left_delimiter,
329                    right_delimiter,
330                    delimited_arguments,
331                } => todo!(),
332                _ => None,
333            },
334            Role::FnParametersAndReturnType {
335                fn_ident,
336                parameters,
337                return_ty,
338                scope,
339            } => {
340                if idx == parameters {
341                    Some(Role::FnParameters {
342                        fn_ident,
343                        has_return_ty: true,
344                        scope,
345                    })
346                } else if idx == return_ty {
347                    Some(Role::FnOutputType { fn_ident })
348                } else {
349                    unreachable!()
350                }
351            }
352            Role::FnOutputType { fn_ident } => todo!(),
353            Role::LetStmtTypedVariables { variables, ty } => {
354                if idx == variables {
355                    Some(Role::LetStmtVariables)
356                } else if idx == ty {
357                    Some(Role::LetStmtVariablesType)
358                } else {
359                    unreachable!()
```

```
360              }
361          }
362          Role::LetStmtVariablesType => todo!(),
363          Role::LetStmtVariables => todo!(),
364      }
365  }
```

## G.4   Defns

```
1  #[derive(Debug, Clone, Copy, PartialEq, Eq)]
2  pub struct SymbolDefn {
3      pub symbol: Symbol,
4      pub scope: Option<Scope>,
5  }
```

```
1  pub fn calc_symbol_defns(
2      asts: Seq<Option<Ast>>,
3      scopes: Seq<Option<Scope>>,
4      n: usize,
5  ) -> Seq<Option<SymbolDefn>> {
6      let roles = calc_roles(asts, scopes, n);
7      calc_symbol_defn.apply_enumerated(asts, roles, scopes)
8  }
```

```
1   fn calc_symbol_defn(
2       idx: Idx,
3       ast: Option<Ast>,
4       role: Option<Role>,
5       scope: Option<Scope>,
6   ) -> Option<SymbolDefn> {
7       match ast?.data {
8           AstData::Ident(ident) => match role? {
9               Role::LetStmt { .. } => unreachable!(),
10              Role::LetStmtVariables | Role::LetStmtIdent => Some(SymbolDefn {
11                  symbol: Symbol {
12                      ident,
13                      source: idx,
14                      data: SymbolData::Variable,
15                  },
16                  scope,
17              }),
18              Role::FnParameterIdent { scope } => Some(SymbolDefn {
19                  symbol: Symbol {
20                      ident,
21                      source: idx,
22                      data: SymbolData::Variable,
23                  },
24                  scope: Some(scope),
25              }),
26              _ => None,
27          },
28          AstData::Defn {
29              keyword,
30              ident_idx,
31              ident,
32              content,
33          } => Some(SymbolDefn {
34              symbol: Symbol {
35                  ident,
36                  source: idx,
37                  data: SymbolData::Item {
38                      kind: keyword.into(),
39                  },
40              },
41              scope,
42          }),
43          _ => None,
44      }
45  }
```

## G.5   Resolutions

```
1  pub enum SymbolResolution {
2      Ok(Symbol),
3      Err(SymbolResolutionError),
4  }
```

```
1  pub enum SymbolResolutionError {
2      NotResolved,
3      NotYetDeclared(Symbol),
4  }
```

```
1  pub fn calc_symbol_resolutions(asts: Seq<Option<Ast>>, n: usize) -> Seq<Option<SymbolResolution>> {
2      let scopes = infer_scopes(asts, n);
3      let symbol_defns = calc_symbol_defns(asts, scopes, n);
4      let idents = asts.map(|ast| match ast?.data {
5          AstData::Ident(ident) => Some(ident),
6          _ => None,
7      });
8      let symbols = symbol_defns
9          .map(|symbol_defn| Some(symbol_defn?.symbol))
10         .first_filtered_by_attention(
11             (|ident, scope| (ident, scope)).apply(idents, scopes),
12             symbol_defns,
13             |(ident, scope), symbol_defn| {
14                 let Some(ident) = ident else { return false };
15                 let Some(symbol_defn) = symbol_defn else {
16                     return false;
17                 };
18                 if let Some(symbol_defn_scope) = symbol_defn.scope {
19                     if !symbol_defn_scope.contains(scope.unwrap()) {
20                         return false;
21                     }
22                 }
23                 symbol_defn.symbol.ident == ident
24             },
25         )
26         .map(|s| s.flatten());
27      finalize.apply_enumerated(idents, symbols)
28  }
```

In the above code, we use a somehow complicated attention which we should illustrate why it's representable by transformers. The essence is to prove `symbol_defn_scope.contains(scope.unwrap())` can be represented as part of the inner product in $Q^\top K$. This can be done by looking closer to what `contains` does. Consider two scopes, `scope1` and `scope2`, which are sequences of bracket ast indices (can be null). The function returns true if the sequence of `scope1` contains the sequence of `scope2` as prefix, which can be achieved by $\sum_i x_i^\top y_i$ where $x_i, y_i$ are the encoding of $i$th ast indices of `scope1` and `scope2` after some transformations (different transformations because the function is asymmetric) so that $x_i^\top y_i = 0$ if and only if either $x_i$ is a None or $x_i$ represents the same thing as $y_i$, and $x_i^\top y_i < 0$ otherwise. More concretely, if $x_i$ is a None, $x_i = \mathbf{0}$ by choice, and equal to $(1, u_i)$ otherwise where $u_i$ corresponds to the encoding of the $i$th ast index of `scope1`; if $y_i$ is a None, $y_i = \mathbf{0}$ by choice, and equal to $(-1, v_i)$ otherwise where $A > 0$ and $v_i$ corresponds to the encoding of the $i$th ast index of `scope2`. We should choose the encoding $u_i, v_i$ such that $u_i^\top v_i = 1$ if and only if they encode the same index, which is obviously easy enough.

```
1  fn finalize(idx: Idx, ident: Option<Ident>, symbol: Option<Symbol>) -> Option<SymbolResolution> {
2      let _ = ident?;
3      let Some(symbol) = symbol else {
4          return Some(SymbolResolution::Err(SymbolResolutionError::NotResolved));
5      };
6      match symbol.data {
7          SymbolData::Item { .. } => (),
8          SymbolData::Variable => {
9              if idx < symbol.source {
10                 return Some(SymbolResolution::Err(
11                     SymbolResolutionError::NotYetDeclared(symbol),
12                 ));
13             }
14         }
15     }
16     Some(SymbolResolution::Ok(symbol))
17  }
```

# H   Transformer Type Checking Proof

Here we lay down the code for type analysis. It should be noted that we didn't completely implement all the details. Things like struct fields, enum variant fields are left out. However, we already cover the essential mechanism of type analysis, making it sufficient for proof purposes.

## H.1   Type Signatures

```
1  #[deri
2  ve(Debug, PartialEq, Eq, Clone, Copy)]
3  pub struct TypeSignature {
4      pub key: TypeSignatureKey,
5      pub ty: Type,
6  }
```

```
1  #[derive(Debug, PartialEq, Eq, Clone, Copy)]
2  pub enum TypeSignatureKey {
3      FnParameter { fn_ident: Ident, rank: Rank },
4      FnOutput { fn_ident: Ident },
5      StructField { ty_ident: Ident, field_ident: Ident },
6  }
```

```
1  pub(super) fn calc_ty_signatures(
2      asts: Seq<Option<Ast>>,
3      roles: Seq<Option<Role>>,
4      ty_terms: Seq<Option<Type>>,
5  ) -> Seq<Option<TypeSignature>> {
6      calc_ty_signature.apply(roles, ty_terms)
7  }
```

```
1   fn calc_ty_signature(role: Option<Role>, ty_term: Option<Type>) -> Option<TypeSignature> {
2       let key = match role? {
3           Role::FnParameterType { fn_ident, rank } => {
4               TypeSignatureKey::FnParameter { fn_ident, rank }
5           }
6           Role::StructFieldType {
7               ty_ident,
8               field_ident,
9           } => TypeSignatureKey::StructField {
10              ty_ident,
11              field_ident,
12          },
13          Role::FnOutputType { fn_ident } => TypeSignatureKey::FnOutput { fn_ident },
14          Role::FnParameters {
15              fn_ident,
16              has_return_ty: false,
17              scope,
18          } => {
19              let key = TypeSignatureKey::FnOutput { fn_ident };
20              let ty = Type::new_ident(Ident::new("unit"));
21              return Some(TypeSignature { key, ty });
22          }
23          _ => return None,
24      };
25      // put it here!
26      let ty = ty_term?;
27      Some(TypeSignature { key, ty })
28  }
```

## H.2   Type Inference

```
1  pub struct TypeInference {
2      pub ty: Type,
3  }
```

```
1  pub fn calc_ty_inferences(
2      asts: Seq<Option<Ast>>,
3      symbol_resolutions: Seq<Option<SymbolResolution>>,
4      roles: Seq<Option<Role>>,
5      ty_terms: Seq<Option<Type>>,
6      ty_signatures: Seq<Option<TypeSignature>>,
7      n: usize,
```

```
 8  ) -> Seq<Option<TypeInference>> {
 9      let mut ty_inferences = infer_tys_initial(asts, ty_signatures);
10      let mut ty_designations =
11          calc_initial_ty_designations(asts, roles, symbol_resolutions, ty_inferences, ty_terms);
12      for _ in 0..n {
13          ty_inferences |= infer_tys_step(asts, symbol_resolutions, ty_inferences, ty_designations);
14          ty_designations |= calc_ty_designations_step(roles, symbol_resolutions, ty_inferences);
15      }
16      ty_inferences
17  }
```

```
 1  fn infer_tys_initial(
 2      asts: Seq<Option<Ast>>,
 3      ty_signatures: Seq<Option<TypeSignature>>,
 4  ) -> Seq<Option<TypeInference>> {
 5      inference_literal_tys(asts).or(infer_fn_call_tys(asts, ty_signatures))
 6  }
```

```
 1  fn inference_literal_tys(asts: Seq<Option<Ast>>) -> Seq<Option<TypeInference>> {
 2      asts.map(|ast| match ast?.data {
 3          AstData::Literal(lit) => match lit {
 4              Literal::Int(_) => Some(TypeInference {
 5                  ty: Type::new_ident(Ident::new("Int")),
 6              }),
 7              Literal::Float(_) => Some(TypeInference {
 8                  ty: Type::new_ident(Ident::new("Float")),
 9              }),
10          },
11          _ => None,
12      })
13  }
```

```
 1  fn infer_fn_call_tys(
 2      asts: Seq<Option<Ast>>,
 3      ty_signatures: Seq<Option<TypeSignature>>,
 4  ) -> Seq<Option<TypeInference>> {
 5      ty_signatures
 6          .first_filtered_by_attention(asts, ty_signatures, |ast, ty_signature| {
 7              let Some(ast) = ast else { return false };
 8              let Some(TypeSignature {
 9                  key: TypeSignatureKey::FnOutput { fn_ident },
10                  ..
11              }) = ty_signature
12              else {
13                  return false;
14              };
15              match ast.data {
16                  AstData::Call {
17                      caller,
18                      caller_ident,
19                      left_delimiter,
20                      right_delimiter,
21                      delimited_arguments,
22                  } if caller_ident == Some(fn_ident) => true,
23                  _ => false,
24              }
25          })
26          .map(|ty_inference| {
27              Some(TypeInference {
28                  ty: ty_inference??.ty,
29              })
30          })
31  }
```

## H.3 Type Expectations

```
 1  pub struct TypeExpectation {
 2      pub ty: Type,
 3      pub source: TypeExpectationSource,
 4  }
```

```
 1  pub enum TypeExpectationSource {
 2      CallArgument { caller_ident: Ident, rank: Rank },
 3  }
```

```
1  pub fn calc_ty_expectations(
2      asts: Seq<Option<Ast>>,
3      ranks: Seq<Option<Rank>>,
4      ty_signatures: Seq<Option<TypeSignature>>,
5  ) -> Seq<Option<TypeExpectation>> {
6      let parent_asts = asts.index(asts.map(|ast| ast?.parent)).map(Option::flatten);
7      let grandparent_asts = asts
8          .index(parent_asts.map(|parent_ast| parent_ast?.parent))
9          .map(Option::flatten);
10     let ty_expectation_sources = calc_ty_expectation_source.apply(grandparent_asts, ranks);
11     let retrieved_ty_signatures = ty_signatures
12         .first_filtered_by_attention(
13             ty_expectation_sources,
14             ty_signatures,
15             |ty_expection_source, ty_signature| {
16                 let Some(type_expectation_source) = ty_expection_source else {
17                     return false;
18                 };
19                 let Some(type_signature) = ty_signature else {
20                     return false;
21                 };
22                 match (type_expectation_source, type_signature.key()) {
23                     (
24                         TypeExpectationSource::CallArgument {
25                             caller_ident,
26                             rank: rank0,
27                         },
28                         TypeSignatureKey::FnParameter {
29                             fn_ident,
30                             rank: rank1,
31                         },
32                     ) if caller_ident == fn_ident && rank0 == rank1 => true,
33                     _ => false,
34                 }
35             },
36         )
37         .map(Option::flatten);
38     (|ty_expectation_source: Option<TypeExpectationSource>,
39       retrieved_ty_signature: Option<TypeSignature>| {
40         Some(TypeExpectation {
41             ty: retrieved_ty_signature?.ty(),
42             source: ty_expectation_source?,
43         })
44     })
45     .apply(ty_expectation_sources, retrieved_ty_signatures)
46 }
```

```
1  fn calc_ty_expectation_source(
2      grandparent_ast: Option<Ast>,
3      rank: Option<Rank>,
4  ) -> Option<TypeExpectationSource> {
5      let grandparent_ast = grandparent_ast?;
6      let rank = rank?;
7      match grandparent_ast.data {
8          AstData::Call {
9              caller,
10             caller_ident: Some(caller_ident),
11             left_delimiter,
12             right_delimiter,
13             delimited_arguments,
14         } => Some(TypeExpectationSource::CallArgument { caller_ident, rank }),
15         _ => None,
16     }
17 }
```

## H.4 Type Errors

```
1  pub enum TypeError {
2      TypeMismatch { expected: Type, actual: Type },
3  }
```

```
1  pub fn calc_ty_errors(
2      ty_inferences: Seq<Option<TypeInference>>,
3      ty_expectations: Seq<Option<TypeExpectation>>,
4  ) -> Seq<Option<TypeError>> {
5      calc_ty_error.apply(ty_inferences, ty_expectations)
6  }
```

```
1  fn calc_ty_error(
2      ty_inference: Option<TypeInference>,
3      ty_expectation: Option<TypeExpectation>,
4  ) -> Option<TypeError> {
5      let ty_inference = ty_inference?;
6      let ty_expectation = ty_expectation?;
7      if ty_inference.ty == ty_expectation.ty {
8          None
9      } else {
10         Some(TypeError::TypeMismatch {
11             expected: ty_expectation.ty,
12             actual: ty_inference.ty,
13         })
14     }
15 }
```

# I Lower Bounds

```
1  struct <ty-ident-1> {}
2  struct <ty-ident-2> {}
3  struct <ty-ident-3> {}
4  struct <ty-ident-4> {}
5
6  fn <f-ident-1>(a: <arg-ty-ident-1>) {}
7  fn <f-ident-2>(a: <arg-ty-ident-2>) {}
8  fn <f-ident-3>(a: <arg-ty-ident-3>) {}
9  fn <f-ident-4>(a: <arg-ty-ident-4>) {}
10
11 fn g() {
12     let x: <ty-ident> = ...;
13     <f-ident>(x);
14 }
```

## I.1 Lower bounds for RNN: Easy Bounds due to Memory

*Proof of Theorem 4.* Our proof resonates with the proof of Theorem 4.6 in Wen et al. (2024) and Theorem 8 in Bhattamishra et al. (2024). For $L, D, H \in \mathbb{N}$, suppose that $D$ makes MiniHuskyAnnotated$_{D,H}$ to be nontrivial, i.e., one can define functions with one parameter and use function calls. Simple calculations shows we can choose $D = 7$ and $H = 1$. If a RNN represents a function maps any token sequence of length $L$ in MiniHuskyAnnotated$_{D,H}$ to its type errors represented as a sequence of values of type `Option<TypeError>`, then the memory right before type checking must store all previous type signatures, the number of which can be as many as $\Omega(L)$ in the worst case. Assuming proper numerical discretization, the memorization of these type signatures would require the memory size to be $\Omega(L)$ in the worst case. □

# J Additional Experiment Details

## J.1 An Example of Input Data

Below is a data piece with $f = 10, a = 5, c = 5, d = 3, v = 0.2, e = 0.5$:

```
1  fn rename_file ( i : Float , sum : Float ) { }
2  fn parse_data ( list : Int , value : Bool , stack : Float , k : Float , msg : Float ) { }
3  fn parse_json ( position : Bool ) { }
4  fn find_by_id ( error : Float ) { rename_file ( 60.1 , 94.1 ) ; }
5  fn merge ( group : Int , table : Float , error : Bool , count : Int ) { parse_data ( 7 , false , 49.1 , 33.1
       , 4.1 ) ; }
6  fn log_info ( val : Bool , m : Bool , xml : Float , path : Float ) { parse_json ( true ) ; }
7  fn process ( function : Int , value : Float , keys : Bool ) { find_by_id ( 88.1 ) ; rename_file ( value ,
       40.1 ) ; }
8  fn validate_response ( end : Int , z : Float , max : Bool ) { merge ( 1 , true , 27.1 , 72 ) ; parse_data (
       11 , 85 , 35.1 , 14.1 , true ) ; }
9  fn print_message ( algorithm : Float ) { parse_json ( 92 ) ; log_info ( true , algorithm , false , 26.1 ) ; }
10 fn print_help ( max : Bool , tree : Int , method : Int , item : Bool ) { process ( 25 , 28 , false ) ;
       rename_file ( 48 , 80.1 ) ; }
```

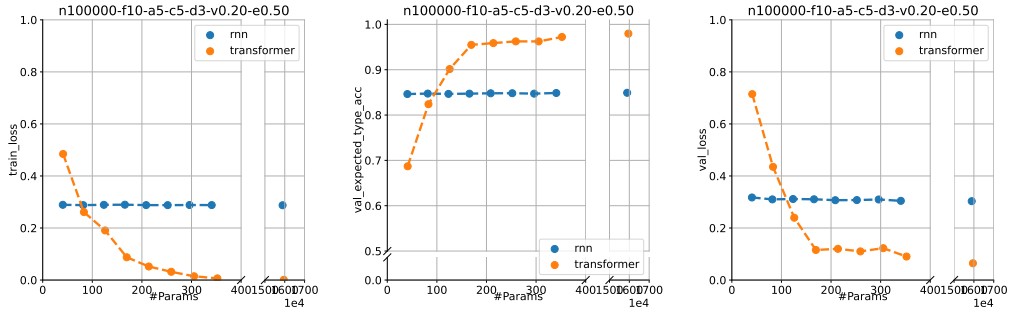

Figure 4: Figures for the dataset with $(f, a, c, d, v, e) = (10, 5, 5, 3, 0.2, 0.5)$.

## J.2  Setups

Model details are shown in Table 1, and other hyperparameters are shown in Table 2.

Table 1: Model specification

| Specification | Value |
|---|---|
| Transformer | |
| - Hidden size ($d_h$) | $\{8k \mid 1 \leq k \leq 8\} \cup \{240\}$ |
| - Num attention heads | 1 |
| - Num hidden layers | 8 |
| - Intermediate size | $2d_h$ |
| - Max position embeddings | $\leq 2048$ |
| RNN | |
| - Hidden size | $\{8k \mid 1 \leq k \leq 8\} \cup \{256\}$ |
| - Num layers | 8 |

Table 2: Hyperparameters of experiments

| Hyperparameter | Value |
|---|---|
| Dataset | |
| - $(n, f, d)$ | $\{(100000, 10, 3), (200000, 20, 5), (300000, 40, 10), (400000, 80, 20)\}$ |
| - $(a, c, v, e)$ | $(5, 5, 0.2, 0.5)$ |
| Number of epochs | 80 |
| Train batch size | 512 |
| Optimizer | Adam |
| LR scheduler | Linear warmup-decay |
| - Warmup min lr | $1 \times 10^{-5}$ |
| - Warmup max lr | $1 \times 10^{-3}$ |
| - Warmup steps | 990 |

## J.3  Additional Results

Figures 4,5,6,7 include other metrics (train loss, accuracies for expected type in validation set, and validation loss) in the experiments. Note that for the expressive power of the models, training accuracies are better indicators.

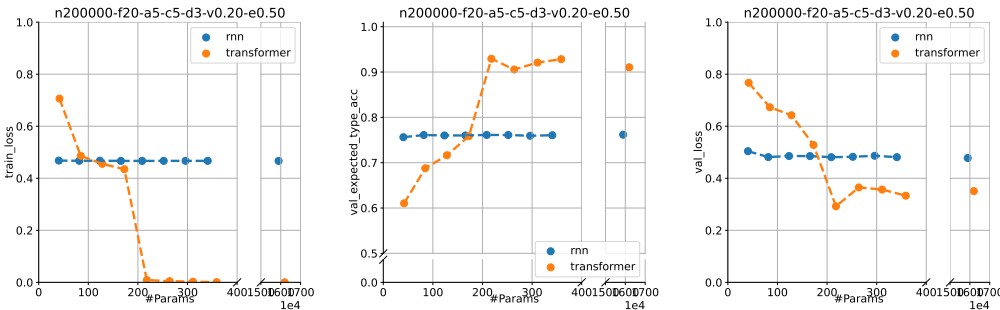

Figure 5: Figures for the dataset with $(f, a, c, d, v, e) = (20, 5, 5, 3, 0.2, 0.5)$.

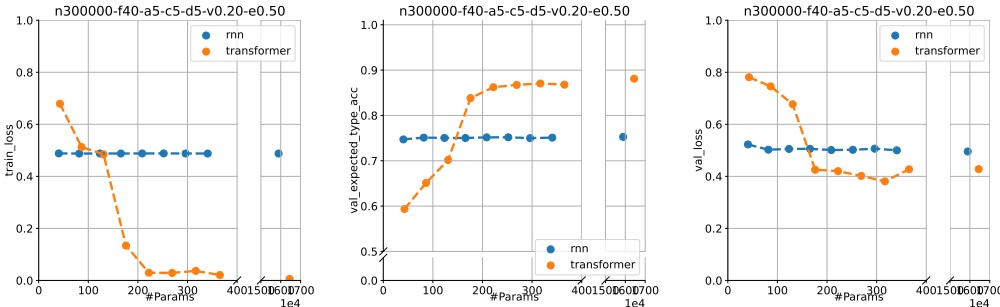

Figure 6: Figures for the dataset with $(f, a, c, d, v, e) = (40, 5, 5, 5, 0.2, 0.5)$.

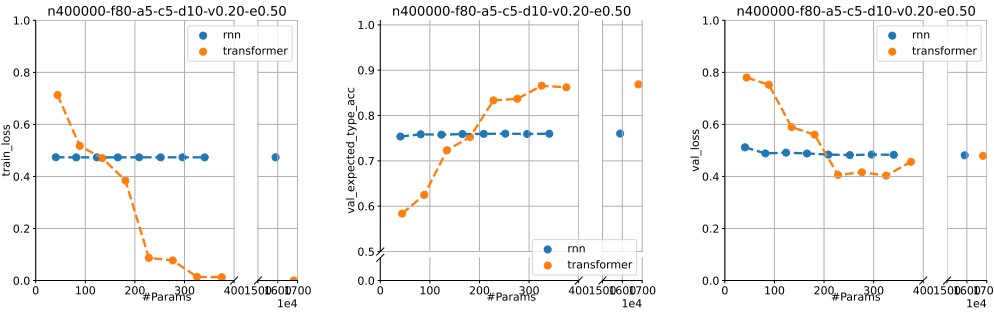

Figure 7: Figures for the dataset with $(f, a, c, d, v, e) = (80, 5, 5, 10, 0.2, 0.5)$.

