# OpenReview forum: "Transformers are Efficient Compilers, Provably"
_colmweb.org/COLM/2025/Conference — COLM 2025_

### Official Review · Reviewer_Nhkd · 2025-05-12

**Rating:** 5
**Confidence:** 5
**Ethics Flag:** 1

**Summary:**

This paper proves that transformer encoders can perform various compiler-related tasks on programs with bounded syntactic depth. The proofs mainly consist of programs written in a RASP-like language called Cybertron.

**Questions To Authors:**

I'd like to see some discussion of why we want neural networks to be compilers -- surely not to actually do compilation?

line 109-110: The citation to Reddit User, 2013 is really strange. For one thing, it was a post from 2024. Furthermore, it is cited as a "fine-grained characterization of the expressive power for certain functions in different settings," but it's not.

line 154: consider using the terminology "average hard attention" (Hao, et al., 2022). The citation to Hahn (2019) is perhaps misplaced, because he used a different kind of hard attention. Perez et al (2019) used average hard attention.

line 231: could you clarify whether the 'a' bound at line 6 is in scope between line 7 and 8?

At line 1440, can you please give an exhaustive list of attention methods? Is it just nearest_left and nearest_right, or are there others?

line 1355/eq 16: How does this select the nearest position? It looks like all unmasked positions p' where a_{flag,p'} is true have the same attention score, and since you are averaging over tied positions, they all get equal attention weight.

**Reasons To Accept:**

The main results of the paper are interesting, and the Cybertron language provides a higher-level and more convenient alternative to RASP.

**Reasons To Reject:**

The restriction to programs of bounded depth is critical, and the results on the compiler-related tasks are what one would expect, with the depth of the transformer depending linearly on the depth of the program.

Additionally, the results depend on an upper bound on the sequence length. These two bounds together seem to me to be rather limiting.

Cybertron is described only briefly in Section 5.4 of the paper; I would have preferred more details here. In the appendix, on the other hand, the description of Cybertron sprawls over 15 pages. A formal syntax, type system, and operational semantics would have been preferable. I did not understand some aspects of the language clearly and have some questions below, especially regarding the implementation of nearest_left using hard attention.

Most of the innovation of Cybertron as compared with RASP pertains to position-wise operations ("local world"), where it's obvious that arbitrary computations can be implemented in an FFNN. The part of the language that has the important connection with transformers is the attention methods, and here Cybertron looks basically the same as RASP (to the extent that it has been specified at all; this part of the language is left vague and open-ended).

In short, while Cybertron brings some high-level programming conveniences to RASP, I would say that it does not add to our understanding of transformers.

---

> ### Author Response · Authors · 2025-06-01
> **Respon to Reviewer Nhkd (2/2)**
>
> ### Questions To Authors:
>
> **[Q1]** I'd like to see some discussion of why we want neural networks to be compilers -- surely not to actually do compilation?
>
> [Reply] We apologize for the confusion. In fact, we don't intend to suggest that transformers should do the job of compilers. Rather, we use compilers as a formalizable task to argue about the code understanding capabilities of LLMs. Amazing coding powers of LLMs, including auto-completion, debugging, refactoring, and implementing algorithms, require basic code analysis capabilities, included in the frontend part of compilers.
>
> **[Q2]** line 109-110: The citation to Reddit User, 2013 is really strange. For one thing, it was a post from 2024. Furthermore, it is cited as a "fine-grained characterization of the expressive power for certain functions in different settings," but it's not.
>
>
> [Reply] Thank you very much for pointing out this error. We shall fix it.  It's a typo during our revision of the related work.
>
>
> **[Q3]** line 154: consider using the terminology "average hard attention" (Hao, et al., 2022). The citation to Hahn (2019) is perhaps misplaced, because he used a different kind of hard attention. Perez et al (2019) used average hard attention.
>
> [Reply] Thank you very much for pointing out. We will use "average hard attention".
>
> **[Q4]** line 231: could you clarify whether the 'a' bound at line 6 is in scope between line 7 and 8?
>
> [Reply] Yes, it is.
>
> **[Q5]** At line 1440, can you please give an exhaustive list of attention methods? Is it just nearest_left and nearest_right, or are there others?
>
> [Reply] It's hard to be exhaustive. We're not aware of an exhaustive way of characterizing all possible attentions. We expect users of Cybertron to prove their own version of attention-based methods if necessary. In this paper, we used `nearest_left`, `nearest_right`, `nearest_left2`, `nearest_right2`, `count_past_by_attention`, `nearest_left_filtered_by_attention`. For future generalizations, we might need more methods.
>
>
> **[Q6]** line 1355/eq 16: How does this select the nearest position? It looks like all unmasked positions p' where a_{flag,p'} is true have the same attention score, and since you are averaging over tied positions, they all get equal attention weight.
>
> [Reply] We apologize for the mistake. We missed an additional term $a(p'-p)$ for some proper constant $a$ that disrupts the equal attention weight. We thank the reviewer for pointing out the mistake.

---

> > ### Comment · Reviewer_Nhkd · 2025-06-02
> > **Disruptor term**
> >
> > Wouldn't $a(p'-p)$ fail if there is a "some" at position 1, a "none" at all other positions, and $p=1/a+3$? Then the score at position $p'=1$ would be $1+a(-1/a-2) = -2a$, while the score at position $p'=1/a+2$ would be $-a$ and would win. I think $1/(p-p'+1)$ would work better.

---

> > > ### Author Response · Authors · 2025-06-02
> > > **Addressing Disruptor Term**
> > >
> > > Thank you for your comment! We assume that the context length is bounded by say L, and a is taken to be way smaller than 1/L, so that it's a disruption everywhere. Then $p=1/a+3$ would be too large, not possible within the context.

---

> > > > ### Comment · Reviewer_Nhkd · 2025-06-02
> > > > **Maximum length**
> > > >
> > > > Are there other places where you assume a maximum length L?

---

> > > > > ### Author Response · Authors · 2025-06-04
> > > > > **L is stated in each individual theorem**
> > > > >
> > > > > L frequently appears in many of our arguments. For this particular case, L doesn't appear in the final theorem because we're using hard attention, and we can take a to be arbitrarily small. In other words, "hard attention" make it less obvious that things are depending on L.

---

> ### Author Response · Authors · 2025-06-01
> **Respon to Reviewer Nhkd (1/2)**
>
> We appreciate Reviewer Nhkd's informative comments.
> ### Reasons To Reject:
>
> **[R1]** The restriction to programs of bounded depth is critical, and the results on the compiler-related tasks are what one would expect, with the depth of the transformer depending linearly on the depth of the program.
>
> [Reply] The assumption is critical (also see discussions in As discus Yao et al. (2021)). The transformer architecture is about "finite-depth" computation. Theoretically speaking, transformers can only express functions in AC0. Empirically speaking, transformers without chain of thought tend to perform badly for more complicated math and coding tasks.
>
> From a practical viewpoint, bounded depth is fairly reasonable. The Linux kernel contains tens of millions of lines of code, however, most functions are pretty short. This is for better readability. For humans, deeply nested code is way harder to understand. The depth roughly speaking is upper bounded by the maximal length of a single function or type definition, which is usually bounded despite the growth of the codebase.
>
> **[R2]** Cybertron is described only briefly in Section 5.4 of the paper; I would have preferred more details here. In the appendix, on the other hand, the description of Cybertron sprawls over 15 pages. A formal syntax, type system, and operational semantics would have been preferable. I did not understand some aspects of the language clearly and have some questions below, especially regarding the implementation of nearest_left using hard attention.
>
> [Reply] Thanks for your comment. Please see our responses below to your specific questions.
>
> **[R3]** Most of the innovation of Cybertron as compared with RASP pertains to position-wise operations ("local world"), where it's obvious that arbitrary computations can be implemented in an FFNN. The part of the language that has the important connection with transformers is the attention methods, and here Cybertron looks basically the same as RASP.
>
> [Reply] The design of transformers is amazingly simple despite its powers. The high level intuitions provided by Cybertron and RASP are indeed quite similar if not identical. However, it's impossible to use RASP to prove the results in this paper due to the lack of essential features for dealing with more advanced tasks. Cybertron is designed to make the proof possible.
>
>
> In particular, RASP lacks features to describe complex types (struct and enum), making it impossible to represent things like ASTs and type signatures. RASP itself doesn't have type checking, unable to check correctness through the type system. Cybertron, as a subset of Rust, can describe complex types with their neural representation implied automatically and can provide type correctness checking for Cybertron code through the Rust compiler.

---

### Official Review · Reviewer_a8E2 · 2025-05-12

**Rating:** 8
**Confidence:** 4
**Ethics Flag:** 2

**Summary:**

This paper investigates whether Transformers can efficiently perform compilation from a theoretical and empirical POVs. The authors introduce a C/Rust-like simplified programming language called Mini-Husky as a playground to assess common compilation tasks with more control than in real-world programming languages. They evaluate analysis tasks: AST construction, symbol resolution, and type analysis. Authors show that, under bounded AST and type inference depth assumptions, Transformers' require logarithmic parameter scaling with input size, while RNNs require linear scaling. Cybertron, a new DSL, is also introduced to facilitate formal proofs of these capabilities. The paper also includes empirical results validating the findings. Using author's own words, without Cybertron, "writing an equivalent natural language proof would be too complex and intractable".

**Questions To Authors:**

- Missing citation of early work (2021) introducing neural compilation [1]. There's also [2] as a relevant missing citation.
- Will the code be open-sourced?
- Even if also used in previous work, can you comment on the potential limitations of using hard attention in your theoretical reasoning? Similarly, can you comment on using encoder-only Transformers?
- Small note: Compiler literature typically refers to "semantic analysis" rather than information (re Figure 1).
- sentence in line 195: "The raw text is firstly segmented into parts like literals, identifiers, punctuations,
 keywords, etc, called token stream, then parsed into a tree-like structure representation
generated from the input, finally syntactic and semantic analysis is performed on the tree." doesn't read well.
- Any particular reason to follow Rust-like keywords/structure/types?
- In Section 5.2, 5.3, the italics don't seem to play well with other fonts/symbols in the text.
- More writing notes: In Section 6: "Now we compare"... Perhaps "This section compares transformers and RNNs...". Immediately after, the sentence in 370 lacks a "been".
- I find the clean code principle usage very interesting as an assumption, particularly as measured by D, H << L. Could you expand on this? In particular, this assumption should be explicitly explained before, perhaps in the preliminaries.
- The setting of the appendix is quite uncommon. Appendices typically contain either additional results/experimental details in empirical papers, or additional proofs in theoretical papers. It's odd to see, for example, the explanation of a very basic concept in computer science such as trees in Appendix A. Appendix D.2 is also uncommon, it could be an interesting alternative introduction to the paper, or perhaps a blogpost about the paper. I'm not saying they need to be removed, but it's unusual. Then, there are some important concepts, perhaps critical to understand the paper, that due to space limitations need to be in the Appendix. As I said above, while in my view this paper should be accepted to COLM, perhaps it's not the optimal venue.
- 288: readility -> readability
- 379: "what RNNs requires” -> require
- Proofs would definitely benefit from more intuitive summaries or diagram. The only diagram in the paper (at least the main body) uses real estate that, given the tightness of the space requirements, perhaps should have been spent explaining concepts that are more difficult to understand?


[1] https://openreview.net/forum?id=444ug_EYXet
[2] https://openreview.net/forum?id=9S7osCL5Q8

**Reasons To Accept:**

- Novelty: The paper presents a novel theoretical result.
- Very strong contribution, both theoretically and in implementation (Contributes a new programming language and a new DSL to study LLM-based compilation).
- Empirical validation of the theory.

**Reasons To Reject:**

- The paper is not very well-motivated: The paper itself doesn't make a compelling case for LLM-based compilation.
- Not handling code generation: in compilation, generating code is at least as important as analyzing the input code. This approach is limited in that it only tackles half of the problem, while e.g. the title and abstract seem to imply the work tackles compilation in general.
- Lack of intuitive insights: The paper presents interesting devices (theoretical, + practical if we account for the actual implementation of its empirical results in Figure 2). However, after reading the paper, I find a lack of insights. A concrete result is the lower bound in 6.1, and the empirical results in line 403, corroborating that BERT models perform better at type checking. But there is no intuitive explanation on the implications of these results.
- The paper has a very long appendix, and it's not only complementary information. Some of it seems to be critical to understand more deeply the contribution of the paper. Perhaps this should have been a journal paper, rather than a conference paper? The paper itself seems to be sketching higher-level points while key points are offloaded to the appendix.

---

> ### Author Response · Authors · 2025-06-01
> **Respon to Reviewer a8E2 (2/2)**
>
> **[Q6]** Any particular reason to follow Rust-like keywords/structure/types?
>
> [Reply] Rust is arguably one of the most feature-rich languages. It incorporates both system and functional programming. Following Rust-like conventions makes it easy to be consistent. Cybertron is actually a subset of Rust. One can validate Cybertron's correctness by running tests in Rust.
>
> **[Q7]** In Section 5.2, 5.3, the italics don't seem to play well with other fonts/symbols in the text.
> More writing notes: In Section 6: "Now we compare"... Perhaps "This section compares transformers and RNNs...". Immediately after, the sentence in 370 lacks a "been".
>
> [Reply] Thanks for pointing these out. We will make the changes accordingly.
>
> **[Q8]** I find the clean code principle usage very interesting as an assumption, particularly as measured by D, H << L. Could you expand on this? In particular, this assumption should be explicitly explained before, perhaps in the preliminaries.
>
> [Reply] Thank you for your suggestion. We will move the assumption to preliminaries.
>
> The assumption is critical (also see discussions in As discus Yao et al. (2021)). The transformer architecture is about "finite-depth" computation. Theoretically speaking, transformers can only express functions in AC0. Empirically speaking, transformers without chain of thought tend to perform badly for more complicated math and coding tasks.
>
> From a practical viewpoint, bounded depth is fairly reasonable. The Linux kernel contains tens of millions of lines of code, however, most functions are pretty short. This is for better readability. For humans, deeply nested code is way harder to understand. The depth roughly speaking is upper bounded by the maximal length of a single function or type definition, which is usually bounded despite the growth of the codebase.
>
>
> **[Q9]** The setting of the appendix is quite uncommon. Appendices typically contain either additional results/experimental details in empirical papers, or additional proofs in theoretical papers. It's odd to see, for example, the explanation of a very basic concept in computer science such as trees in Appendix A. Appendix D.2 is also uncommon, it could be an interesting alternative introduction to the paper, or perhaps a blogpost about the paper. I'm not saying they need to be removed, but it's unusual. Then, there are some important concepts, perhaps critical to understand the paper, that due to space limitations need to be in the Appendix. As I said above, while in my view this paper should be accepted to COLM, perhaps it's not the optimal venue.
>
>
> [Reply]  We believe COLM is one of the best venues to present our findings on transformers. We will have an arXiv version that lists all major contributions in the main text.
>
> We didn't intend to explain the concept of trees in Appendix A. Rather, we aim to provide a clear formalization of the concept of trees compatible with type notions that are essential for the integrity of our proof.
>
> **[Q10]** 288: readility -> readability
>
> **[Q11]** 379: "what RNNs requires” -> require
>
> Thanks for pointing out. We will fix these accordingly.
>
> **[Q12]** Proofs would definitely benefit from more intuitive summaries or diagram. The only diagram in the paper (at least the main body) uses real estate that, given the tightness of the space requirements, perhaps should have been spent explaining concepts that are more difficult to understand?
>
>
> [Reply]  We appreciate the reviewer’s suggestion regarding diagrams and intuitive summaries. Given the unusual nature of our proof, we agree that certain parts would benefit from visual explanation. We specifically plan the following:
>
> * **Section 5.1 (AST Construction):** We will add a diagram illustrating how a transformer constructs an AST by representing trees as sequences using the arena pattern (see Appendix A.2). This will visually clarify how the transformer layers map token positions to tree structures under the bounded depth assumption.
>
> * **Section 5.2 (Symbol Resolution):** We will include a diagram based on the Mini-Husky example (page 5), showing how variable bindings are resolved using attention heads (e.g., nearest-left) to select the correct variable under lexical scoping. This connects directly to Proposition 15 (page 32) and demonstrates the utility of hard attention for structured name resolution.
>
> These targeted additions aim to improve clarity while respecting space constraints.
>
> [1] https://openreview.net/forum?id=444ug_EYXet
>
> [2] https://openreview.net/forum?id=9S7osCL5Q8

---

> ### Author Response · Authors · 2025-06-01
> **Respon to Reviewer a8E2 (1/2)**
>
> We appreciate your thoughtful reviews.
>
> ### Reasons To Reject:
>
> **[R1]** The paper is not very well-motivated: The paper itself doesn't make a compelling case for LLM-based compilation.
>
>
> [Reply] Compilers can be viewed as a clean-to-formalize task of code understanding, which underpins applications such as Github Copilot, cursor, Devin, lovable, etc. The amazing coding capabilities of LLMs, including auto-completion, code generation, and debugging, require basic capabilities of syntax parsing and type checking, as included in compilers.
>
> **[R2]** Not handling code generation: in compilation, generating code is at least as important as analyzing the input code. This approach is limited in that it only tackles half of the problem, while e.g. the title and abstract seem to imply the work tackles compilation in general.
>
> [Reply] Thank you for your suggestion. We will make the abstract clearer. The code generation part is way harder to formalize. For example, the user might want the LLM to write code for a certain algorithm, which LLM can do due to training on a large corpus and is not easily characterized clearly.
>
> **[R3]** Lack of intuitive insights: The paper presents interesting devices (theoretical, + practical if we account for the actual implementation of its empirical results in Figure 2). However, after reading the paper, I find a lack of insights. A concrete result is the lower bound in 6.1, and the empirical results in line 403, corroborate that BERT models perform better at type checking. But there is no intuitive explanation on the implications of these results.
>
> [Reply] One intuition is that transformers behave like a trainable memory (which we tried to formalize). It's easier to keep information intact in transformers than RNNs. For type checking, it's critical that type signatures of functions are kept intact. We will add more discussions in the final versions.
>
> **[R4]** The paper has a very long appendix, and it's not only complementary information. Some of it seems to be critical to understand more deeply the contribution of the paper. Perhaps this should have been a journal paper, rather than a conference paper? The paper itself seems to be sketching higher-level points while key points are offloaded to the appendix.
>
> [Reply] Thanks for your suggestion. We believe COLM is one of the best venues to present our findings on transformers. We will have an arXiv version that lists all major contributions in the main text.
>
> ### Questions To Authors:
>
> **[Q1]** Missing citation of early work (2021) introducing neural compilation [1]. There's also [2] as a relevant missing citation.
>
> [Reply]
> We thank the reviewer greatly for pointing out these important prior works. We shall add them to our paper.
>
> The first one "Learning C to x86 Translation: An Experiment in Neural Compilation" trains and evaluates a sequence-to-sequence Transformer to output x86 assembly from C code. It shows the empirical result of using transformers to approximate actual compilers, aligning with the theoretical results of our paper.
>
> The second one "Enabling Transformers to Understand Low-Level Programs" is about transformer understanding of low-level languages. However, the topic of this paper is more about transformers understanding high-level languages. It's relevant but not directly aligned.
>
>
>
> **[Q2]** Will the code be open-sourced?
>
> [Reply] Yes. We've open-sourced the code in the supplementary materials.
>
> **[Q3]** Even if also used in previous work, can you comment on the potential limitations of using hard attention in your theoretical reasoning? Similarly, can you comment on using encoder-only Transformers?
>
> [Reply] We use hard attention for theoretical clarity. It approximates soft attention when the attention gap is large, as shown in prior work (e.g., Yao et al., 2021). This simplification enables precise proofs while retaining practical relevance. As for encoder-only Transformers, our choice depends on the task—syntactic and semantic analysis are naturally framed as sequence-to-sequence mappings over input tokens. Our analysis applies to general Transformers, but we instantiate encoder-only models in experiments for concreteness.
>
> **[Q4]** Small note: Compiler literature typically refers to "semantic analysis" rather than information (re Figure 1).
>
> [Reply] Thank you for your suggestion. We will make the change accordingly.
>
> **[Q5]** sentence in line 195: "The raw text is firstly segmented into parts like literals, identifiers, punctuations, keywords, etc, called token stream, then parsed into a tree-like structure representation generated from the input, finally syntactic and semantic analysis is performed on the tree." doesn't read well.
>
> [Reply] Thank you for your suggestion. We will improve the writing accordingly.

---

> > ### Comment · Reviewer_a8E2 · 2025-06-11
> > **Reply to authors' rebuttal**
> >
> > [R1] I'm convinced by your explanation. But as a reader of the paper, I don't necessarily get that impresssion. Perhaps consider making more clear the motivation in the paper.
> > [R2] I understad, thank you for the clarification. But as you said, you should acknowledge that more explicitly in the paper.
> > [R3] Thank you for the clarification, I do find that compelling. The paper would benefit from more intuitive explanations on the implications of the findings.
> > [R4] That's a valid view.
> >
> > [Q1] The paper will benefit from adding those citations. Thanks for the contextualization.
> > [Q2] That's great news.
> > [Q3] Thank you for the clarification.
> > [Q4] Ok.
> > [Q5] Ok.
> >
> > I'm happy with the clarifications and comments and I definitely recommend this paper for acceptance. I'm updating my confidence score given these clarifications.

---

### Official Review · Reviewer_UePS · 2025-05-13

**Rating:** 8
**Confidence:** 2
**Ethics Flag:** 1

**Summary:**

The paper investigates the expressive power of transformers as compilers by introducing a representative programming language, Mini-Husky. Another main contribution of the paper is introducing Cybertron, a domain-specific language that generates formal proofs of the transformer’s expressive power for compiler tasks. Three programming language processing tasks are studied in the paper, namely, abstract syntax tree construction, symbol resolution, and type analysis (including type inference and checking).
The paper also provides a proof that if the input code sequence has bounded depth in both Abstract Syntax Tree and type inference, then the number of parameters required by transformers depends only on the logarithm of the input sequence length to handle compilation tasks, such as AST construction, symbol resolution, and type analysis.

The paper provides a Mini-Husky-based testbed for compilation tasks as well as Cybertron, which is useful for the community. The empirical experiments, in addition to the theoretical proofs, further strengthen the paper. The paper provides strong proof of the expressive power of transformers for compilation tasks, given the fact that several works in the past have experimented with AST generation models, and is significant to the community.
The paper's clarity will benefit from elaborating on the dataset construction, model training, and evaluation (section 6.2).

**Questions To Authors:**

* It would be really helpful if the authors can elaborate more on the dataset construction and model training and evaluation. Are the experiments only conducted on type accuracy?
* Authors mention that AST-depth D, is generally small in practice. Could the authors provide more context here, if this holds true for specific programming languages etc.?
* Could the authors elaborate what they mean in L297, "We note log L is small because 64-bit computers can only process context length at most $2^64$"
* Minor suggestion: The x-axis labels can be rotated by 45 degrees to avoid the overlap between x-ticks text.
* A short note on future directions of this work would be highly appreciated.

**Reasons To Accept:**

* The paper investigates the expressive power of transformers for compilation tasks which is highly relevant from the perspective of understanding the capabilities of transformers.
* The paper presents a significant theoretical result that for depth bounded input code sequences, there exists a transformer encoder with model dimension and number of layers depending on log input sequence length plus depth, that can generate the abstract syntax tree and symbol resolution sequences.
* The development of Cybertron, as a proof of vehicle is rigorous and lays the foundation of using it for future formal proofs for various other compilation tasks.

**Reasons To Reject:**

I do not have any strong reasons to reject the paper. Few comments on clarity and a few questions are mentioned in the next section.

---

> ### Author Response · Authors · 2025-06-01
> **Respon to Reviewer UePS**
>
> We sincerely thank the reviewer for the constructive comments.
> ### Questions To Authors:
>
> **[Q1]** It would be really helpful if the authors can elaborate more on the dataset construction and model training and evaluation. Are the experiments only conducted on type accuracy?
>
> [Reply] For dataset construction, we sequentially generate the functions. For each function, the number of arguments is randomly chosen, and each argument's type is also randomly chosen. In the function body, an arbitrary number of previously defined functions are called. We ensure that the number of arguments passed into these calls is correct, so the compiler only needs to check their types. The dataset is mapped to token space for transformers and RNNs to handle, and the labels of types are on the tokens of arguments.
> Training is standard token classification. The evaluation metric is whether the argmax corresponds to the correct type.
> In the final experiments, we only present the results on type accuracy. In preliminary experiments, we also tested AST construction and symbol resolution, but the separation in these tasks is not as significant as that in type analysis.
>
> **[Q2]** Authors mention that AST-depth D, is generally small in practice. Could the authors provide more context here, if this holds true for specific programming languages etc.?
>
>
> [Reply] This holds for all programming languages. The assumption is critical (also see discussions in As discus Yao et al. (2021)). The transformer architecture is about "finite-depth" computation. Theoretically speaking, transformers can only express functions in AC0. Empirically speaking, transformers without chain of thought tend to perform badly for more complicated math and coding tasks.
>
> From a practical viewpoint, bounded depth is fairly reasonable. The Linux kernel contains tens of millions of lines of code, however, most functions are pretty short. This is for better readability. For humans, deeply nested code is way harder to understand. The depth roughly speaking is upper bounded by the maximal length of a single function or type definition, which is usually bounded despite the growth of the codebase.
>
> **[Q3]** Could the authors elaborate what they mean in L297, "We note log L is small because 64-bit computers can only process context length at most $2^{64}$"
>
>
> [Reply] We thank the reviewer for pointing this confusing sentence out. In Line 297, we stated that “$\log L$ is small because 64-bit computers can only process context length at most $2^{64}$.” We clarify the reasoning as follows:
>
> Modern 64-bit computers use 64-bit pointers to address memory. This means that the maximum length of any directly indexable data structure — such as an input sequence or array — is bounded by $2^{64}$, since each index must fit in a 64-bit word. In the context of transformer models, where each token position is typically indexed for attention and positional encoding, this sets a hard upper bound on the possible context length $L$.
>
> Therefore, $\log L \leq \log(2^{64}) = 64$, and in practice it is even smaller due to physical memory constraints and software limitations. As a result, it is reasonable to treat $\log L$ and polylogarithmic factors as constants in complexity analysis for real-world systems.
>
> We will revise the sentence to clarify this point. A more accurate phrasing is:
>
> > “We note that $\log L$ is at most 64 in practice, since on a 64-bit machine, a single sequence of length greater than $2^{64}$ cannot be represented or processed due to address space limits.”
>
> **[Q4]** Minor suggestion: The x-axis labels can be rotated by 45 degrees to avoid the overlap between x-ticks text.
>
> [Reply] Thanks for this suggestion! We will make the change accordingly.
>
> **[Q5]** A short note on future directions of this work would be highly appreciated.
>
> [Reply] In this paper, we only discuss the most basic kind of static code analysis. One interesting direction is to see whether we can generalize to broader types of static analysis, including things like async. Second, we only studied code analysis, not code generation. It would be interesting to try to formalize practical code generation and try to prove transformers' capabilities.

---

> > ### Comment · Reviewer_UePS · 2025-06-07
> >
> > Thank you for the response to the questions. I believe that the response to the questions is helpful and should be added to the paper for improved clarity. The answers improve the overall clarity of the paper. Since my original evaluation was already positive, I will retain my score.

---

### Official Review · Reviewer_7WyJ · 2025-05-13

**Rating:** 6
**Confidence:** 3
**Ethics Flag:** 1

**Summary:**

The authors explore using transformer models as compilers. To do so, they
introduce a new DSL named Mini-Husky, which is a simplified C-like lanaguage,
and explore their results in the context of this language; they hope it
becomes a new standard for such work.

They explore the expressivity of Transformers, showing that input code with
bounded depth can be parsed in a number of parameters logarithmic in sequence
length. RNNs, in contrast, require a number of parameters linear in sequence
length.

They tackle a number of tasks on program inputs, including:
* Abstract Syntax Tree construction
* Symbol resolution
* Type analysis

They also introduce a domain-specific language Cybertron to use in such
proofs.

Both transformer and RNN approaches are evaluated on a set of synthetic data.

The main paper presents a summary of the key results, though the details and
proofs are mostly pushed to a very long set of appendices.

**Questions To Authors:**

Lines 231 to 235 talk about variables `a`, but the description is confusing.
Instead of just saying "the first is accessible", maybe say "`a` from line 4
is accessible", perhaps?

Is the assumption of bounded depth ASTs reasonable?

Is it possible to evaluate on real code rather than synthetic? Do the results
scale to larger codebases?

MiniHusky seems to lack features commonly used in modern languages such as
exception handling and async support, which may introduce some non-local
movements. Do the results presented here extend to those cases?

Will SGD find the transformer parameters that your proofs assume? How
learnable are they?

**Reasons To Accept:**

It's interesting to capture some of the power of transformers as well as the
systematic differences with recurrent architectures. The Cybertron DSL
introduced in the paper seems to provide a useful formalism for analysis. The
authors have clearly placed a good deal of effort in developing this formalism
and it feels potentially reusable.

The focus here on parameter efficiency, not just expressive power, is an
interesting step beyond prior work.

The authors do perform an empirical evaluation in Section 6 that demonstrates
the gap between RNNs and Transformers -- good to see theory backed up in
practice.

Cybertron introduces capabilities not present in prior DSLs such as RASP,
including things like algebraic types, computation graphs, and higher order
composition. This may be useful for additional tasks.

**Reasons To Reject:**

From a very practical standpoint I'm not sure the value of this. We can already see in shipped
systems (Github Copilot, cursor, Devin, lovable, etc.) that LLMs have the
capacity to parse and author well-structured code. Although it's nice to have
some theoretical underpinning, I'm not sure how it'll change the field going
forward.

The analysis focused on hard attention, but most (nearly all?) real systems
use soft attention. Minor difference perhaps, but not clear to me how much
difference that makes in analysis.

IMO, the paper would be better presented as a long-form journal, rather than a
conference paper with huge appendices. The current presentation pushes many of
the contributions outside the main paper.

Bounded depth is potentially problematic for real codebases -- sometimes real
code can go quite deep (e.g. with macros, templating, complex types, etc.)

Cybertron may not have that much practical value, and there are limited
competitive benchmarks.

The empirical evaluation is on synthetic data rather than real code -- we
might not read too much into these results.

The RNN baseline is somewhat outdated these days. Modern competitors such as
MAMBA were not considered; would be good to have some analysis of other
architectures.

From a novelty standpoint, there are gains here, but there is prior work in
neural program synthesis, for instance. Is this really the first paper "to show
transformers are efficient compilers"?

---

> ### Author Response · Authors · 2025-06-01
> **Respon to Reviewer 7WyJ (2/2)**
>
> ### Questions To Authors:
>
> **[Q1]** Lines 231 to 235 talk about variables a, but the description is confusing. Instead of just saying "the first is accessible", maybe say "a from line 4 is accessible", perhaps?
>
>
> [Reply] Thank you for your suggestion! We will modify it as you suggested.
>
> **[Q2]** Is the assumption of bounded depth ASTs reasonable?
>
> [Reply] The assumption is critical (also see discussions in As discus Yao et al. (2021)). The transformer architecture is about "finite-depth" computation. Theoretically speaking, transformers can only express functions in AC0. Empirically speaking, transformers without chain of thought tend to perform badly for more complicated math and coding tasks.
>
> From a practical viewpoint, bounded depth is fairly reasonable. The Linux kernel contains tens of millions of lines of code, however, most functions are pretty short. This is for better readability. For humans, deeply nested code is way harder to understand. The depth roughly speaking is upper bounded by the maximal length of a single function or type definition, which is usually bounded despite the growth of the codebase.
>
> **[Q3]** Is it possible to evaluate on real code rather than synthetic? Do the results scale to larger codebases?
>
> [Reply] We want to clarify that our paper is theoretically focused, and the experiments only serve to validate our theory. Here, using synthetic gives a clean, controllable setting to compare transformers and RNNs.
>
> **[Q4]** MiniHusky seems to lack features commonly used in modern languages such as exception handling and async support, which may introduce some non-local movements. Do the results presented here extend to those cases?
>
>
> [Reply] We note that non-local movements only occur in runtime, not in compile time. For example, in Rust, the async function signatures are more complicated but still follow the same code analysis pipeline as ordinary functions. We didn't exactly prove this in our paper due to the limit of scope, but we strongly believe this wouldn't be too hard to do.
>
> **[Q5]** Will SGD find the transformer parameters that your proofs assume? How learnable are they?
>
> [Reply] This paper only studies the expressive power of transformers. The optimization analysis of deep neural networks is notoriously difficult and beyond the scope of this paper: e.g., it is a decade-old open problem of proving SGD can recover a two-layer ReLU teacher network.

---

> > ### Comment · Reviewer_7WyJ · 2025-06-09
> >
> > Thanks for addressing my questions.

---

> ### Author Response · Authors · 2025-06-01
> **Response to Reviewer 7WyJ (1/2)**
>
> We sincerely appreciate the reviewer for the constructive comments on our work!
>
> ### Reasons To Reject:
>
> **[R1]** From a very practical standpoint I'm not sure the value of this. We can already see in shipped systems (Github Copilot, cursor, Devin, lovable, etc.) that LLMs have the capacity to parse and author well-structured code. Although it's nice to have some theoretical underpinning, I'm not sure how it'll change the field going forward.
>
>
> [Reply] Copilot, Devin, and similar tools generate or modify code in ways that require nontrivial code understanding—like completing partial programs, refactoring, or inserting type-correct expressions. These are not just pattern-matching tasks; they often rely on parsing syntax, resolving variables, understanding scopes and types, and preserving program semantics.
> These tasks align closely with what the frontend of a compiler does: parsing, name resolution, type checking, etc. So the compiler frontend provides a *formal abstraction* of code understanding. By studying whether and how neural networks like transformers can simulate this frontend under precise assumptions, we clarify what structural capabilities are required for code understanding. Our results show that transformers can express these capabilities in a principled way, whereas RNNs fundamentally cannot.
>
>
> **[R2]** The analysis focused on hard attention, but most (nearly all?) real systems use soft attention. Minor difference perhaps, but not clear to me how much difference that makes in analysis.
>
>
> [Reply] While our analysis uses hard attention, this serves as a principled approximation to soft attention in the regime where one attention logit dominates — that is, when the gap between the largest logit and the others is large. Specifically, if the logits are $\{a_i\}$, softmax assigns weight $\exp(a_i) / \sum_j \exp(a_j)$; as the difference $a_{\text{max}} - a_j$ grows, the softmax weight on non-max positions decays exponentially. In the limit, soft attention converges to a one-hot distribution — hard attention. This approximation is widely used in theoretical work (e.g., Yao et al., 2021) and reflects empirical behavior when attention distributions are sharply peaked, as often observed in real models due to scaling or learned sparsity.
>
>
>
> **[R3]** IMO, the paper would be better presented as a long-form journal, rather than a conference paper with huge appendices. The current presentation pushes many of the contributions outside the main paper.
>
> [Reply] Thanks for your suggestion. We believe COLM is one of the best venues to present our findings on transformers. We will have an arXiv version that lists all major contributions in the main text.
>
>
> **[R4]** Bounded depth is potentially problematic for real codebases -- sometimes real code can go quite deep (e.g. with macros, templating, complex types, etc.)
>
>
> [Reply] It is true that real code can go quite deep. In fact, one can write code that is NP-hard to analyze. From a theoretical computer science point of view, transformers by themselves can only express functions in AC0. Thus it's not surprising that bounded depth is needed, as also used in Yao et al. (2021).
>
> **[R5]** Cybertron may not have that much practical value, and there are limited competitive benchmarks.
>
> [Reply] We want to clarify that Cybertron is mostly a tool for theoretical analysis, and thus its importance is hard to characterize through empirical benchmarks.
>
> **[R6]** The empirical evaluation is on synthetic data rather than real code -- we might not read too much into these results.
>
> [Reply] We want to clarify that our paper is theoretically focused, and the experiments only serve to validate our theory. Here, using synthetic gives a clean, controllable setting to compare transformers and RNNs.
>
> **[R7]** The RNN baseline is somewhat outdated these days. Modern competitors such as MAMBA were not considered; would be good to have some analysis of other architectures.
>
>
> [Reply] We agree that recurrent models like MAMBA are of current interest. However, our work is theoretical in nature and aims to establish foundational capabilities of transformer-style architectures. We chose a basic RNN as a baseline to highlight fundamental differences rather than to benchmark against the latest models.
>
> **[R8]** From a novelty standpoint, there are gains here, but there is prior work in neural program synthesis, for instance. Is this really the first paper "to show transformers are efficient compilers"?
>
>
> [Reply] Indeed there are prior works in neural program synthesis. But their focus is empirical, without theoretical analysis. We are the first to show theoretically tasks close to actual compilers can be done by transformers.

---

> > ### Comment · Reviewer_7WyJ · 2025-06-09
> >
> > Thank you for your detailed and careful responses. I think we have a shared understanding of the contributions of this paper.

---

### Decision · Program_Chairs · 2025-07-08

**Decision:**

Accept

**Comment:**

The paper explores the expressivity of transformers, using a new formalism called Cybertron, which extends prior work on Rasp by enabling proofs about more complex algorithms. Reviewers are generally positive, with the caveat that the contributions are at this stage restricted to the theoretical side.

One concern is the assumption of bounded depth programs. The authors may wish to consider an extension of their analysis to universal transformers, which can potentially handle this issue [1].

[1] Shaw et al, 2024. ALTA: Compiler-Based Analysis of Transformers. https://arxiv.org/abs/2410.18077